# Mixture of Dynamical Variational Autoencoders for Multi-Source Trajectory Modeling and Separation

**Xiaoyu Lin**                                                            *xiaoyu.lin@inria.fr*
*Inria @ Univ. Grenoble Alpes, LJK, CNRS, France*

**Laurent Girin**                                                 *laurent.girin@grenoble-inp.fr*
*Univ. Grenoble Alpes, Grenoble-INP, GIPSA-lab, France*

**Xavier Alameda-Pineda**                                  *xavier.alameda-pineda@inria.fr*
*Inria @ Univ. Grenoble Alpes, LJK, CNRS, France*

**Reviewed on OpenReview:** *https: // openreview. net/ forum? id= sbkZKBVC31*

## Abstract

In this paper, we propose a latent-variable generative model called *mixture of dynamical variational autoencoders* (MixDVAE) to model the dynamics of a system composed of multiple moving sources. A DVAE model is pre-trained on a single-source dataset to capture the source dynamics. Then, multiple instances of the pre-trained DVAE model are integrated into a multi-source mixture model with a discrete observation-to-source assignment latent variable. The posterior distributions of both the discrete observation-to-source assignment variable and the continuous DVAE variables representing the sources content/position are estimated using a variational expectation-maximization algorithm, leading to multi-source trajectories estimation. We illustrate the versatility of the proposed MixDVAE model on two tasks: a computer vision task, namely multi-object tracking, and an audio processing task, namely single-channel audio source separation. Experimental results show that the proposed method works well on these two tasks, and outperforms several baseline methods.

## 1 Introduction

### 1.1 Latent-variable generative models: From GMMs to DVAEs

Latent-variable generative models (LVGMs) are a very general class of probabilistic models that introduce latent (unobserved) variables to model complex distributions over the observed variables. Depending on the type of considered latent variables, different LVGM structures can be defined. *Mixture models*, such as the Gaussian Mixture Model (GMM), are LVGMs with a discrete latent variable (McLachlan & Basford, 1988). They are widely used in various pattern recognition and signal processing tasks to model the distribution of a signal that can take several different states, each state being encoded by a different value of the latent variable. The (marginal) distribution of the observed data is modeled as a linear combination of component distributions, which are conditioned on the latent variable (in the case of GMM, these are Gaussian distributions), each corresponding to a different possible state. The mixing coefficients represent the prior distribution of the discrete latent variable and determine the relative weight of each component.

Simple mixture models are appropriate to model 'static' data since they do not consider possible temporal correlations in data sequences. More sophisticated LVGMs can be designed to model the dynamics, i.e. the temporal dependencies, of sequential data. First, mixture models can be generalized by applying a Markov model on the latent variable at different time steps, resulting in a hidden Markov model (HMM) (Rabiner & Juang, 1986). For example, when applying a first-order Markov model, the prior distribution of the latent

variable at time $t$ depends on the latent variable at time $t-1$ via a transition matrix. The mixture component, often called the *observation model* in this context, is conditioned on the latent variable at time $t$. A variety of different conditional distributions can be used. For example, the observation model can be a Gaussian (as in the GMM) or it can be itself a GMM (i.e., one GMM for each state). The latter HMM-GMM combination has been the state-of-the-art in automatic speech recognition for a decade in the pre-deep-learning era (Yu & Deng, 2016). Another notable extension of HMMs is the factorial HMMs (Ghahramani & Jordan, 1995), which consider a set of parallel factorial discrete latent variables instead of a single one. Each latent variable follows a Markov model and has its own transition matrix. The conditional distribution of the observed variable at time step $t$ is a Gaussian distribution with the mean being a weighted combination of the latent variables at the same time step, and with a common covariance matrix. Factorial HMMs are suitable for modeling sequential data generated by the interaction of multiple independent processes.

A discrete latent variable can only be used to model categorical latent generative factors. If we consider that the latent state evolves continuously and is better represented by a continuous latent variable, we enter in the world of continuous dynamical models and state space models (SSMs) (Aoki, 2013). The simplest and most commonly used continuous dynamical model is the linear(-Gaussian) dynamical system (LDS), in which the distribution of the latent variable at time step $t$ is a Gaussian with the mean being a linear function of the latent variable at time step $t-1$ (Ghahramani & Hinton, 1996). If the observation model is also Gaussian with the mean being a linear function of the latent variable at time $t$, a famous analytical solution[1] for the LDS is the Kalman filter (Kalman, 1960). However, the linear-Gaussian assumption can be a significant limitation for real-world signals with complex dynamics. A widely-used generalization of the Kalman filter to sequential data with non-linear dynamics is the extended Kalman filter (Einicke & White, 1999; Zarchan, 2005), which is a first-order Gaussian approximation to the Kalman filter based on local linearization using the Taylor series expansion. Another interesting extension is to combine a set of LDSs with an HMM, resulting in the switching state space model (also named switching Kalman filter) (Murphy, 1998; Ghahramani & Hinton, 2000). This model segments the sequential data into different regimes, each regime being modeled by an LDS, and the succession of regimes is ruled by the HMM. Recently, deep neural networks (DNNs), and in particular recurrent neural networks (RNNs), have been used within LVGM structures to model sequential data. In this line, the dynamical variational autoencoders (DVAEs) (Girin et al., 2021) are a family of powerful LVGMs that extend the famous variational autoencoder (VAE) (Kingma & Welling, 2014; Rezende et al., 2014) to model complex non-linear temporal dependencies within a sequence of observed data vectors and corresponding (continuous) latent vectors. DVAEs have been successfully applied on different types of sequential data such as speech signals (Bie et al., 2021) and 3D human motion data (Bie et al., 2022a).

## 1.2 Contribution: Multi-Source Mixture of DVAEs – Model and solution

All the LVGMs discussed so far have been used to model the distribution of single-source data, the source being either static and have several possible states (in the case of mixture models) or sequential with different types of underlying dynamics (e.g., locally linear). In real-life scenarios, we often encounter situations where a number of sequential source signals appear concurrently in a natural scene for a certain period of time and are observed jointly. Each underlying source can have its own dynamics, and the problem is to obtain an estimation of the content and/or the position along time of each source separately, which includes consistently recovering the identity of each source over time. In short, we want to estimate and separate each source's trajectory from a set of mixed-up observations.

In this paper, we propose to tackle this problem within a deep LVGM probabilistic framework, with a model combining the following two bricks: (a) a deep LVGM for modeling the dynamics of each source independently; in this work, we propose to use a DVAE to model each individual source. The DVAE-generated random vector represents the source vector, i.e. the source content/position that we want to track over time, and the latent random vector represents the underlying (continuous) hidden state/factor that governs the source dynamics. (b) A discrete latent assignment variable which assigns each observation in the set of (mixed-up) observations to a source. We name the resulting model as Multi-Source Mixture of DVAEs (MixDVAE). In addition to the MixDVAE model, we propose a multi-source trajectory estimation algorithm (i.e., a solution to the MixDVAE model). Importantly, this estimation method does not require a

---

[1]In the present context, a model solution is an algorithm for model parameters estimation and latent variables inference.

massive multi-source annotated dataset for model training. Instead, we first pre-train the DVAE model on an unlabeled (synthetic or natural) single-source trajectory dataset, to capture the dynamics of an individual source type. Afterwards, the pre-trained DVAE is plugged into the MixDVAE model together with the observation-to-source assignment latent variable to solve the problem for each multi-source test data sequence to process. For each test data sequence, the (approximate) posterior distributions of both the observation-to-source assignment variable and the source vector of each source are derived using the variational inference methodology – more specifically, we propose a variational expectation-maximization (VEM) algorithm (Jordan et al., 1999; Bishop, 2006; Wainwright et al., 2008).

The proposed model and method are versatile in essence. They can be easily adapted and applied to a variety of estimation problems with multiple dynamical sources with different configurations. For example, if all sources are assumed to have similar dynamics, a single DVAE model can be used to model all sources (more specifically, a different instance of the same DVAE model can be used for each source) and only one pre-training is made on a single single-source dataset. If different types of sources are considered, with different dynamics, one can use different instances of the same DVAE model, but pre-trained on different single-source datasets, or one can use (different instances of) different DVAE models, also pre-trained on different single-source datasets. In any case, as stated above, there is no need for a massive dataset containing annotated mixtures of simultaneous sources, as would be the case with a fully-supervised approach. Labeled multi-source datasets must be much larger than single-source datasets, since the mixture process intrinsically multiplies the content diversity, and thus can be very costly and difficult to obtain. Therefore, our method can be considered as data-frugal and weakly supervised compared to a fully-supervised method. One limitation of the proposed model, though, is that the VEM algorithm applied at test time is relatively costly in computation (this point is investigated in our study). Another limitation is the fact that all sources are assumed to behave independently. In other words, the proposed MixDVAE does not explicitly model the possible interactions between the different sources. This is planned for future work. We illustrate the versatility of MixDVAE by applying it to two notably different tasks, in computer vision and in audio processing – namely multi-object tracking (MOT) and single-channel audio source separation (SC-ASS) – and we report corresponding experimental results.

In short, the contributions of this paper are:

- A generic latent-variable generative model called MixDVAE for the separation of mixed observations into independent sources with non-linear dynamics.

- A learning and inference (variational EM) algorithm associated with MixDVAE, derived from the corresponding variational lower bound.

- A set of experiments demonstrating the interest of MixDVAE for various tasks and using different types of data.

- MixDVAE is data-frugal and weakly supervised since it does not require a massive labeled multi-source dataset at training time, but only one or several single-source dataset(s) of much moderate size.

### 1.3 Application to multi-object tracking

In multi-object tracking (MOT) (also called multi-target tracking (MTT) depending on the context, scientific community and applications), a number of objects/targets appear simultaneously in the same visual scene and each of them has its own moving trajectory, e.g., pedestrians/vehicles in videos or aircrafts in radar scans. The problem is to estimate the position of each object/target at every time step, and track it along time by assigning a unique identity to it (Vo et al., 2015; Ciaparrone et al., 2020). In the popular 'tracking-by-detection' configuration, a set of detection bounding boxes (DBBs) are given at each time step by a front-end detection algorithm, each of them potentially corresponding to one of the targets. These DBBs are then used as the observations. In Section 5, we apply the MixDVAE model to the MOT problem in this configuration (i.e., using the set of DBBs as observations). We do that in a simplified scenario where the number of objects is assumed known and constant during the observation measurements. A complete and fully-operational MOT system would require to include a module managing the 'birth' and 'death' of target

tracks (i.e., objects disappearing of the scene and new objects appearing in the scene). We do not address this problem since the purpose of this work is not to propose a fully-operational MOT system but rather to focus on the problem of multi-source dynamics modeling with DVAEs. It must be noted however that even if we assume that the actual number of objects present in the scene is known and does not vary across the modeled sequence, at any time step $t$, it is not necessarily equal to the number of DBBs, since occlusions (leading to missed detections) may occur. We will see in our reported experiments that MixDVAE is able to deal with these difficulties.

**Related work in MOT/MTT.** In the MOT/MTT literature, few works have considered the detection assignment problem (which is also called the data association problem) and the target dynamics modeling problem jointly in a unified probabilistic framework. In fact, the data association problem is usually solved by designing a complicated association algorithm (such as the joint probabilistic data association filter (JPDAF) or the multiple hypothesis tracking (MHT) algorithm) and the target trajectories estimation is obtained by applying a post-processing track filtering algorithm (such as the Kalman filter, the extended Kalman filter, or a particle filter) separately on each estimated track (Vo et al., 2015). Recent MOT algorithms use DNNs such as RNNs or convolutional neural networks (CNNs) to extract several features (e.g., visual features, motion features, targets interaction features) from the videos and use these features for data association (Luo et al., 2021; Ciaparrone et al., 2020). The motion models are generally used for track refinement. The work that is the closest to ours is that of Ban et al. (2021), who proposed a unified probabilistic framework for audio-visual multi-speaker tracking. However, the dynamical model that was used in their work is a simple linear-Gaussian dynamical model. In this paper, we use DVAEs to model non-linear objects dynamics.

### 1.4 Application to audio source separation

The second problem on which we apply the proposed MixDVAE model is the single-channel audio source separation (SC-ASS) task. Here, the recorded (observed) signal is a unique (digital) waveform that is assumed to result from the physical summation of individual source waveforms, such as several speakers speaking simultaneously or several musical instruments playing together. The goal is to estimate the different source signals composing the mixture (Vincent et al., 2011). At first sight, this scenario is not an appropriate configuration for MixDVAE because, for each discretized time, we do not have a set of observed samples to assign to one of the source signals. However, a widely-used approach in SC-ASS is to work in the time-frequency (TF) domain, most often using the short-time Fourier transform (STFT), and exploit the *sparsity* of audio signals in the TF domain (Yilmaz & Rickard, 2004). This means that at each TF bin of the STFT, the observed mixture signal is assumed to be composed of one dominant source (with a power that is much larger than that of the other sources), so that the observation at that TF bin can be (totally or mainly) attributed to that dominant source. This can be done using an assignment variable, which is often referred to as a *TF mask* in the SC-ASS literature (Wang & Chen, 2018). Therefore, we can apply the proposed MixDVAE model by combining this assignment variable with a set of DVAEs modeling the dynamics of the audio sources in the TF domain. The main difference compared to the MOT problem is that we have to consider the frequency dimension in addition to time, and at a given TF bin, we have here only one single observation to assign to a source, instead of a set of observations. We apply this principle and illustrate the use of MixDVAE for SC-ASS in Section 6. It can be noted that the dynamics of different types of audio source signals (speech, musical instruments, noises, etc.) can be very different. So, this is a typical use-case where we can pre-train different DVAE models on different single-source datasets to capture the dynamics of different types of source.

**Related work in SC-ASS.** State-of-the-art SC-ASS methods are based on the use of huge and sophisticated DNNs that directly map the mixture signal to the individual source signals or to the TF masks (Chandna et al., 2017; Wang & Chen, 2018). These methods obtain impressive separation performance, but they require to adopt a fully-supervised approach using huge parallel datasets (that contain both the mixture and the aligned separate source signals). This contrasts with the spirit of MixDVAE which, again, can be considered as weakly supervised, and does not require a huge parallel dataset, but only a reasonable-size single-source dataset for any type of source to separate. Weakly-supervised (or at the extreme unsupervised) methods for audio source separation are still quite rare and their development is a largely open topic. We can find some connection with the SC-ASS method of Ozerov et al. (2009) based on factorial HMMs. A

DVAE-based unsupervised speech enhancement method was recently presented by (Bie et al., 2022b), but a unique DVAE instance was used to model the speech signal and the noise (power spectrogram) was modeled with a nonnegative matrix factorization (NMF) model. To our knowledge, the present work is the first time a mixture of DVAEs is used for SC-ASS (and for audio processing in general).

## 1.5 Organization of the paper

In Section 2, we present the general methodological background for developing the MixDVAE model, including the variational inference principle and the DVAEs. In Section 3, we present the MixDVAE model and the general principle we used to derive its solution. The solution itself, i.e. the MixDVAE algorithm, is presented in Section 4. Section 5 and Section 6 illustrate the application of MixDVAE to the MOT and SC-ASS problems, respectively, including experiments. Section 7 concludes the paper.

## 2 Methodological background

### 2.1 Latent-variable generative models and variational inference

An LVGM depicts the relationship between an observed $\mathbf{o}$ and a latent $\mathbf{h}$ random (vector) variable from which $\mathbf{o}$ is assumed to be generated. We consider an LVGM defined via a parametric joint probability distribution $p_\theta(\mathbf{o}, \mathbf{h})$, where $\theta$ denotes the set of parameters. In a general manner, we are interested in two problems closely related to each other: (i) estimate the parameters $\theta$ that maximize the observed data (marginal) likelihood $p_\theta(\mathbf{o})$, and (ii) derive the posterior distribution $p_\theta(\mathbf{h}|\mathbf{o})$ so as to infer the latent variable $\mathbf{h}$ from the observation $\mathbf{o}$.

A prominent tool to estimate $\theta$ is the family of expectation-maximisation (EM) algorithms (Bishop, 2006; McLachlan & Krishnan, 2007), that maximizes the following lower bound of the marginal likelihood, called the evidence lower-bound (ELBO):

$$\mathcal{L}(\theta, q; \mathbf{o}) = \mathbb{E}_{q(\mathbf{h}|\mathbf{o})}\big[ \log p_\theta(\mathbf{o}, \mathbf{h}) - \log q(\mathbf{h}|\mathbf{o}) \big] \leq \log p_\theta(\mathbf{o}), \tag{1}$$

where $q(\mathbf{h}|\mathbf{o})$ is a distribution on $\mathbf{h}$ conditioned on $\mathbf{o}$. The EM algorithm is an iterative alternate optimisation procedure that maximizes the ELBO w.r.t. the distribution $q$ (E-step) and the parameters $\theta$ (M-step). Maximizing the ELBO w.r.t. $q$ is equivalent to minimising the Kullback-Leibler divergence (KLD) between $q(\mathbf{h}|\mathbf{o})$ and the exact posterior distribution $p_\theta(\mathbf{h}|\mathbf{o})$ (Bishop, 2006; McLachlan & Krishnan, 2007). When $p_\theta(\mathbf{h}|\mathbf{o})$ is computationally tractable, it is optimal to choose $q(\mathbf{h}|\mathbf{o}) = p_\theta(\mathbf{h}|\mathbf{o})$, then the ELBO is tight and the EM is called "exact." Otherwise, the optimization w.r.t. $q$ is constrained within a given family of computationally tractable distributions and the bound is not tight anymore. We then have to step in the variational inference (VI) framework (Jordan et al., 1999; Wainwright et al., 2008).

A first family of VI approaches is the structured mean-field method (Parisi & Shankar, 1988) which consists in splitting $\mathbf{h}$ into a set of disjoint variables $\mathbf{h} = (\mathbf{h}_1, ..., \mathbf{h}_M)$. The approximate posterior distribution $q$ is thus assumed to factorize over this set, i.e. $q(\mathbf{h}|\mathbf{o}) = \prod_{i=1}^{M} q_i(\mathbf{h}_i|\mathbf{o})$, which leads to the following optimal factor, given the parameters computed at the previous M-step, $\theta^{\text{old}}$:

$$q_i^*(\mathbf{h}_i|\mathbf{o}) \propto \exp\left( \mathbb{E}_{\prod_{j\neq i} q_j(\mathbf{h}_j|\mathbf{o})}\big[ \log p_{\theta^{\text{old}}}(\mathbf{o}, \mathbf{h}) \big] \right). \tag{2}$$

Since each factor is expressed as a function of the others, this formula is iteratively applied in the E-step of the EM algorithm until some convergence criterion is met, leading to the VEM family of algorithms (Bishop, 2006).

A second approach is to rely on amortized inference (Hoffman et al., 2013), where a set of shared parameters is used to compute the parameters of the approximate posterior distribution. A very well-known example is the variational autoencoder (VAE) (Kingma & Welling, 2014; Rezende et al., 2014). For a reason that will become clear in Section 3.1, let us here denote by $\mathbf{s}$ the observed variable and by $\mathbf{z}$ the latent one. In a VAE, the joint distribution $p_\theta(\mathbf{s}, \mathbf{z}) = p_\theta(\mathbf{s}|\mathbf{z})p(\mathbf{z})$ is defined via the prior distribution on $\mathbf{z}$, generally chosen as the standard Gaussian distribution $p(\mathbf{z}) = \mathcal{N}(\mathbf{z}; \mathbf{0}, \mathbf{I})$, and via the conditional distribution on $\mathbf{s}$,

generally chosen as a Gaussian with diagonal covariance matrix $p_\theta(\mathbf{s}|\mathbf{z}) = \mathcal{N}\big(\mathbf{s}; \boldsymbol{\mu}_\theta(\mathbf{z}), \text{diag}(\boldsymbol{v}_\theta(\mathbf{z}))\big)$. The mean and variance vectors $\boldsymbol{\mu}_\theta(\mathbf{z})$ and $\boldsymbol{v}_\theta(\mathbf{z})$ are nonlinear functions of $\mathbf{z}$ provided by a DNN, called the *decoder network*, taking $\mathbf{z}$ as input. $\theta$ is here the (amortized) set of parameters of the DNN. The posterior distribution $p_\theta(\mathbf{z}|\mathbf{s})$ corresponding to this model does not have an analytical expression and it is approximated by a Gaussian distribution with diagonal covariance matrix $q_\phi(\mathbf{z}|\mathbf{s}) = \mathcal{N}\big(\mathbf{z}; \boldsymbol{\mu}_\phi(\mathbf{s}), \text{diag}(\boldsymbol{v}_\phi(\mathbf{s}))\big)$, where the mean and variance vectors $\boldsymbol{\mu}_\phi(\mathbf{s})$ and $\boldsymbol{v}_\phi(\mathbf{s})$ are non-linear functions of $\mathbf{s}$ implemented by another DNN called the *encoder network* and parameterized by $\phi$. The ELBO is here given by:

$$\mathcal{L}(\theta, \phi; \mathbf{s}) = \mathbb{E}_{q_\phi(\mathbf{z}|\mathbf{s})}\big[\log p_\theta(\mathbf{s}, \mathbf{z}) - \log q_\phi(\mathbf{z}|\mathbf{s})\big]. \tag{3}$$

In practice, the ELBO is jointly optimized w.r.t. $\theta$ and $\phi$ on a training dataset using a combination of stochastic gradient descent (SGD) and sampling (Kingma & Welling, 2014). This is in contrast with the EM algorithm where $q$ and $\theta$ are optimized alternatively.

## 2.2 From VAE to DVAE

In the VAE, each observed data vector $\mathbf{s}$ is considered independently of the other data vectors. The dynamical variational autoencoders (DVAEs) are a class of models that extend and generalize the VAE to model sequences of data vectors correlated in time (Girin et al., 2021). Roughly speaking, DVAE models combine a VAE with temporal models such as RNNs and/or SSMs.

Let $\mathbf{s}_{1:T} = \{\mathbf{s}_t\}_{t=1}^T$ and $\mathbf{z}_{1:T} = \{\mathbf{z}_t\}_{t=1}^T$ be a discrete-time sequence of observed and latent vectors, respectively, and let $\mathbf{p}_t = \{\mathbf{s}_{1:t-1}, \mathbf{z}_{1:t-1}\}$ denote the set of past observed and latent vectors at time $t$. Using the chain rule, the most general DVAE generative distribution can be written as the following causal generative process:

$$p_\theta(\mathbf{s}_{1:T}, \mathbf{z}_{1:T}) = \prod_{t=1}^T p_{\theta_\mathbf{s}}(\mathbf{s}_t|\mathbf{p}_t, \mathbf{z}_t)p_{\theta_\mathbf{z}}(\mathbf{z}_t|\mathbf{p}_t), \tag{4}$$

where $p_{\theta_\mathbf{s}}(\mathbf{s}_t|\mathbf{p}_t, \mathbf{z}_t)$ and $p_{\theta_\mathbf{z}}(\mathbf{z}_t|\mathbf{p}_t)$ are arbitrary generative distributions, which parameters are provided sequentially by RNNs taking the respective conditioning variables as inputs. A common choice is to use Gaussian distributions with diagonal covariance matrices:

$$p_{\theta_\mathbf{s}}(\mathbf{s}_t|\mathbf{p}_t, \mathbf{z}_t) = \mathcal{N}\big(\mathbf{s}_t; \boldsymbol{\mu}_{\theta_\mathbf{s}}(\mathbf{p}_t, \mathbf{z}_t), \text{diag}(\boldsymbol{v}_{\theta_\mathbf{s}}(\mathbf{p}_t, \mathbf{z}_t))\big), \tag{5}$$

$$p_{\theta_\mathbf{z}}(\mathbf{z}_t|\mathbf{p}_t) = \mathcal{N}\big(\mathbf{z}_t; \boldsymbol{\mu}_{\theta_\mathbf{z}}(\mathbf{p}_t), \text{diag}(\boldsymbol{v}_{\theta_\mathbf{z}}(\mathbf{p}_t))\big). \tag{6}$$

It can be noted that the distribution of $\mathbf{z}_t$ is more complex than the standard Gaussian used in the vanilla VAE. Also, the different models belonging to the DVAE class differ in the possible conditional independence assumptions that can be made in (4).

Similarly to the VAE, the exact posterior distribution $p_\theta(\mathbf{z}_{1:T}|\mathbf{s}_{1:T})$ corresponding to the DVAE generative model is not analytically tractable. Again, an inference model $q_{\phi_\mathbf{z}}(\mathbf{z}_{1:T}|\mathbf{s}_{1:T})$ is defined to approximate the exact posterior distribution. This inference model factorises as:

$$q_{\phi_\mathbf{z}}(\mathbf{z}_{1:T}|\mathbf{s}_{1:T}) = \prod_{t=1}^T q_{\phi_\mathbf{z}}(\mathbf{z}_t|\mathbf{q}_t), \tag{7}$$

where $\mathbf{q}_t = \{\mathbf{z}_{1:t-1}, \mathbf{s}_{1:T}\}$ denotes the set of past latent variables and all observations. Again, the Gaussian distribution with diagonal covariance matrix is generally used:

$$q_{\phi_\mathbf{z}}(\mathbf{z}_t|\mathbf{q}_t) = \mathcal{N}\big(\mathbf{z}_t; \boldsymbol{\mu}_{\phi_\mathbf{z}}(\mathbf{q}_t), \text{diag}(\boldsymbol{v}_{\phi_\mathbf{z}}(\mathbf{q}_t))\big), \tag{8}$$

where the mean and variance vectors are provided by an RNN (the encoder network) taking $\mathbf{q}_t$ as input and parameterized by $\phi$. With the most general generative model defined in (4), the conditional distribution in (7) cannot be simplified. However, if conditional independence assumptions are made in (4), the dependencies in $q_{\phi_\mathbf{z}}(\mathbf{z}_t|\mathbf{q}_t)$ can be simplified using the D-separation method (Bishop, 2006; Geiger et al., 1990), see (Girin et al., 2021) for details. In addition, we can force the inference model to be causal by replacing $\mathbf{s}_{1:T}$ with

Table 1: Summary of the variable notations.

| Variable notation | Definition |
|---|---|
| $T, t \in \{1, \ldots, T\}$ | Sequence length and frame index |
| $N, n \in \{1, \ldots, N\}$ | Total number of sources and source index |
| $K_t, k \in \{1, \ldots, K_t\}$ | Number of observations at $t$, and obs. index |
| $\mathbf{s}_{tn} \in \mathbb{R}^S$ | True position/content of source $n$ at time $t$ |
| $\mathbf{z}_{tn} \in \mathbb{R}^L$ | Latent variable of source $n$ at time $t$ |
| $\mathbf{o}_{tk} \in \mathbb{R}^O$ | Observation $k$ at time $t$ |
| $w_{tk} \in \{1, \ldots, N\}$ | Assignment variable of observation $k$ at time $t$ |
| $\mathbf{s}_{:,n} = \mathbf{s}_{1:T,n}$ | Source vector sequence for source $n$ |
| $\mathbf{s}_{t,:} = \mathbf{s}_{t,1:N}$ | Set of all source vectors at time $t$ |
| $\mathbf{s} = \mathbf{s}_{1:T,1:N}$ | Set of all source vectors |
| $\mathbf{z}_{:,n}, \mathbf{z}_{t,:}, \mathbf{z}$ | Analogous for the latent variable |
| $\mathbf{o} = \mathbf{o}_{1:T,1:K_t}$ | Set of all observations |
| $\mathbf{w} = \mathbf{w}_{1:T,1:K_t}$ | Set of all assignment variables |

$\mathbf{s}_{1:t}$ in $\mathbf{q}_t$. This is particularly suitable for on-line processing. In the rest of the paper, we will use the causal inference model, i.e. $\mathbf{q}_t = \{\mathbf{z}_{1:t-1}, \mathbf{s}_{1:t}\}$ in (7) and (8).

Similar to the VAE, and following the general VI principle, the DVAE model is also trained by maximizing the ELBO with a combination of SGD and sampling, the sampling being here recursive. The ELBO has here the following general form (Girin et al., 2021):

$$\mathcal{L}(\theta_\mathbf{s}, \theta_\mathbf{z}, \phi_\mathbf{z}; \mathbf{s}_{1:T}) = \mathbb{E}_{q_{\phi_\mathbf{z}}(\mathbf{z}_{1:T}|\mathbf{s}_{1:T})}\big[\log p_\theta(\mathbf{s}_{1:T}, \mathbf{z}_{1:T}) - \log q_{\phi_\mathbf{z}}(\mathbf{z}_{1:T}|\mathbf{s}_{1:T})\big]. \tag{9}$$

## 3 MixDVAE model

### 3.1 Problem formulation and notations

Let us consider a sequence containing $N$ sources or targets that we observe over time. Let $n \in \{1, ..., N\}$ denote the source index and let $\mathbf{s}_{tn} \in \mathbb{R}^S$ be here the true (unknown) $n$-th *source vector* at time frame $t$. At every time frame $t$, we gather $K_t$ observations, and this number can vary over time. We denote by $\mathbf{o}_{tk} \in \mathbb{R}^O$, $k \in \{1, ..., K_t\}$, the $k$-th observation at frame $t$. The problem tackled in this paper consists in estimating the sequence of hidden source vectors $\mathbf{s}_{1:T,n} = \{\mathbf{s}_{tn}\}_{t=1}^T$, for each source $n$, from the complete set of observations $\mathbf{o}_{1:T,1:K_t} = \{\mathbf{o}_{tk}\}_{t=1,k=1}^{T,K_t}$.

To solve this problem, we define two additional sets of latent variables. First, for each source $n$ at time frame $t$, we define a latent variable $\mathbf{z}_{tn} \in \mathbb{R}^L$ associated with $\mathbf{s}_{tn}$ through a DVAE model. This DVAE model, which might be identical for all sources or not, is used to model the dynamics of each individual source and is plugged into the proposed probabilistic MixDVAE model. Second, for each observation $\mathbf{o}_{tk}$, we define a discrete observation-to-source assignment variable $w_{tk}$ taking its value in $\{1, ..., N\}$. $w_{tk} = n$ means that observation $k$ at time frame $t$ is assigned to/was generated by source $n$. This results in per-source sequences of assigned observations.

Hereinafter, to simplify the notations, we use ":" as a shortcut subscript for the set of all values of the corresponding index. For example, $\mathbf{s}_{:,n} = \mathbf{s}_{1:T,n}$ is the complete trajectory of source $n$ and $\mathbf{s}_{t,:} = \mathbf{s}_{t,1:N}$ is the set of all source vectors at time frame $t$. All notations are summarized in Table 1.

### 3.2 General principle of the proposed model and solution

The general methodology of MixDVAE is to define a parametric joint distribution of all variables $p_\theta(\mathbf{o}, \mathbf{s}, \mathbf{z}, \mathbf{w})$, then estimate its parameters $\theta$ and (an approximation of) the corresponding posterior distribution $p_\theta(\mathbf{s}, \mathbf{z}, \mathbf{w}|\mathbf{o})$, from which we can deduce an estimate of $\mathbf{s}_{1:T,n}$ for each source $n$. The proposed MixDVAE

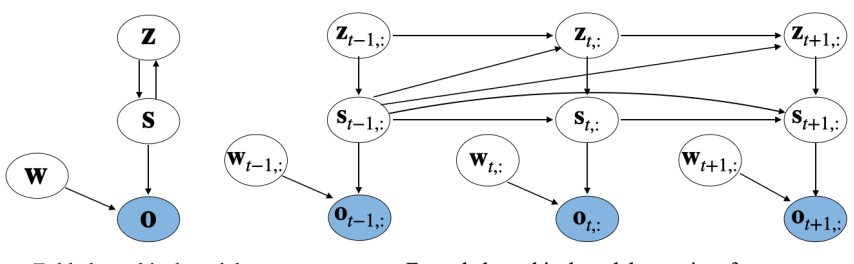

Figure 1: Graphical representation of the proposed MixDVAE model.

generative model $p_\theta(\mathbf{o}, \mathbf{s}, \mathbf{z}, \mathbf{w})$ is presented in Section 3.3. As briefly stated above, it integrates the DVAE generative model (4)–(6) for modeling the sources dynamics and does not use any human-annotated data for training. As is usually the case in (D)VAE-based generative models, both the exact posterior distribution $p_\theta(\mathbf{s}, \mathbf{z}, \mathbf{w}|\mathbf{o})$ and the marginalization of the joint distribution $p_\theta(\mathbf{o}, \mathbf{s}, \mathbf{z}, \mathbf{w})$ w.r.t. the latent variables are analytically intractable. Therefore we cannot directly use an exact EM algorithm and we resort to VI. We propose the following strategy, inspired by the structured mean-field method that we summarized in Section 2.1, with $\mathbf{h}$ being here equal to $\{\mathbf{s}, \mathbf{z}, \mathbf{w}\}$. In Section 3.4, we define an approximate posterior distribution $q_\phi(\mathbf{s}, \mathbf{z}, \mathbf{w}|\mathbf{o})$ that partially factorizes over $\{\mathbf{s}, \mathbf{z}, \mathbf{w}\}$. Just like the proposed MixDVAE generative model includes the DVAE generative model, the approximate posterior distribution includes the DVAE inference model as one of the factors. This factorization makes possible the derivation of a model solution in the form of a VEM algorithm, as detailed in Section 4.

### 3.3 Generative model

Let us now specify the joint distribution of observed and latent variables $p_\theta(\mathbf{o}, \mathbf{w}, \mathbf{s}, \mathbf{z})$. We assume that the observation variable $\mathbf{o}$ only depends on $\mathbf{w}$ and $\mathbf{s}$, while the assignment variable $\mathbf{w}$ is a priori independent of the other variables. The graphical representation of MixDVAE is shown in Figure 1. Applying the chain rule and these conditional dependency assumptions, the joint distribution can be factorised as follows:

$$p_\theta(\mathbf{o}, \mathbf{w}, \mathbf{s}, \mathbf{z}) = p_{\theta_\mathbf{o}}(\mathbf{o}|\mathbf{w}, \mathbf{s}) p_{\theta_\mathbf{w}}(\mathbf{w}) p_{\theta_{\mathbf{sz}}}(\mathbf{s}, \mathbf{z}). \tag{10}$$

**Observation model.** We assume that the observations are conditionally independent through time and independent of each other, that is to say, at any time frame $t$, the observation $\mathbf{o}_{tk}$ only depends on its corresponding assignment $w_{tk}$ and source vector at the same time frame. The observation model $p_{\theta_\mathbf{o}}(\mathbf{o}|\mathbf{w}, \mathbf{s})$ can thus be factorised as:[2]

$$p_{\theta_\mathbf{o}}(\mathbf{o}|\mathbf{w}, \mathbf{s}) = \prod_{t=1}^{T} \prod_{k=1}^{K_t} p_{\theta_\mathbf{o}}(\mathbf{o}_{tk}|w_{tk}, \mathbf{s}_{t,:}). \tag{11}$$

Given the value of the assignment variable, the distribution $p(\mathbf{o}_{tk}|w_{tk}, \mathbf{s}_{t,:})$ is modeled by a Gaussian distribution:

$$p_{\theta_\mathbf{o}}(\mathbf{o}_{tk}|w_{tk} = n, \mathbf{s}_{tn}) = \mathcal{N}(\mathbf{o}_{tk}; \mathbf{s}_{tn}, \boldsymbol{\Phi}_{tk}). \tag{12}$$

This equation models only the observation noise via the covariance $\boldsymbol{\Phi}_{tk} \in \mathbb{R}^{O \times O}$ and thus assumes that the assigned observation lies close to the true source vector.[3]

---

[2]In this equation, we use $\mathbf{s}_{t,:}$ and not $\mathbf{s}_{tn}$, since the value of $w_{tk}$ is not specified.

[3]For simplicity of presentation, we state the case in which the observation and source vector dimensions are the same, i.e. $O = S$. In a more general case where $O \neq S$, we can consider the use of a projection matrix $\mathbf{P}_k \in \mathbb{R}^{O \times S}$ and define $p_{\theta_\mathbf{o}}(\mathbf{o}_{tk}|w_{tk} = n, \mathbf{s}_{tn}) = \mathcal{N}(\mathbf{o}_{tk}; \mathbf{P}_k \mathbf{s}_{tn}, \boldsymbol{\Phi}_{tk})$. Again, for simplicity, we consider $\mathbf{P}_k = \mathbf{I}$ in the rest of the paper. All derivations and results are generalizable to $\mathbf{P}_k \neq \mathbf{I}$.

**Assignment model.** Similarly, we assume that, a priori, the assignment variables are independent across time and observations:

$$p_{\theta_{\mathbf{w}}}(\mathbf{w}) = \prod_{t=1}^{T} \prod_{k=1}^{K_t} p_{\theta_{\mathbf{w}}}(w_{tk}). \tag{13}$$

For each time frame $t$ and each observation $k$, the assignment variable $w_{tk}$ is assumed to follow a uniform prior distribution:

$$p_{\theta_{\mathbf{w}}}(w_{tk}) = \frac{1}{N}. \tag{14}$$

**Dynamical model.** Finally, $p_{\theta_{\mathbf{sz}}}(\mathbf{s}, \mathbf{z})$ is modeled with a DVAE. The different sources are assumed to be independent of each other. This implies that in the present work we do not consider possible interactions among sources. More complex dynamical models including source interaction are beyond the scope of this paper. With this assumption, the joint distribution of all source vectors and corresponding latent variable $p_{\theta_{\mathbf{sz}}}(\mathbf{s}, \mathbf{z})$ can be factorized across sources as:

$$p_{\theta_{\mathbf{sz}}}(\mathbf{s}, \mathbf{z}) = \prod_{n=1}^{N} p_{\theta_{\mathbf{sz}}}(\mathbf{s}_{:,n}, \mathbf{z}_{:,n}), \tag{15}$$

where $p_{\theta_{\mathbf{sz}}}(\mathbf{s}_{:,n}, \mathbf{z}_{:,n})$ is the DVAE model defined in (4)–(6) and applied to $\mathbf{s}_{:,n}$ and $\mathbf{z}_{:,n}$ (defining $\mathbf{p}_{t,n} = \{\mathbf{s}_{1:t-1,n}, \mathbf{z}_{1:t-1,n}\}$).[4] As mentioned before, the DVAE model can be either the same architecture for all sources, pre-trained on a unique single-source dataset, or the same architecture but pre-trained on different single-source datasets for different sources, or completely different architectures for each source.

Overall, the parameters in the generative model to be estimated are $\theta = \{\theta_{\mathbf{o}} = \{\mathbf{\Phi}_{tk}\}_{t,k=1}^{T,K_t}, \theta_{\mathbf{s}}, \theta_{\mathbf{z}}\}$ (note that $\theta_{\mathbf{w}} = \emptyset$).

### 3.4 Inference model

The exact posterior distribution corresponding to the MixDVAE generative model described in Section 3.3 is neither analytically nor computationally tractable. Therefore, we propose the following factorized approximation that leads to a computationally tractable inference model:

$$q_\phi(\mathbf{s}, \mathbf{z}, \mathbf{w}|\mathbf{o}) = q_{\phi_{\mathbf{w}}}(\mathbf{w}|\mathbf{o}) q_{\phi_{\mathbf{z}}}(\mathbf{z}|\mathbf{s}) q_{\phi_{\mathbf{s}}}(\mathbf{s}|\mathbf{o}), \tag{16}$$

where $q_{\phi_{\mathbf{z}}}(\mathbf{z}|\mathbf{s})$ corresponds to the inference model of the DVAE and the optimal distributions $q_{\phi_{\mathbf{s}}}(\mathbf{s}|\mathbf{o})$ and $q_{\phi_{\mathbf{w}}}(\mathbf{w}|\mathbf{o})$ are derived below in the E-steps of the MixDVAE algorithm. The factorization (16) is inspired by the structured mean-field method (Parisi & Shankar, 1988), since we break the posterior dependency between $\mathbf{w}$ and $\{\mathbf{s}, \mathbf{z}\}$. However, we keep the dependency between $\mathbf{s}$ and $\mathbf{z}$ at inference time since it is the essence of the DVAE. In addition, we assume that the posterior distribution of the DVAE latent variable is independent for each source, so that we have:

$$q_{\phi_{\mathbf{z}}}(\mathbf{z}|\mathbf{s}) = \prod_{n=1}^{N} q_{\phi_{\mathbf{z}}}(\mathbf{z}_{:,n}|\mathbf{s}_{:,n}), \tag{17}$$

where $q_{\phi_{\mathbf{z}}}(\mathbf{z}_{:,n}|\mathbf{s}_{:,n})$ is given by (7) and (8) applied to $\{\mathbf{z}_{:,n}, \mathbf{s}_{:,n}\}$. This is coherent with the generative model, where we assumed that the dynamics of the various sources are independent of each other.

## 4 MixDVAE solution: A variational expectation-maximization algorithm

Let us now present the proposed algorithm for jointly deriving the terms of the inference model (other than the DVAE terms) and estimating the parameters of the complete MixDVAE model, based on the maximization of the corresponding ELBO. The inference is done directly on each multi-source test sequence

---

[4]Here we denote the DVAE parameters by $\theta_{\mathbf{sz}}$ instead of $\theta$, to differentiate the DVAE parameters from the other parameters.

to process and does not require previous supervised training with a labeled multi-source dataset. It only requires to pre-train the DVAE model on synthetic or natural single-source sequences.

As discussed in Section 2, in many generative models, the optimization of the ELBO is done either following the structured mean-field method (2) or using amortized inference as in (D)VAEs. In our case, we cannot directly use the generic structured mean-field inference procedure, since the proposed approximation (16) does not factorize completely in a set of disjoint latent variables (e.g., $q_{\phi_{\mathbf{z}}}(\mathbf{z}|\mathbf{s})$ is conditioned on $\mathbf{s}$). Alternatively, one could resort to purely amortized inference and conceive a deep encoder that approximates the distributions in (16), leading to a looser approximation bound. We propose a strategy that is a middle ground between these two worlds. We use the structured mean-field principles that provide a tighter bound since they do not impose a distribution family for $q_{\phi_{\mathbf{w}}}$ and $q_{\phi_{\mathbf{s}}}$, and we use the philosophy of amortized inference for $q_{\phi_{\mathbf{z}}}$ so as to exploit the pre-trained DVAE encoder.

To do so, we have to go back to the fundamentals of VI and iteratively maximize the MixDVAE model ELBO defined by:

$$\mathcal{L}(\theta, \phi; \mathbf{o}) = \mathbb{E}_{q_\phi(\mathbf{s},\mathbf{z},\mathbf{w}|\mathbf{o})}[\log p_\theta(\mathbf{o}, \mathbf{s}, \mathbf{z}, \mathbf{w}) - \log q_\phi(\mathbf{s}, \mathbf{z}, \mathbf{w}|\mathbf{o})]. \tag{18}$$

By injecting (10) and (16) into (18), we can develop $\mathcal{L}(\theta, \phi; \mathbf{o})$ as follows:

$$\begin{aligned}
\mathcal{L}(\theta, \phi; \mathbf{o}) = {} & \mathbb{E}_{q_{\phi_{\mathbf{w}}}(\mathbf{w}|\mathbf{o})q_{\phi_{\mathbf{s}}}(\mathbf{s}|\mathbf{o})}\big[\log p_{\theta_{\mathbf{o}}}(\mathbf{o}|\mathbf{w},\mathbf{s})\big] + \mathbb{E}_{q_{\phi_{\mathbf{w}}}(\mathbf{w}|\mathbf{o})}\big[\log p_{\theta_{\mathbf{w}}}(\mathbf{w}) - \log q_{\phi_{\mathbf{w}}}(\mathbf{w}|\mathbf{o})\big] \\
& + \mathbb{E}_{q_{\phi_{\mathbf{s}}}(\mathbf{s}|\mathbf{o})}\Big[\mathbb{E}_{q_{\phi_{\mathbf{z}}}(\mathbf{z}|\mathbf{s})}\big[\log p_{\theta_{\mathbf{sz}}}(\mathbf{s},\mathbf{z}) - \log q_{\phi_{\mathbf{z}}}(\mathbf{z}|\mathbf{s})\big]\Big] - \mathbb{E}_{q_{\phi_{\mathbf{s}}}(\mathbf{s}|\mathbf{o})}\big[\log q_{\phi_{\mathbf{s}}}(\mathbf{s}|\mathbf{o})\big]. \tag{19}
\end{aligned}$$

The ELBO maximization is done by alternatively and iteratively maximizing the different terms corresponding to the various posterior and generative distributions. In our case, we obtain a series of variational E and M steps. While the E steps associated to $q_{\phi_{\mathbf{w}}}$ and $q_{\phi_{\mathbf{s}}}$ follow the structured mean-field principle, the E step associated to $q_{\phi_{\mathbf{z}}}$ is based on the principle of amortized inference commonly used in (D)VAEs.

### 4.1 E-S step

We first consider the computation of the optimal posterior distribution $q_{\phi_{\mathbf{s}}}(\mathbf{s}|\mathbf{o})$. To this aim, we first select the terms in (19) that depend on $\mathbf{s}$, the other terms being here considered as a constant:

$$\mathcal{L}_{\mathbf{s}}(\theta, \phi; \mathbf{o}) = \mathbb{E}_{q_{\phi_{\mathbf{s}}}(\mathbf{s}|\mathbf{o})}\Big[\mathbb{E}_{q_{\phi_{\mathbf{w}}}(\mathbf{w}|\mathbf{o})}\big[\log p_{\theta_{\mathbf{o}}}(\mathbf{o}|\mathbf{w},\mathbf{s})\big] + \mathbb{E}_{q_{\phi_{\mathbf{z}}}(\mathbf{z}|\mathbf{s})}\big[\log p_{\theta_{\mathbf{sz}}}(\mathbf{s},\mathbf{z}) - \log q_{\phi_{\mathbf{z}}}(\mathbf{z}|\mathbf{s})\big] - \log q_{\phi_{\mathbf{s}}}(\mathbf{s}|\mathbf{o})\Big]. \tag{20}$$

Let us define:

$$\tilde{p}(\mathbf{s}|\mathbf{o}) = \mathcal{C}' \exp\Big(\mathbb{E}_{q_{\phi_{\mathbf{w}}}(\mathbf{w}|\mathbf{o})}\big[\log p_{\theta_{\mathbf{o}}}(\mathbf{o}|\mathbf{w},\mathbf{s})\big] + \mathbb{E}_{q_{\phi_{\mathbf{z}}}(\mathbf{z}|\mathbf{s})}\big[\log p_{\theta_{\mathbf{sz}}}(\mathbf{s},\mathbf{z}) - \log q_{\phi_{\mathbf{z}}}(\mathbf{z}|\mathbf{s})\big]\Big), \tag{21}$$

where $\mathcal{C}' > 0$ is the appropriate normalisation constant. (20) rewrites:

$$\mathcal{L}_{\mathbf{s}}(\theta, \phi; \mathbf{o}) = -D_{\mathrm{KL}}\big(q_{\phi_{\mathbf{s}}}(\mathbf{s}|\mathbf{o}) \parallel \tilde{p}(\mathbf{s}|\mathbf{o})\big) + \mathcal{C}, \tag{22}$$

where $D_{\mathrm{KL}}(\cdot|\cdot)$ denotes the Kullback-Leibler divergence (KLD). Therefore, the optimal distribution is the one minimising the above KLD:

$$q_{\phi_{\mathbf{s}}}(\mathbf{s}|\mathbf{o}) = \tilde{p}(\mathbf{s}|\mathbf{o}) \propto \exp\Big(\mathbb{E}_{q_{\phi_{\mathbf{w}}}(\mathbf{w}|\mathbf{o})}\big[\log p_{\theta_{\mathbf{o}}}(\mathbf{o}|\mathbf{w},\mathbf{s})\big] + \mathbb{E}_{q_{\phi_{\mathbf{z}}}(\mathbf{z}|\mathbf{s})}\big[\log p_{\theta_{\mathbf{sz}}}(\mathbf{s},\mathbf{z}) - \log q_{\phi_{\mathbf{z}}}(\mathbf{z}|\mathbf{s})\big]\Big). \tag{23}$$

Since for any pair $(t, k)$, the assignment variable $w_{tk}$ follows a discrete posterior distribution, we can denote the corresponding probability values by $\eta_{tkn} = q_{\phi_{\mathbf{w}}}(w_{tk} = n|\mathbf{o}_{tk})$. These values will be computed in the E-W step below. The expectation with respect to $q_{\phi_{\mathbf{w}}}(\mathbf{w}|\mathbf{o})$ in (23) can be calculated using these values. However, the expectation with respect to $q_{\phi_{\mathbf{z}}}(\mathbf{z}|\mathbf{s})$ cannot be calculated in closed form. As usually done in the (D)VAE methodology, it is thus replaced by a Monte Carlo estimate using sampled sequences drawn from the DVAE inference model at the previous iteration (see Section 4.5). Replacing the distributions in

(23) with (11), (15), and (17), and calculating the expectations with respect to $q_{\phi_{\mathbf{w}}}(\mathbf{w}|\mathbf{o})$ and $q_{\phi_{\mathbf{z}}}(\mathbf{z}|\mathbf{s})$, we find that $q_{\phi_{\mathbf{s}}}(\mathbf{s}|\mathbf{o})$ factorizes with respect to $n$ as follows:

$$q_{\phi_{\mathbf{s}}}(\mathbf{s}|\mathbf{o}) = \prod_{n=1}^{N} q_{\phi_{\mathbf{s}}}(\mathbf{s}_{:,n}|\mathbf{o}). \tag{24}$$

Each of these factors corresponds to the posterior distribution of the $n$-th source vector. Given (23) and the DVAE generative and inference models, we see that at a given time $t$, the distribution over $\mathbf{s}_{tn}$ has non-linear dependencies w.r.t. the previous and current DVAE latent variables $\mathbf{z}_{1:t,n}$ and the previous source vectors $\mathbf{s}_{1:t-1,n}$. These non-linear dependencies impede to obtain an efficient closed-form solution. We resort to point sample estimates obtained using samples of $\mathbf{z}_{1:t,n}$ and of $\mathbf{s}_{1:t-1,n}$, at the current iteration, denoted $\mathbf{z}_{1:t,n}^{(i)}$ and $\mathbf{s}_{1:t-1,n}^{(i)}$. Using these samples, the posterior distribution is approximated with (details can be found in Appendix A.1):

$$q_{\phi_{\mathbf{s}}}(\mathbf{s}_{:,n}|\mathbf{o}) \approx \prod_{t=1}^{T} q_{\phi_{\mathbf{s}}}(\mathbf{s}_{tn}|\mathbf{s}_{1:t-1,n}^{(i)}, \mathbf{z}_{1:t,n}^{(i)}, \mathbf{o}), \tag{25}$$

where each term of the product is shown to be a Gaussian:

$$q_{\phi_{\mathbf{s}}}(\mathbf{s}_{tn}|\mathbf{s}_{1:t-1,n}^{(i)}, \mathbf{z}_{1:t,n}^{(i)}, \mathbf{o}) = \mathcal{N}(\mathbf{s}_{tn}; \mathbf{m}_{tn}, \mathbf{V}_{tn}), \tag{26}$$

with covariance matrix and mean vector given by:

$$\mathbf{V}_{tn} = \left( \sum_{k=1}^{K_t} \eta_{tkn} \mathbf{\Phi}_{tk}^{-1} + \mathrm{diag}(\boldsymbol{v}_{\theta_{\mathbf{s}},tn}^{(i)})^{-1} \right)^{-1}, \tag{27}$$

$$\mathbf{m}_{tn} = \mathbf{V}_{tn} \left( \sum_{k=1}^{K_t} \eta_{tkn} \mathbf{\Phi}_{tk}^{-1} \mathbf{o}_{tk} + \mathrm{diag}(\boldsymbol{v}_{\theta_{\mathbf{s}},tn}^{(i)})^{-1} \boldsymbol{\mu}_{\theta_{\mathbf{s}},tn}^{(i)} \right), \tag{28}$$

where $\boldsymbol{v}_{\theta_{\mathbf{s}},tn}^{(i)}$ and $\boldsymbol{\mu}_{\theta_{\mathbf{s}},tn}^{(i)}$ are simplified notations for $\boldsymbol{v}_{\theta_{\mathbf{s}}}(\mathbf{s}_{1:t-1,n}^{(i)}, \mathbf{z}_{1:t,n}^{(i)})$ and $\boldsymbol{\mu}_{\theta_{\mathbf{s}}}(\mathbf{s}_{1:t-1,n}^{(i)}, \mathbf{z}_{1:t,n}^{(i)})$, respectively denoting the variance and mean vector provided by the DVAE decoder network for source $n$ at time frame $t$. As we have to sample both $\mathbf{s}_{:,n}$ and $\mathbf{z}_{:,n}$, we need to pay attention to the sampling order. This will be discussed in detail in Section 4.5. Importantly, in practice, $\mathbf{m}_{tn}$ is used as the estimate of $\mathbf{s}_{tn}$.

Eq. (28) shows that the estimated $n$-th source vector is obtained by combining the observations $\mathbf{o}_{tk}$ and the mean source vector $\boldsymbol{\mu}_{\theta_{\mathbf{s}},tn}^{(i)}$ predicted by the DVAE generative model. The balance between these two terms depends on the assignment variable $\eta_{tkn}$, the observation model covariance matrix $\mathbf{\Phi}_{tk}$ and the source variance predicted by the DVAE generative model $\boldsymbol{v}_{\theta_{\mathbf{s}},tn}^{(i)}$. Ideally, the model should be able to appropriately balance these two terms so as to optimally exploit both the observations and the DVAE predictions.

## 4.2   E-Z step

In the E-Z step, we consider the DVAE inference model $q_{\phi_{\mathbf{z}}}(\mathbf{z}|\mathbf{s})$, defined by (17), (7) and (8). In (19), the corresponding term is the third one, which we denote by $\mathcal{L}_{\mathbf{z}}(\theta_{\mathbf{s}}, \theta_{\mathbf{z}}, \phi_{\mathbf{z}}; \mathbf{o})$ and which factorizes across sources as follows (see Appendix A.2):

$$\mathcal{L}_{\mathbf{z}}(\theta_{\mathbf{s}}, \theta_{\mathbf{z}}, \phi_{\mathbf{z}}; \mathbf{o}) = \sum_{n=1}^{N} \mathcal{L}_{\mathbf{z},n}(\theta_{\mathbf{s}}, \theta_{\mathbf{z}}, \phi_{\mathbf{z}}; \mathbf{o}), \tag{29}$$

with

$$\mathcal{L}_{\mathbf{z},n}(\theta_{\mathbf{s}}, \theta_{\mathbf{z}}, \phi_{\mathbf{z}}; \mathbf{o}) = \mathbb{E}_{q_{\phi_{\mathbf{s}}}(\mathbf{s}_{:,n}|\mathbf{o})} \Big[ \mathbb{E}_{q_{\phi_{\mathbf{z}}}(\mathbf{z}_{:,n}|\mathbf{s}_{:,n})} \big[ \log p_{\theta_{\mathbf{sz}}}(\mathbf{s}_{:,n}, \mathbf{z}_{:,n}) \big] - \mathbb{E}_{q_{\phi_{\mathbf{z}}}(\mathbf{z}_{:,n}|\mathbf{s}_{:,n})} \big[ \log q_{\phi_{\mathbf{z}}}(\mathbf{z}_{:,n}|\mathbf{s}_{:,n}) \big] \Big]. \tag{30}$$

Inside the expectation $\mathbb{E}_{q_{\phi_{\mathbf{s}}}(\mathbf{s}_{:,n}|\mathbf{o})}[\cdot]$, we recognize the DVAE ELBO defined in (9) and applied to source $n$. This suggests the following strategy. Previously to and independently of the MixDVAE algorithm, we

pre-train the DVAE model on a dataset of synthetic or natural unlabeled single-source sequences (this is detailed in Sections 5.1 and 6.1). This is done only once, and the resulting DVAE is then plugged into the MixDVAE algorithm to process multi-source sequences. This provides the E-Z step with very good initial values of the DVAE parameters $\theta_{\mathbf{s}}$, $\theta_{\mathbf{z}}$ and $\phi_{\mathbf{z}}$. As for the following of the E-Z step, the expectation over $q_{\phi_{\mathbf{s}}}(\mathbf{s}_{:,n}|\mathbf{o})$ in (30) is not analytically tractable. A Monte Carlo estimate is thus used instead, using samples of both $\mathbf{z}$ and $\mathbf{s}$, similarly to what was done in the E-S step. Finally, SGD is used to maximize (the Monte Carlo estimate of) $\mathcal{L}_{\mathbf{z}}(\theta_{\mathbf{s}}, \theta_{\mathbf{z}}, \phi_{\mathbf{z}}; \mathbf{o})$, jointly updating $\theta_{\mathbf{s}}$, $\theta_{\mathbf{z}}$ and $\phi_{\mathbf{z}}$; that is, we fine-tune the DVAE model within the MixDVAE algorithm, using the observations $\mathbf{o}$. Note that in our experiments, we also consider the case where we neutralize the fine-tuning, i.e. we remove the E-Z step and use the DVAE model as provided by the pre-training phase.

### 4.3 E-W step

Thanks to the separation of $\mathbf{w}$ from the two other latent variables in (16), the posterior distribution $q_{\phi_{\mathbf{w}}}(\mathbf{w}|\mathbf{o})$ can be calculated in closed form by directly applying the optimal structured mean-field update equation (2) to our model. It can be shown that this is equivalent to maximizing (19) w.r.t. $q_{\phi_{\mathbf{w}}}(\mathbf{w}|\mathbf{o})$. We obtain (see Appendix A for details):

$$q_{\phi_{\mathbf{w}}}(\mathbf{w}|\mathbf{o}) \propto \prod_{t=1}^{T} \prod_{k=1}^{K_t} q_{\phi_{\mathbf{w}}}(w_{tk}|\mathbf{o}), \tag{31}$$

with

$$q_{\phi_{\mathbf{w}}}(w_{tk} = n|\mathbf{o}) = \eta_{tkn} = \frac{\beta_{tkn}}{\sum_{i=1}^{N} \beta_{tki}}, \tag{32}$$

where

$$\beta_{tkn} = \mathcal{N}(\mathbf{o}_{tk}; \mathbf{m}_{tn}, \boldsymbol{\Phi}_{tk}) \exp\Big( -\frac{1}{2}\text{Tr}\big(\boldsymbol{\Phi}_{tk}^{-1}\mathbf{V}_{tn}\big)\Big). \tag{33}$$

The parameters $\mathbf{m}_{tn}$ and $\mathbf{V}_{tn}$ in the above equation have been defined in (28) and (27), respectively.

### 4.4 M step

As discussed in Section 2, the maximization step generally consists in estimating the parameters $\theta$ of the generative model by maximizing the ELBO over $\theta$. We recall that $\theta = \{\theta_{\mathbf{o}} = \{\boldsymbol{\Phi}_{tk}\}_{t,k=1}^{T,K_t}, \theta_{\mathbf{s}}, \theta_{\mathbf{z}}\}$. In this work, the parameters of the DVAE decoder $\theta_{\mathbf{s}}$ and $\theta_{\mathbf{z}}$ are first estimated (offline) during the pre-training of the DVAE and then fine-tuned in the E-Z step in an amortized way, all this jointly with the parameters of the encoder $\phi_{\mathbf{z}}$. Therefore, in the M-step, we only need to estimate the observation model covariance matrices $\theta_{\mathbf{o}} = \{\boldsymbol{\Phi}_{tk}\}_{t,k=1}^{T,K_t}$. In (19), only the first term depends on $\theta_{\mathbf{o}}$. Setting its derivative with respect to $\boldsymbol{\Phi}_{tk}$ to zero, we obtain (see Appendix A for details):

$$\boldsymbol{\Phi}_{tk} = \sum_{n=1}^{N} \eta_{tkn}\Big((\mathbf{o}_{tk} - \mathbf{m}_{tn})(\mathbf{o}_{tk} - \mathbf{m}_{tn})^T + \mathbf{V}_{tn}\Big). \tag{34}$$

In practice, it is difficult to obtain a reliable estimation using only a single observation. We address this issue in Sections 5.2 and 6.2.

### 4.5 MixDVAE complete algorithm

As already mentioned in Section 4.1, we must pay attention to the sampling order of $\mathbf{s}$ and $\mathbf{z}$ when running the iterations of the E-S and E-Z steps. As indicated in the pseudo-code of Algorithm 1, in practice, the E-S and E-Z steps are processed jointly. We start with the initial source vectors sequence $\mathbf{s}_{1:T,1:N}^{(0)}$ and initial mean source vectors sequence $\mathbf{m}_{1:T,1:N}^{(0)}$. At any iteration $i$ of the E-Z and E-S steps, for each source $n$ and each time frame $t$, we sample in the following order:

---

**Algorithm 1** MixDVAE algorithm

---

**Input:**

    Observation vectors $\mathbf{o} = \mathbf{o}_{1:T,1:K_t}$;

**Output:**

    Parameters of $q_{\phi_{\mathbf{s}}}(\mathbf{s})$ : $\{\mathbf{m}_{tn}^{(I)}, \mathbf{V}_{tn}^{(I)}\}_{t,n=1}^{T,N}$ (the estimated $n$-th source vector at time frame $t$ is $\mathbf{m}_{tn}$);

    Values of the assignment variable $\{\eta_{tkn}^{(I)}\}_{t,n,k=1}^{T,N,K_t}$;

 1: **Initialization**

 2:    See Sections 5.2 and 6.2

 3: **for** $i \leftarrow 1$ to $I$ **do**

 4:    **E-W Step**

 5:    **for** $n \leftarrow 1$ to $N$ **do**

 6:        **for** $t \leftarrow 1$ to $T$ **do**

 7:            **for** $k \leftarrow 1$ to $K_t$ **do**

 8:                Compute $\eta_{tkn}^{(i)}$ using (32) and (33);

 9:            **end for**

10:        **end for**

11:    **end for**

12:    **E-Z and E-S Step**

13:    **for** $n \leftarrow 1$ to $N$ **do**

14:        **for** $t \leftarrow 1$ to $T$ **do**

15:            *Encoder*;

16:            Compute $\boldsymbol{\mu}_{\phi_{\mathbf{z}},tn}^{(i)}$, $\boldsymbol{v}_{\phi_{\mathbf{z}},tn}^{(i)}$ with input $\mathbf{s}_{1:T,n}^{(i-1)}$ and $\mathbf{z}_{1:t-1,n}^{(i)}$;

17:            Sample $\mathbf{z}_{tn}^{(i)}$ from $q_{\phi_{\mathbf{z}}}(\mathbf{z}_{tn}|\mathbf{s}_{1:T,n}^{(i-1)}, \mathbf{z}_{1:t-1,n}^{(i)}) = \mathcal{N}(\mathbf{z}_{tn}; \boldsymbol{\mu}_{\phi_{\mathbf{z}},tn}^{(i)}, \mathrm{diag}(\boldsymbol{v}_{\phi_{\mathbf{z}},tn}^{(i)}))$;

18:            *Decoder*;

19:            Compute $\boldsymbol{\mu}_{\theta_{\mathbf{z}},tn}^{(i)}$ and $\boldsymbol{v}_{\theta_{\mathbf{z}},tn}^{(i)}$ with input $\mathbf{s}_{1:t-1,n}^{(i)}$ and $\mathbf{z}_{1:t-1,n}^{(i)}$;

20:            Compute $\boldsymbol{\mu}_{\theta_{\mathbf{s}},tn}^{(i)}$ and $\boldsymbol{v}_{\theta_{\mathbf{s}},tn}^{(i)}$ with input $\mathbf{s}_{1:t-1,n}^{(i)}$ and $\mathbf{z}_{1:t,n}^{(i)}$;

21:            *E-S update*;

22:            Compute $\mathbf{m}_{tn}^{(i)}$, $\mathbf{V}_{tn}^{(i)}$ using (28) and (27);

23:            Sample $\mathbf{s}_{tn}^{(i)}$ from $\mathcal{N}(\mathbf{s}_{tn}; \mathbf{m}_{tn}^{(i)}, \mathbf{V}_{tn}^{(i)})$;

24:        **end for**

25:        *E-Z update*;

26:        Compute $\widehat{\mathcal{L}}_n(\theta_{\mathbf{s}}, \theta_{\mathbf{z}}, \phi_{\mathbf{z}}; \mathbf{o})$ using (35);

27:    **end for**

28:    Compute $\widehat{\mathcal{L}}(\theta_{\mathbf{s}}, \theta_{\mathbf{z}}, \phi_{\mathbf{z}}; \mathbf{o}) = \sum_{n=1}^{N} \widehat{\mathcal{L}}_n(\theta_{\mathbf{s}}, \theta_{\mathbf{z}}, \phi_{\mathbf{z}}; \mathbf{o})$;

29:    Fine-tune the DVAE parameters $\{\theta_{\mathbf{s}}, \theta_{\mathbf{z}}, \phi_{\mathbf{z}}\}$ by applying SGD on $\widehat{\mathcal{L}}(\theta_{\mathbf{s}}, \theta_{\mathbf{z}}, \phi_{\mathbf{z}}; \mathbf{o})$;

30:    **M Step**

31:    Compute $\boldsymbol{\Phi}_{tk}^{(i)}$ using (34) or following Sections 5.2 and 6.2;

32: **end for**

---

1. Compute the parameters $\boldsymbol{\mu}_{\phi_{\mathbf{z}},tn}^{(i)}$ and $\boldsymbol{v}_{\phi_{\mathbf{z}},tn}^{(i)}$ [5] of the posterior distribution of $\mathbf{z}_t$ using the DVAE encoder network with inputs $\mathbf{s}_{1:T,n}^{(i-1)}$ sampled at the previous iteration and $\mathbf{z}_{1:t-1,n}^{(i)}$ sampled at the current iteration. Then, sample $\mathbf{z}_{tn}^{(i)}$ from $q_{\phi_{\mathbf{z}}}(\mathbf{z}_{tn}|\mathbf{s}_{1:T,n}^{(i-1)}, \mathbf{z}_{1:t-1,n}^{(i)})$.

2. Compute the parameters $\boldsymbol{\mu}_{\theta_{\mathbf{z}},tn}^{(i)}$ and $\boldsymbol{v}_{\theta_{\mathbf{z}},tn}^{(i)}$ [6] of the generative distribution of $\mathbf{z}_t$ using the corresponding DVAE decoder network with inputs $\mathbf{s}_{1:t-1,n}^{(i)}$ and $\mathbf{z}_{1:t-1,n}^{(i)}$, both sampled at the current iteration.

3. Compute the parameters $\boldsymbol{\mu}_{\theta_{\mathbf{s}},tn}^{(i)}$ and $\boldsymbol{v}_{\theta_{\mathbf{s}},tn}^{(i)}$ of the generative distribution of $\mathbf{s}_t$ using the corresponding DVAE decoder network with inputs $\mathbf{s}_{1:t-1,n}^{(i)}$ and $\mathbf{z}_{1:t,n}^{(i)}$, both sampled at the current iteration. Compute

---

[5]$\boldsymbol{\mu}_{\phi_{\mathbf{z}},tn}^{(i)}$ and $\boldsymbol{v}_{\phi_{\mathbf{z}},tn}^{(i)}$ are shortcuts for $\boldsymbol{\mu}_{\phi_{\mathbf{z}}}(\mathbf{s}_{1:T,n}^{(i-1)}, \mathbf{z}_{1:t-1,n}^{(i)})$ and $\boldsymbol{v}_{\phi_{\mathbf{z}}}(\mathbf{s}_{1:T,n}^{(i-1)}, \mathbf{z}_{1:t-1,n}^{(i)})$ respectively.
[6]Analogous definitions hold.

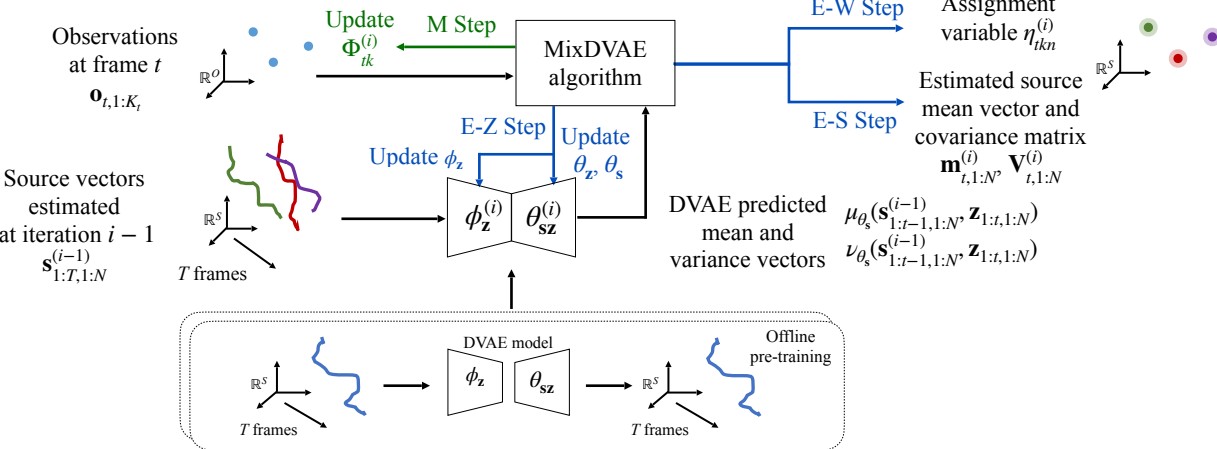

Figure 2: Overview of the proposed MixDVAE algorithm at a given time frame $t$. The DVAE model is pretrained offline using a (synthetic or natural) single-source dataset. It takes as input the sequence of source vectors, encodes them into a sequence of latent vectors, which are then decoded into the reconstructed sequence of source vectors. For a given time frame $t$, the MixDVAE algorithm takes as input the observations at time $t$ as well as the mean and variance vectors estimated by the DVAE model. By iterating the E-S, E-Z, E-W and M steps, we obtain estimates of the assignment variable and of each source vector.

the parameters $\mathbf{m}_{tn}^{(i)}$ and $\mathbf{V}_{tn}^{(i)}$ of the posterior distribution of $\mathbf{s}_t$ with (27) and (28), and sample $\mathbf{s}_{tn}^{(i)}$ from it.

Note that with the above sampling order, the Monte Carlo estimate of the ELBO term maximized in the E-Z step (30) is given by (for source $n$):

$$\widehat{\mathcal{L}}_{\mathbf{z},n}(\theta_{\mathbf{s}},\theta_{\mathbf{z}},\phi_{\mathbf{z}};\mathbf{o}) = \sum_{t=1}^{T} \log p_{\theta_{\mathbf{s}}}(\mathbf{s}_{tn}^{(i)}|\mathbf{s}_{1:t-1,n}^{(i)},\mathbf{z}_{1:t,n}^{(i)}) - \sum_{t=1}^{T} D_{\mathrm{KL}}\big(q_{\phi_{\mathbf{z}}}(\mathbf{z}_{tn}|\mathbf{s}_{1:T,n}^{(i-1)},\mathbf{z}_{1:t-1,n}^{(i)})||p_{\theta_{\mathbf{z}}}(\mathbf{z}_{tn}|\mathbf{s}_{1:t-1,n}^{(i)},\mathbf{z}_{1:t-1,n}^{(i)})\big).$$

(35)

The whole MixDVAE algorithm, taking into account these practical aspects, is summarized in the form of pseudo-code in Algorithm 1.[7] In addition, Figure 2 shows a schematic overview of the algorithm.

## 4.6 Choice of the DVAE model

We recall that the DVAE is a general class of models that differ by adopting different conditional independence assumptions for the generative distributions in the right-hand-side of (4). In Girin et al. (2021), seven DVAE models from the literature have been extensively discussed, and six of them have been benchmarked on the analysis-resynthesis task (on speech signals and 3D human motion data). We chose to use here the stochastic recurrent neural network (SRNN) model initially proposed in Fraccaro et al. (2016), because it was shown in Girin et al. (2021) to provide a very good trade-off between model complexity and modeling power. The probabilistic dependencies of the SRNN generative model are defined as follows:

$$p_{\theta_{\mathbf{sz}}}(\mathbf{s}_{1:T},\mathbf{z}_{1:T}) = \prod_{t=1}^{T} p_{\theta_{\mathbf{s}}}(\mathbf{s}_t|\mathbf{s}_{1:t-1},\mathbf{z}_t)p_{\theta_{\mathbf{z}}}(\mathbf{z}_t|\mathbf{s}_{1:t-1},\mathbf{z}_{t-1}).$$

(36)

To perform online estimation, we use the following causal SRNN inference model:

$$q_{\phi_{\mathbf{z}}}(\mathbf{z}_{1:T}|\mathbf{s}_{1:T}) = \prod_{t=1}^{T} q_{\phi_{\mathbf{z}}}(\mathbf{z}_t|\mathbf{s}_{1:t},\mathbf{z}_{t-1}).$$

(37)

---

[7]As illustrated in Sections 5.2 and 6.2, in practice, we can choose different VEM step orders. Here we present the algorithm with the order E-S/E-Z Step, E-W Step and M Step.

The implementation details of the SRNN model can be found in Appendix D.

# 5  Application of MixDVAE to multiple object tracking

As mentioned in Section 1.3, under the tracking-by-detection configuration, the objective of the MOT task is to estimate the trajectories of moving objects from a set of given DBBs. In this case, the source vector $\mathbf{s}_{tn}$ represents the position of object $n$ at time frame $t$, which is given in practice by the coordinates of the (top-left and bottom-right points of the) "true" corresponding bounding box, i.e. $\mathbf{s}_{tn} = (s_{tn}^{\mathrm{L}}, s_{tn}^{\mathrm{T}}, s_{tn}^{\mathrm{R}}, s_{tn}^{\mathrm{B}}) \in \mathbb{R}^4$. The observation vector $\mathbf{o}_{tk} = (o_{tk}^{\mathrm{L}}, o_{tk}^{\mathrm{T}}, o_{tk}^{\mathrm{R}}, o_{tk}^{\mathrm{B}}) \in \mathbb{R}^4$ contains the coordinates of the (top-left and bottom-right points of the) $k$-th DBB at frame $t$. In a VAE or DVAE, the dimension $L$ of the latent vector $\mathbf{z}_{tn}$ is usually smaller than the dimension of the observed vector, in order to obtain a compact data representation. Since in the MOT task the data dimension is already small ($O = S = 4$), we also set $L = 4$. The sequence of estimated source position vectors is given directly by (28), for $n = 1$ to $N$ and $t = 1$ to $T$, directly forming source trajectories, with no further post-processing.

## 5.1  DVAE pre-training

**Dataset.** We consider pedestrian tracking for the MOT task and assume that all the moving sources have similar dynamical patterns. We thus pre-train a single DVAE model on a synthetic single-source trajectory dataset. This dataset contains synthetic bounding box trajectories in the form of $T$-frame sequences ($T = 60$) of 4D vectors $\{(x_t^{\mathrm{L}}, x_t^{\mathrm{T}}, x_t^{\mathrm{R}}, x_t^{\mathrm{B}})\}_{t=1}^T$. These trajectories are generated using piece-wise combinations of several elementary functions, namely: static $a(t) = a_0$, constant velocity $a(t) = a_1 t + a_0$, constant acceleration $a(t) = a_2 t^2 + a_1 t + a_0$, and sinusoidal (allowing for circular trajectories) $a(t) = a\sin(\omega t + \phi_0)$. The parameters $a_1$, $a_2$, $\omega$, and $\phi_0$ are sampled from some pre-defined distributions, whose parameters are estimated from the detections on the training subset of the MOT17 dataset (Dendorfer et al., 2021), which is a widely-used pedestrian tracking dataset (rapidly described it in the next subsection). The two remaining parameters, $a_0$ and $a$, are set to the values that ensure continuous trajectories. More details about the single-source synthetic trajectories generation can be found in Appendix E.1. Overall, we generated 12,105 sequences for the training dataset and 3,052 sequences for the validation dataset.

**Training details.** The SRNN model used in our experiments is an auto-regressive model, i.e., it uses the past source vectors $\mathbf{s}_{1:t-1}$ to predict the current one $\mathbf{s}_t$. In practice, the estimated past vectors are used for this prediction, rather than the ground-truth past vectors. To make the model robust to this problem, we trained the model in the scheduled sampling mode (Bengio et al., 2015). This means that during training, we gradually replace the ground-truth past values with the previously generated ones to predict the current value (see (Girin et al., 2021, Chapter 4) for a discussion on this issue). The model was trained using the Adam optimizer (Kingma & Ba, 2014) with a learning rate set to 0.001 and a batch size set to 256. An early-stopping strategy was adopted, with a patience of 50 epochs.

## 5.2  MixDVAE evaluation set-up

**Dataset.** For the evaluation of the proposed MixDVAE algorithm, we used the training set of MOT17. MOT17 contains pedestrian scenes filmed in different places such as in a shopping mall or in a street, with static or moving cameras. The motion patterns of the pedestrians in these videos are quite diverse and challenging. The MOT17 training set contains seven sequences with length varying from twenty seconds to one minute, with different frame rates (14, 25, and 30 fps). The ground-truth bounding boxes are provided, as well as the detection results obtained with three customized detectors, namely DPM (Felzenszwalb et al., 2010), Faster-RCNN (Ren et al., 2015), and SDP (Yang et al., 2016). As briefly stated in the introduction, we focus our study on modeling the source dynamics for multiple-source tasks. Therefore, we leave aside the problem of appearing/disappearing sources (usually referred to as birth/death processes) and consider a fixed number of $N = 3$ tracks. We have thus designed a new dataset from the MOT17 training set, which we call the MOT17-3T dataset. The MOT17-3T dataset uses the publicly-released DBBs of the MOT17 dataset. We split a complete video sequence into subsequences of sequence length $T$. Three values of $T$ are evaluated in our experiments: 60, 120, and 300 frames (respectively corresponding to 2, 4, and 10 seconds at 30 fps).

Each test sequence contains three source trajectories with possible occlusions and detection absences, see an example in Fig. 3. More details on the design of the MOT17-3T dataset can be found in Appendix E.2. We have finally created 1,712, 1,161, and 1,137 3-source test sequences of length $T = 60, 120$, and $300$ frames, respectively. Notice that the pre-trained DVAE is not fine-tuned on these test sequences.

**Algorithm initialization.** Before starting the iterations of the proposed VEM algorithm, we need to initialize the values of several parameters and variables. Theoretically, there is no preference in the order of the three E-steps. In practice, however, for initialization convenience, we followed the order E-W Step, E-Z/E-S Step. Indeed, starting with E-W Step requires the initialization of the mean vector and covariance matrix of the source vector posterior distribution $\mathbf{m}_{tn}, \mathbf{V}_{tn}$, the input vectors of the DVAE encoder $\mathbf{s}_{1:T,n}$ and the observation covariance matrices $\mathbf{\Phi}_{tk}$. For MOT, $\mathbf{m}_{tn}$ can be easily initialised over a short sequence by assuming that the source does not move too much. Indeed, the initial values of $\mathbf{m}_{tn}$ can be set to the value of the observed bounding box at the beginning of the sequence $\mathbf{m}_{0n}$. While this strategy is very straightforward to implement, it is too simple for many tracking scenarios, especially for long sequences. We thus propose to split a long sequence into sub-sequences. For each sub-sequence, we initialise $\mathbf{m}_{tn}$ to the value at the beginning of the sub-sequence. After this initialisation, we run a few iterations of the VEM algorithm over the sub-sequence, allowing us to have an estimate of the source position at the end of the sub-sequence. This value is then used to provide a constant initialisation for the next sub-sequence. At the end, all these initializations are concatenated, providing a piece-wise constant initialization for $\mathbf{m}_{tn}$ over the entire long sequence. More implementation details, as well as the pseudo-code of this cascade initialization strategy, are provided in Appendix C. The input vectors of the DVAE encoder are initialized with the same values as the ones used for $\mathbf{m}_{tn}$.

**Observation covariance matrix.** In our experiments, we observed that the estimated values of both $\mathbf{\Phi}_{tk}$ and $\boldsymbol{v}_{\theta_{\mathbf{s}},tn}$ in (28) increased very quickly with the VEM algorithm iterations. This caused instability and unbalance between these two terms, which finally conducted the whole model to diverge. To solve this problem, we set $\mathbf{\Phi}_{tk}$ to a given fixed value, which is constant on the whole analyzed $T$-frame sequence and not updated during the VEM iterations. Specifically, for the MOT task, $\mathbf{\Phi}_{tk}$ is set to a diagonal matrix, and the diagonal entries are set to $r_{\mathbf{\Phi}}^2 \left[ (o_{1k}^{\mathrm{R}} - o_{1k}^{\mathrm{L}})^2, (o_{1k}^{\mathrm{T}} - o_{1k}^{\mathrm{B}})^2, (o_{1k}^{\mathrm{R}} - o_{1k}^{\mathrm{L}})^2, (o_{1k}^{\mathrm{T}} - o_{1k}^{\mathrm{B}})^2 \right]$, where $r_{\mathbf{\Phi}}$ is a factor lower than 1. In common terms, $\mathbf{\Phi}_{tk}$ is set to a fraction of the (squared) size of the corresponding observation at frame 1. The covariance matrices $\mathbf{V}_{tn}$ are initialized with the same values as $\mathbf{\Phi}_{tk}$.

**Hyperparameters.** The VEM algorithm of MixDVAE has four hyperparameters to be set. The observation covariance matrix ratio $r_{\mathbf{\Phi}}$ is set to 0.04, the initialization subsequence length $J$ is set to 30, and the initialization iteration number $I_0$ is set to 20. The MixDVAE algorithm itself is run for $I = 70$ iterations, which was experimentally shown to lead to convergence.

**Baselines.** We compare our model with two recent state-of-the-art probabilistic MOT methods: The Autoregressive Tracklet Inpainting and Scoring for Tracking (ArTIST) model of Saleh et al. (2021) and the Variational Kalman Filter (VKF) of Ban et al. (2021). In addition to that, in order to demonstrate the advantage of using a DVAE model for modeling the dynamics of single-source trajectories, we consider replacing the DVAE model with a simpler deep auto-regressive (Deep AR) model. ArTIST is a supervised stochastic autoregressive model that learns the discretized multi-modal distribution of human motion using annotated MOT sequences. It can assign detections to tracks by scoring tracklet[8] proposals with their likelihood. And it can also generate continuations of the source trajectories and inpaint those containing missing detections. We have reused the trained models as well as the tracklet scoring and inpainting code provided by the authors[9] and reimplemented the object tracking part according to the paper, as this part was not provided. Implementation details can be found in Appendix F. Alike the proposed MixDVAE algorithm, the VKF algorithm for MOT (Ban et al., 2021) is also based on the VI methodology to combine source position estimation and detection-to-source assignment. However, a basic one-step linear dynamical model is used in VKF instead of the DVAE model in the proposed MixDVAE algorithm. In short, the dynamical model we use in VKF is $p_{\theta_{\mathbf{s}}}(\mathbf{s}_t|\mathbf{s}_{t-1}) = \prod_{n=1}^{N} \mathcal{N}(\mathbf{s}_{tn}; \mathbf{D}\mathbf{s}_{t-1,n}, \mathbf{\Lambda}_{tn})$, where $\mathbf{D}$ is assumed to be the identity matrix and $\mathbf{\Lambda}_{tn}$ is estimated in the M step. Hence, the VKF MOT algorithm is a combination of VI and Kalman filter update equations. In (Ban et al., 2021), the method was proposed in an audiovisual set-up. The

---

[8]A tracklet indicates a sequence of estimated position vectors consistent over time and assigned to the same object.

[9]available at https://github.com/fatemeh-slh/ArTIST

Table 2: MOT results for short ($T = 60$), medium ($T = 120$), and long ($T = 300$) sequences.

| Dataset | Method | MOTA↑ | MOTP↑ | IDF1↑ | #IDS↓ | %IDS↓ | MT↑ | ML↓ | #FP↓ | %FP↓ | #FN↓ | %FN↓ |
|---------|--------|-------|-------|-------|-------|-------|-----|-----|------|------|------|------|
| Short | ArTIST | 63.7 | **84.1** | 48.7 | 86371 | 28.0 | **4684** | **0** | 9962 | **3.2** | **15525** | **5.0** |
| | VKF | 56.0 | 82.7 | 77.3 | 5660 | 1.8 | 3742 | 761 | 64945 | 21.1 | 64945 | 21.1 |
| | Deep AR | 67.4 | 76.1 | 83.1 | 5248 | 1.7 | 3670 | 129 | 49595 | 16.0 | 49595 | 16.0 |
| | MixDVAE | **79.1** | 81.3 | **88.4** | **4966** | **1.6** | 4370 | 50 | 29808 | 9.7 | 29808 | 9.7 |
| Medium | ArTIST | 61.0 | **84.2** | 43.9 | 102978 | 24.6 | **2943** | **0** | 25388 | **6.1** | 34812 | **8.3** |
| | VKF | 57.5 | 83.3 | 77.6 | 7657 | 1.8 | 2563 | 487 | 85053 | 20.3 | 85053 | 20.3 |
| | Deep AR | 65.3 | 76.0 | 81.8 | **5387** | **1.3** | 2435 | 149 | 71775 | 17.0 | 71775 | 17.0 |
| | MixDVAE | **78.6** | 82.2 | **88.0** | 6107 | 1.5 | 2907 | 120 | 41747 | 9.9 | 41747 | 9.9 |
| Long | ArTIST | 53.5 | 84.5 | 40.7 | 205263 | 20.1 | 2513 | **4** | 135401 | 13.2 | 135401 | 13.2 |
| | VKF | 74.4 | **86.2** | 84.4 | 30069 | 2.9 | 2756 | 100 | 116160 | 11.4 | 116160 | 11.4 |
| | Deep AR | 75.5 | 76.6 | 87.1 | 26506 | 2.6 | 2555 | 18 | 123262 | 12.1 | 123262 | 12.1 |
| | MixDVAE | **83.2** | 82.4 | **90.0** | **23081** | **2.3** | **2890** | 12 | **74550** | **7.3** | **74550** | **7.3** |

observations contain not only the DBB coordinates, but also appearance features and multichannel audio recordings. For a fair comparison with MixDVAE, we use here the same observations, i.e., we simplified VKF by using only the DBB coordinates. For both ArTIST and VKF, the tracked sequences are initialized using the DBBs at the first frame, as what we have done for MixDVAE. For VKF, similarly to MixDVAE, we need to provide initial values for $\mathbf{m}_{tn}$ and $\mathbf{V}_{tn}$. For a fair comparison, we applied the same cascade initialization as the one presented above, except that a linear dynamical model is used in place of the DVAE to ensure the transition between two consecutive subsequences. The covariance matrices $\mathbf{V}_{tn}$ are initialized with pre-defined values that stabilize the EM algorithm. The covariance matrices $\boldsymbol{\Phi}_{tk}$ are fixed to the same values as for MixDVAE. The covariance matrices of the linear dynamical model (denoted $\boldsymbol{\Lambda}_{tn}$ in (Ban et al., 2021)) are initialized with the same values as $\mathbf{V}_{tn}$. Finally the simpler Deep AR baseline model is a deep generative model without stochastic latent variables. In this baseline, the dynamical model becomes $p_{\theta_\mathbf{s}}(\mathbf{s}_t|\mathbf{s}_{t-1}) = \prod_{n=1}^{N} \mathcal{N}(\mathbf{s}_{t,n}; \boldsymbol{\mu}_{\theta_\mathbf{s}}(\mathbf{s}_{1:t-1,n}), \mathrm{diag}(\mathbf{v}_{\theta_\mathbf{s}}(\mathbf{s}_{1:t-1,n})))$. In practice, the Deep AR model is implemented with an LSTM layer. The hidden dimension of the LSTM layer is set to match that of the LSTM layers employed in the DVAE model, i.e. it is equal to 8.

**Evaluation metrics.** We use the standard MOT metrics (Bernardin & Stiefelhagen, 2008; Ristani et al., 2016) to evaluate the tracking performance of MixDVAE and compare it to the baselines, namely: multi-object tracking accuracy (MOTA), multi-object tracking precision (MOTP), identity F1 score (IDF1), number of identity switches (IDS), mostly tracked (MT), mostly lost (ML), false positives (FP) and false negatives (FN). The three test subsets contain a different number of test sequences, with a different sequence length $T$. Therefore, for IDS, FP and FN, we report both the number of occurrences and the corresponding percentage. Among them, MOTA is considered to be the most representative metric. It is defined by aggregating the frame-wise versions of the metrics $\mathrm{FP}_t$, $\mathrm{FN}_t$, and $\mathrm{ID}_t$ over frames:

$$\mathrm{MOTA} = 1 - \frac{\sum_t (\mathrm{FN}_t + \mathrm{FP}_t + \mathrm{IDS}_t)}{\sum_t \mathrm{GT}_t}, \tag{38}$$

where $\mathrm{GT}_t$ denotes the number of ground-truth tracks at frame $t$. Higher MOTA values imply less errors (in terms of FPs, FNs, and IDS), and hence better tracking performance. MOTP defines the averaged overlap between all correctly matched sources and their corresponding ground truth. Higher MOTP implies more accurate position estimations. IDF1 is the ratio of correctly identified detections over the average number of ground-truth and computed detections. IDS reflects the capability of the model to preserve the identity of the tracked sources, especially in case of occlusion and track fragmentation. MT and ML represent how much the trajectory is recovered by the tracking algorithm. A source track is mostly tracked (resp. mostly lost) if it is covered by the tracker for at least 80% (resp. not more than 20%) of its life span.

## 5.3 Experimental results

**Quantitative analysis.** We now present and discuss the tracking results obtained with the proposed MixDVAE algorithm and compare them with those obtained with the baselines. In these experiments, the

value of the observation variance ratio $r_{\Phi}$ is set to 0.04 and no fine-tuning is applied to SRNN in the E-Z step. Ablation study on these factors is presented in Appendix I.

The values of the MOT metrics obtained on short, medium and long sequence subsets ($T = 60$, 120, and 300 frames, respectively) are shown in Table 2. We see that the proposed MixDVAE algorithm obtains the best MOTA scores for the three subsets (i.e., for the three different sequence length values). This is remarkable given that ArTIST was trained on the MOT17 training dataset, whereas MixDVAE never saw the ground-truth sequences before the test. Furthermore, we notice that both VKF and MixDVAE have much less IDS and much higher IDF1 scores than ArTIST, which implies that the observation-to-source assignment based on the VI method is more efficient than direct estimation of the position likelihood distribution to preserve the correct source identity during tracking. Besides, the MixDVAE model also has better scores than the VKF model for these two metrics, which implies that the DVAE-based dynamical model performs better on identity preservation than the linear dynamical model of VKF. For the 60- and 120-frame sequences, the ArTIST model has lower FP and FN percentages and higher MOTP scores (though the MOTP scores of all three algorithms are quite close for every value of $T$). This is reasonable because, again, ArTIST was trained on the same dataset using the ground-truth sequences while our model is unsupervised. Overall, the adverse effect caused by frequent identity switches is much greater than the positive effect of lower FP and FN for the ArTIST model. That explains why MixDVAE has much better MOTA scores than ArTIST. For the long (300-frame) sequences, MixDVAE obtains an overall much better performance than the ArTIST model, since it obtains here the best scores for 6 metrics out of 8, including FP and FN. This shows that MixDVAE is particularly good at tracking objects on the long term (we recall that $T = 300$ represents 10 s of video at 30 fps).

Besides, MixDVAE also globally exhibits notably better performance than VKF on all of the three datasets. This clearly indicates that the modeling of the sources dynamics with a DVAE model outperforms the use of a simple linear-Gaussian dynamical model and can greatly improve the tracking performance. We can also notice that the VKF algorithm globally performs much better on 300-frame sequences than on 60- and 120-frame sequences. One possible explanation for this phenomenon is that the dynamical patterns of long sequences are simpler than those of short and medium sequences. In fact, the data statistics show that the average velocity in long sequences is much lower than that in short and medium sequences. In this case, the linear dynamical model can perform quite well –although not as well as the DVAE.

Finally, we can see in Table 2 that MixDVAE with SRNN as dynamical model has an overall significantly better performance than MixDVAE with the baseline Deep AR dynamical model. This demonstrates the important role of the latent variables in SRNN for the dynamical modeling of sequential data. We remind that the latent vector $\mathbf{z}_{:,n}$ is assumed to efficiently encode the generative factors of source $n$'s trajectory.

**Qualitative analysis.** To illustrate the behavior of MixDVAE and the baseline models, we present an example of tracking result in Fig. 3. More examples can be found in Appendix G. In the example of Fig. 3, the detection for Source 3 ($o_3$ in the figure) is absent from $t = 2$ and reappears after $t = 20$. But we limit the plot to $t = 10$ for a better visualization. This is a case of long-term detection absence. An immediate identity switch occurs at $t = 2$ for the ArTIST model. Then, the track obtained by ArTIST is no longer stable. We speculate the reason for the frequent identity switches made by ArTIST is that the estimated distributions do not correspond well to the true sequential position distributions, which is possibly due to the way these distributions are discretized. In addition to the identity switches, the estimations generated by ArTIST at $t = 5$, 8, and 10 are not accurate. This causes a decrease of the tracking performance. For the VKF model, the estimated bounding boxes for Sources 2 and 3 ($m_2$ and $m_3$ in the figure) overlap each other. This means that the two observations are both assigned to the same source, which is Source 2. From (32) and (33), we know that the value of the assignment variable depends on the posterior mean and variance vectors $\mathbf{m}_{tn}$ and $\mathbf{V}_{tn}$, which themselves depend on the dynamical model. With a linear dynamical model, VKF is not able to correctly predict distinct $m_2$ and $m_3$ trajectories. The Deep AR model succeed to predict distinct $m_2$ and $m_3$ trajectories. However, the trajectory $m_3$ is not accurate due to the absence of $o_3$. In contrast, the very good dynamical modeling capacity of the DVAE makes MixDVAE able to keep tracking despite of the long-term detection absence and generate reasonable $m_3$ estimations, which correspond well to the ground-truth trajectory of Source 3 ($s_3$ in the figure).

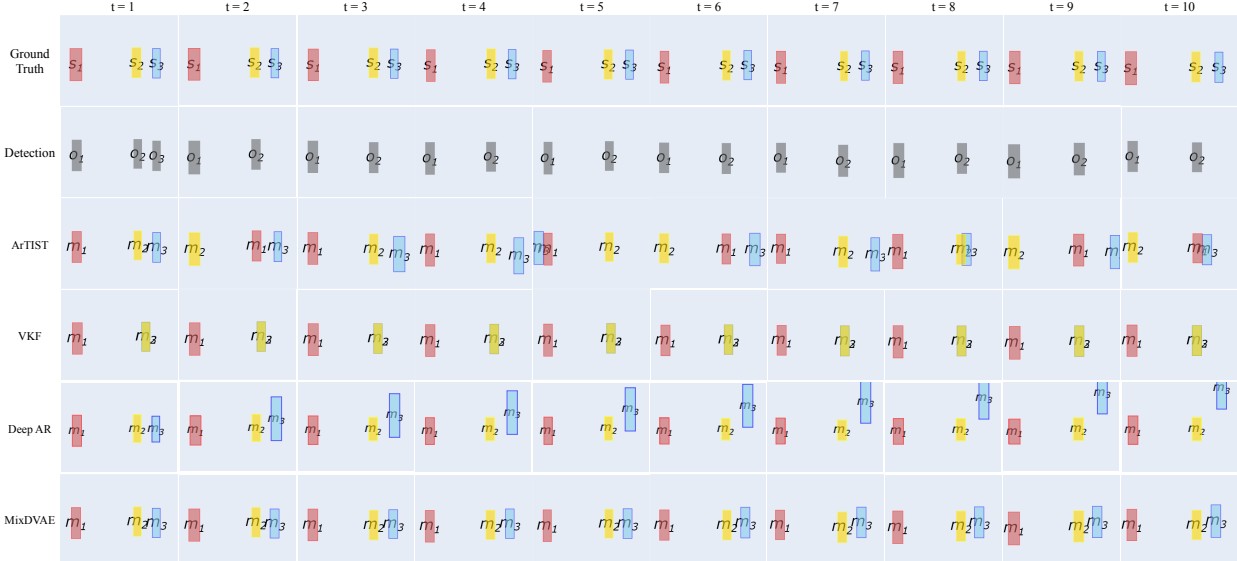

Figure 3: Example of tracking result obtained with the proposed MixDVAE algorithm and the two baselines. For clarity of presentation, the simplified notations $s_1$, $o_1$, and $m_1$ denote the ground-truth source position, the observation, and the estimated source position, respectively (for Source 1, and the same for the two other sources). Best seen in color.

# 6 Application of MixDVAE to single-channel audio source separation

When applying MixDVAE to the SC-ASS task, we work in the short-time Fourier transform (STFT) domain. This implies that both the source and observation vectors are complex-valued. More precisely, the $n$-th source vector $\mathbf{s}_{tn} = \{s_{tn,f}\}_{f=1}^F \in \mathbb{C}^F$ is the short-time spectrum of audio source $n$ at time frame $t$ ($f$ denotes the frequency bin and $\mathbf{s} \in \mathbb{C}^{T \times F}$ is the complete STFT spectrogram). The number of frequency bins, $F$, is typically set to 256, 512 or 1024 (a power of 2 is preferred to use the fast Fourier transform). As is usually adopted in audio processing, $\mathbf{s}_{tn}$ is assumed to follow a zero-mean circularly-symmetric complex Gaussian prior distribution (Févotte et al., 2009; Liutkus et al., 2011; Girin et al., 2019), i.e. (5) becomes $p_{\theta_\mathbf{s}}(\mathbf{s}_t | \mathbf{p}_t, \mathbf{z}_t) = \mathcal{N}_c(\mathbf{s}_t; \mathbf{0}, \mathrm{diag}(\boldsymbol{v}_{\theta_\mathbf{s}}(\mathbf{p}_t, \mathbf{z}_t)))$. The latent space dimension $L$ is typically set to a value significantly lower than $F$. Also, here, $\mathbf{o} = \{o_{tf}\}_{t,f=1}^{TF} \in \mathbb{C}^{F \times T}$ denotes the STFT spectrogram of the observed mixture signal. We define the $k$-th observation variable at frame $t$ as $\mathbf{o}_{tk} = o_{tf} \in \mathbb{C}$, which is the STFT coefficient of the mixture signal at TF bin $(t, f)$. In other words, the observation index $k$ is identified with the frequency bin/index $f$, and the total number of observations at any frame $t$ equals the number of frequency bins, i.e. $K_t = F$ for each $t$. We note that in this case, $\mathbf{s}_{tn}$ and $\mathbf{o}_{tk}$ do not have the same dimension, even though $\mathbf{s}_{tn}$ and $\mathbf{o}_{t,:}$ do. Therefore, as mentioned in Footnote 3, we need to define a projection matrix $\mathbf{P}_k \in \mathbb{C}^{1 \times F}$, which is here the transposed one-hot vector activated at the $k$-th index. Finally, the observation $\mathbf{o}_{tk} = o_{tf}$ is modeled with a conditional circularly-symmetric complex Gaussian, centered at the corresponding source coefficient, and (12) becomes $p_{\theta_\mathbf{o}}(\mathbf{o}_{tk} | w_{tk} = n, \mathbf{s}_{tn}) = \mathcal{N}_c(\mathbf{o}_{tk}; \mathbf{P}_k \mathbf{s}_{tn}, \boldsymbol{\Phi}_{tk})$. We can see that the assignment variable $w_{tk}$ associates each TF-bin of the observed mixture spectrogram to one of the sources, and thus implicitly defines a TF mask. All these adaptations yield minimal changes in the MixDVAE derivation and solution. These changes are provided in Appendix B, including the final source vector estimate. Note that the estimated $n$-th source waveform is obtained by applying the inverse STFT on $\mathbf{m}_{1:T,n}$.

## 6.1 DVAE pre-training

**Dataset.** We illustrate the application of MixDVAE to the SC-ASS problem with the separation of a speech signal and a musical instrument, in the present case the Chinese bamboo flute (CBF). These two audio sources have very different spectral and dynamical patterns. So, we choose here to pre-train two instances of the same DVAE model separately on two single-source datasets, a speech dataset and a CBF dataset.

For the speech dataset, we used the Wall Street Journal (WSJ0) dataset (Garofolo et al., 1993), which is composed of 16-kHz monophonic speech signals, with three subsets: *si_tr_s*, *si_dt_05* and *si_et_05*, used for model training, validation and test, and containing 24.9, 2.2 and 1.5 hours of speech recordings, respectively. For the musical instrument dataset, we used the CBF dataset of Wang et al. (2022), which contains CBF performances recorded by 10 professional CBF performers. The dataset comprises recordings of both isolated playing techniques and full-length pieces. We only used the full pieces recordings in our experiments. The original recordings are stereo and at a sampling frequency of 44.1kHz. In our experiments, we used only one channel and downsampled the signals to 16-kHz, to match the speech signals rate. We selected the second half pieces recordings of player 1 and 2 as the validation set, the second half pieces recordings of player 3, 4 and 5 as the test set, and use all other recordings for DVAE pre-training. The total duration of the training, validation and test sets are 2.1, 0.2 and 0.3 hours respectively. For both datasets, we used the training set for DVAE pre-training and the validation set for early stopping.

**Pre-processing.** For both the speech and CBF dataset, we pre-processed the raw audio signals in the following way. First, the silence at the beginning and end of each signal are trimmed with a voice activity detection threshold of 30 dB. Then, the waveform signals are normalized by dividing their absolute maximum value. The STFT coefficients are computed with a 64-ms sine window (1024 samples) and a 75%-overlap (256-sample hop length), resulting in sequences of 513-dimensional discrete Fourier coefficient vectors (for positive frequencies). Note that in practice, the DVAE model will input speech power spectrograms, i.e., the squared modulus of $\mathbf{s}$, instead of the complex-valued STFT spectrograms (Girin et al., 2021, Chapter 13), and these STFT power spectrograms are split into smaller sequences of length 50 frames (corresponding to audio segments of 0.8 s) for training.

**Training details.** We also used the SRNN model with scheduled sampling training (see Section 5.1). The model was trained with the Adam optimizer with a learning rate set to 0.002 and a batch size set to 256. The latent space dimension $L$ was set to 16. The early-stopping patience was set to 50 epochs for the WSJ0 dataset and 200 epochs for the CBF dataset.

## 6.2 MixDVAE evaluation set-up

**Dataset.** To generate the test mixture signals, we first randomly selected two signals from the WSJ0 test set and the CBF test set, respectively. Then, we removed the silence at the beginning and end in the same way as for the pre-processing. The clipped speech and CBF signals were mixed together with several different speech-to-music (power) ratios, namely $-10, -5, 0$ and 5 dB. The waveform mixture signals were then normalized and transformed to STFT spectrograms in the same way as in the pre-processing. Similar to the MOT scenario, we tested MixDVAE with different test sequence length values. To this aim, the mixed signal STFT spectrograms were split into subsequences of length 50, 100 and 300 frames (respectively corresponding to audio segments of 0.8, 1.6 and 4.8 s). Overall, we generated 878, 491, and 372 mixed test signals of length $T = 50, 100$ and 300 frames, respectively.

**Algorithm initialization.** As for MOT, we need to initialize the values of several parameters and variables. For SC-ASS, it is more difficult to obtain a reasonable initialization for $\mathbf{m}_{tn}$ (complex-valued) using directly the observed mixture signal. We thus choose the following VEM iteration order: E-Z/E-S Step, E-W Step. In this case, we have to initialize the posterior distribution of the assignment variable (i.e. all the values of $\eta_{tkn}$), the input vectors of the DVAE encoder $\mathbf{s}_{1:T,n}$ (for the two sources), and the observation model covariance matrices $\boldsymbol{\Phi}_{tk}$. We initialize $\eta_{tkn}$ with a discrete uniform distribution. As for the DVAE encoder, we first input the power spectrogram of the mixture signal (recall that the two DVAEs were pre-trained on different natural single-source datasets). We then use the reconstructed output power spectrograms as the initialization of the DVAE encoders input.

**Observation variance.** Similar to the MOT case, $\boldsymbol{\Phi}_{tk}$ is not estimated in the M Step, but fixed to $r_{\boldsymbol{\Phi}}^2|o_{tk}|^2$. In plain words, $\boldsymbol{\Phi}_{tk}$ is set to a fraction of the observation power. This setting turned out to stabilize the VEM iteration process and finally led to very satisfying estimation results.

**Hyperparameters.** Regarding the hyperparameters of the MixDVAE VEM algorithm for the SC-ASS task, the observation covariance matrix ratio $r_{\boldsymbol{\Phi}}$ is set to 0.01. The total number of iterations $I$ is set to 70. And the DVAE model is not fine-tuned in the E-Z step for the reported experiments.

**Baselines.** As mentioned in Section 1.4, the state-of-the-art SC-ASS methods are mostly fully supervised, thus requiring a very large amount of paired (aligned) mixture signals and individual source signals for training. Very few methods are under the weakly-supervised or unsupervised settings. Thus, it is difficult to find a fairly comparable baseline model. In the presented experiments, we have compared the proposed MixDVAE method with the unsupervised audio source separation method called MixIT (Wisdom et al., 2020) and with two weakly-supervised methods based on NMF, namely a vanilla NMF model (Févotte et al., 2018) and an NMF model with temporal extensions (Virtanen, 2007). MixIT is a deep-learning-based unsupervised single-channel source separation method. It is trained on a dataset constructed by mixing up the existing mixture audio signals. The model separates them into a variable number of latent source signals that can be remixed to approximate the original mixtures. In a totally unsupervised setting, MixIT does not require having the separated source signals for training. For the implementation of the MixIT model, we have reused the code provided by the authors and adapted it for the speech-CBF source separation task. In the weakly-supervised NMF baseline methods, an NMF model is first pre-trained on each single-source dataset separately, resulting in a dictionary of non-negative spectral templates $\mathbf{W}_n$ for each of the sources to separate. Such pre-training is similar in spirit to the pre-training stage of the MixDVAE method. After that, the obtained spectral template dictionaries of all sources are fixed and concatenated together so as to learn the temporal activation matrix $\mathbf{H}$ for the test mixture signal. Then the $\mathbf{H}$ entries corresponding to the spectral templates in $\mathbf{W}_n$ are used to separate source $n$ (in practice, Wiener filters are build to separate the sources in the STFT domain, see (Févotte et al., 2018) and (Virtanen, 2007) for details). We have re-implemented both NMF-based baselines according to the formula given in the corresponding papers. The latent dimension $K$ of the NMF model for both speech and CBF data is set to 128, which is determined by grid search. To demonstrate the interest of using a DVAE model for modeling the audio source dynamics in MixDVAE, we made additional experiments with replacing the DVAE model in MixDVAE with two other dynamical models: a linear-Gaussian dynamical model and a deep auto-regressive dynamical model, which results in baseline models similar to the VKF model and the Deep AR model that we have already used in our MOT experiments (see Section 5.2). For the VKF model, we initialize the values of $\eta_{tkn}$ in two ways: the ground-truth assignment mask, which is also named as ideal binary mask (IBM) in the audio source separation literature (we call the resulting model VKF-oracle) and the mask defined from the outputs of the pre-trained DVAEs when inputing the mixture signal spectrogram (we call the resulting model VKF-DVAE-init). Note that VKF-oracle provides an (unrealistic) upper bound of separation performance with a linear dynamical model, whereas VKF-DVAE-init uses the same initial information as MixDVAE. $\mathbf{\Phi}_{tk}$ are fixed to the same values as for MixDVAE and $\mathbf{\Lambda}_{tn}$ are initialized with the identity matrix multiplied by a scalar. For the Deep AR model, it is implemented with an LSTM layer, with the hidden dimension set equal to that of the LSTM layers employed in the DVAE model. Finally, to investigate the effects of the VEM algorithm in the MixDVAE model, we also compared our model with the direct reconstruction of the source signals from the output of the pre-trained DVAEs when using the mixture spectrogram as the input, i.e. the information used to intialize both MixDVAE and VKF-DVAE-init (we call this baseline method DVAE-init). As these output spectrograms are power spectrograms, we combined their square root (amplitude spectrogram) with the phase spectrogram of the mixture signal to reconstruct the waveform of the baseline separated signals.

**Evaluation metrics.** We used four source separation performance metrics widely-used in speech/audio processing. The root mean squared error (RMSE), the scale-invariant signal-to-distortion ratio (SI-SDR) (Roux et al., 2019) in dB, and the perceptual evaluation of speech quality (PESQ) score (Rix et al., 2001) (values in $[-0.5, 4.5]$).[10] For all metrics, the higher the better.

### 6.3 Experimental results

**Quantitative analysis.** We report the speech-CBF separation results on the short, medium and long test sequence subsets ($T = 50, 100, 300$ frames, respectively) in Table 3. In addition to the results obtained by the different models, we also report the values of the evaluation metrics when applied on the mixture signal, for reference.

---

[10]The PESQ objective measure was developed mostly for evaluating the quality of speech signals, but since it is largely based on a model of Human auditory perception, we assume we can also use it on the CBF sounds to avoid to complicate the evaluation protocol.

Table 3: SC-ASS results for short ($T = 50$), medium ($T = 100$), and long ($T = 300$) sequences.

| Dataset | Method | Speech | | | Chinese bamboo flute | | |
| | | RMSE ↓ | SI-SDR ↑ | PESQ ↑ | RMSE ↓ | SI-SDR ↑ | PESQ ↑ |
|---|---|---|---|---|---|---|---|
| Short | Mixture | 0.016 | -4.94 | 1.22 | 0.016 | 4.93 | 1.09 |
| | VKF-Oracle | 0.004 | 14.83 | 2.00 | 0.004 | 20.15 | 2.33 |
| | DVAE-init | 0.013 | -0.51 | 1.20 | 0.019 | 3.04 | 1.44 |
| | VKF-DVAE-init | 0.012 | 2.24 | 1.21 | 0.012 | 8.06 | 1.33 |
| | Deep AR | 0.009 | 5.32 | 1.29 | 0.018 | 5.19 | 1.48 |
| | MixIT | 0.011 | 3.26 | - | 0.009 | 7.15 | - |
| | Vanilla NMF | 0.011 | 3.01 | 1.40 | 0.012 | 9.09 | 1.37 |
| | Temporal NMF | 0.009 | 4.99 | 1.53 | 0.011 | 10.26 | 1.53 |
| | MixDVAE | **0.006** | **9.23** | **1.73** | **0.007** | **13.50** | **2.30** |
| Medium | Mixture | 0.016 | -4.44 | 1.17 | 0.016 | 4.44 | 1.08 |
| | VKF-Oracle | 0.004 | 14.88 | 1.88 | 0.003 | 20.24 | 2.41 |
| | DVAE-init | 0.014 | 0.10 | 1.15 | 0.020 | 2.42 | 1.27 |
| | VKF-DVAE-init | 0.013 | 1.25 | 1.12 | 0.013 | 7.42 | 1.26 |
| | Deep AR | 0.010 | 4.88 | 1.21 | 0.017 | 5.17 | 1.35 |
| | MixIT | 0.009 | 4.75 | - | 0.009 | 8.74 | - |
| | Vanilla NMF | 0.011 | 3.28 | 1.41 | 0.011 | 8.88 | 1.35 |
| | Temporal NMF | 0.010 | 5.12 | 1.48 | 0.011 | 9.96 | 1.44 |
| | MixDVAE | **0.007** | **9.32** | **1.65** | **0.007** | **13.05** | **2.16** |
| Long | Mixture | 0.016 | -4.52 | 1.19 | 0.016 | 4.53 | 1.10 |
| | VKF-Oracle | 0.004 | 14.65 | 1.89 | 0.003 | 20.45 | 2.60 |
| | DVAE-init | 0.013 | 0.20 | 1.15 | 0.020 | 2.29 | 1.22 |
| | VKF-DVAE-init | 0.013 | 0.34 | 1.10 | 0.013 | 7.35 | 1.24 |
| | Deep AR | 0.010 | 3.87 | 1.17 | 0.017 | 4.74 | 1.27 |
| | MixIT | **0.006** | **10.2** | - | 0.007 | 11.76 | - |
| | Vanilla NMF | 0.011 | 3.31 | 1.40 | 0.011 | 8.98 | 1.35 |
| | Temporal NMF | 0.010 | 5.01 | 1.47 | 0.011 | 10.06 | 1.42 |
| | MixDVAE | 0.007 | 9.06 | **1.64** | **0.007** | **12.92** | **2.06** |

We observe that on the short and medium sequence subsets, MixDVAE achieves the best performance for all of the evaluation metrics. While on the long sequences subsets, MixIT obtains slightly better results than MixDVAE for the speech. This demonstrates that the proposed method works well on the SC-ASS task. Unsurprisingly, VKF-Oracle obtains the best scores on all metrics because it was initialized with the ground-truth mask. When comparing MixDVAE with the methods of different dynamical models, we find that MixDVAE obtains overall better performance than both VKF-DVAE-init and Deep AR on all of the metrics for all of the three subsets. It is clear that the non-linear DVAE model with stochastic latent variables is much more efficient than the linear-Gaussian model and the deep auto-regressive model without latent variables for modeling the audio source dynamics. Besides, with the increase of the sequence length, the performance of MixDVAE dropped quite moderately (less than 0.6 dB and less than 0.3 dB in SI-SDR gain decrease on speech and CBF respectively), while the performance of VKF dropped by 2.32 dB on speech and by 0.71 dB on CBF, and the performance of Deep AR dropped by 1.45 dB on speech and by 0.45 dB on CBF.

Compared to DVAE-init, both MixDVAE and VKF-DVAE-init exhibit better separation performance (at least in terms of SI-SDR for VKF-DVAE-init). This indicates that the multi-source dynamical model with the observation-to-source assignment latent variable plays an important role in separating the content of different audio sources.

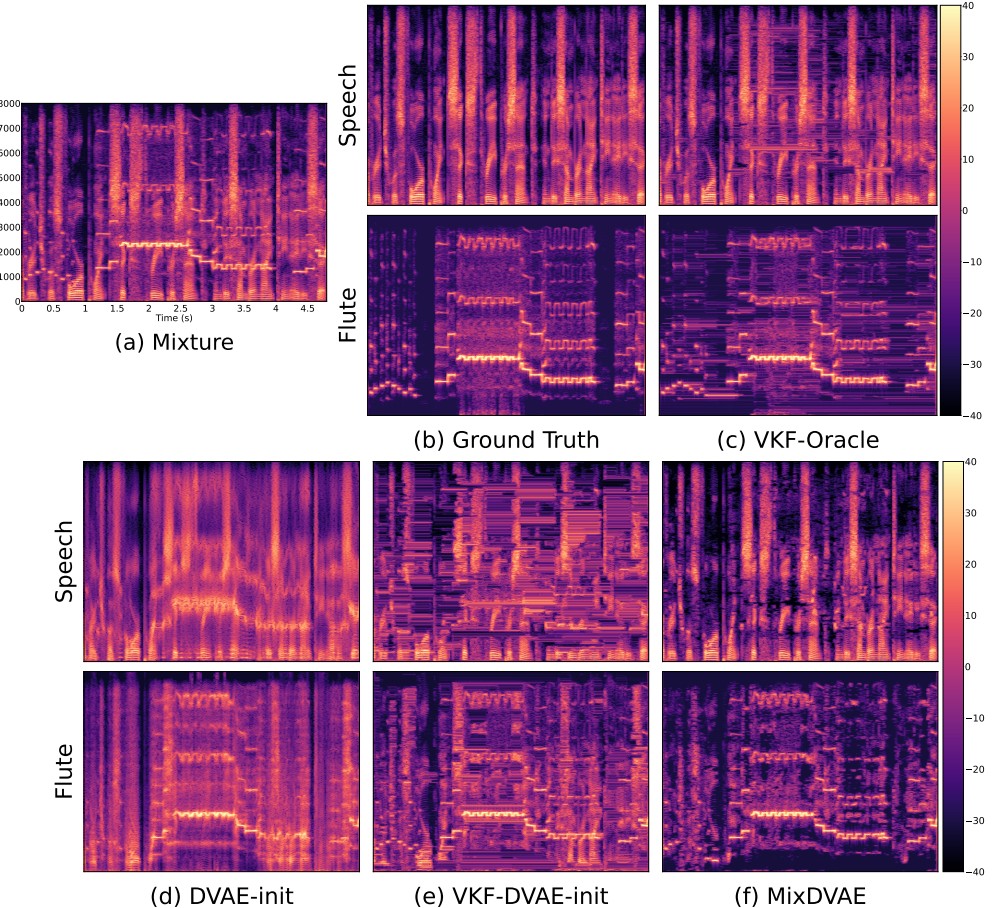

Figure 4: An example of audio source separation result obtained with the proposed MixDVAE algorithm and the baselines (speech and CBF power spectrograms). Best seen in color.

Although MixIT obtains slightly better performance than MixDVAE for speech on the long sequences subsets, its performance on short and medium sequences subsets is quite bad (in terms of SI-SDR, only 3.26 dB for speech and 7.15 dB for CBF on the short sequences subset, and 4.75 dB for speech and 8.74 dB for CBF on the medium sequences subset). As for the NMF based models, though adding temporal extensions to the vanilla NMF model indeed improves the model performance, the obtained results remain significantly inferior to that obtained by MixDVAE.

**Qualitative analysis.** To illustrate the behavior of the different models, we selected an audio source separation example and plotted the spectrograms in Fig. 4. More examples can be found in Appendix H. In the given example, the sequence length of the spectrograms is 300 frames. It is obvious that the ground-truth spectrograms of both the speech and CBF have spectral components with non-linear trajectories over time. Even though VKF-Oracle achieves the best performance, we observe in Subfigure (c) that there are several stationary traces (artifactual horizontal spectral lines) in the spectrograms caused by the inappropriate linear dynamics hypothesis. This phenomenon becomes even worse when VKF is initialized with the mask defined by the outputs of the pre-trained DVAEs (VKF-DVAE-init). In Subfigure (d), we clearly see the stationary traces, especially for the separated speech spectrogram. We believe that this is the reason why VKF-DVAE-init showed poor separation performance in general. When looking at the outputs of the pre-trained DVAE models, we find that the pre-trained DVAE models can provide a relatively good initialization for the VEM algorithm, even if we are still far from separated sources. In fact, since in the SC-ASS task, we pre-trained separately two DVAE models on the speech dataset and on the CBF dataset, the pre-trained DVAE models already have some prior information about the single-source dynamics. Though we only give the mixture

spectrogram as input, the pre-trained DVAE models can, to some extent, enhance the information of the source used in pre-training and attenuate the information of the other source. However, this kind of filtering is not very efficient. As we can see in the top figure of Subfigure (e), the output spectrogram provided by the DVAE model pre-trained on the speech dataset still keeps a significant amount of information on the CBF. Finally, even if the initialization is not that accurate, we see in Subfigure (f) that MixDVAE achieved a good separation of the two sources after running the VEM iterations.

## 7    Conclusion and future work

In this paper, we introduce MixDVAE, an LVGM designed to model the dynamics of multiple, jointly observed, sources. MixDVAE involves two main modules: A DVAE model for capturing the dynamics of each individual source and a discrete latent assignment variable that assign observations to sources, thus enabling us to form complete trajectories. The model learning process consists of two stages. During the first stage, the same or different DVAE model(s) is/are pre-trained on the synthetic or natural single-source trajectory dataset(s) to obtain prior information about the sources dynamics. During the second stage, the pre-trained DVAE model(s) is/are integrated into the general MixDVAE model. The entire MixDVAE model is solved using the VI framework with a VEM algorithm that combines the structured mean-field approximation and the amortized inference principles. The VEM algorithm is run directly on each multi-source test data sequence to process and the entire method does not require massive multi-source annotated datasets for training, which are difficult to obtain, especially for natural data. Hence, we consider it as weakly-supervised, as opposed to the fully-supervised approaches most commonly used in many multi-source processing applications. We illustrate the versatility of MixDVAE by applying it to two distinct scenarios: the MOT task and the SC-ASS task. Experimental results demonstrate that MixDVAE performs well on both tasks. Specifically, thanks to the strong dynamical modeling capacity of DVAE, MixDVAE shows to be more efficient than the combination of a linear dynamical model with the assignment variable. In addition, MixDVAE can generate reasonable predictions of the source vector even in the absence of observations, resulting in smooth and robust trajectories, as demonstrated in the MOT task. Our experiments demonstrate the generalization capability of MixDVAE trained on a synthetic single-target dataset, and evaluated in a multiple-target dataset. Finally, we believe that MixDVAE has a very strong potential for modeling the dynamics of multiple-source systems in general, and can be applied to various other tasks. However, we acknowledge that MixDVAE also has certain limitations, such as the assumption that each source behaves independently and the lack of consideration for interactions among them. We leave this as a challenging topic for future research.

### Acknowledgments

This research was partially funded by the Horizon 2020 SPRING project funded by the European Commission (under GA #871245), by the French Research Agency Young Researchers Program ML3RI project (under GA #ANR-19-CE33-0008-01) and by the Multidisciplinary Institute of Artificial Intelligence (under GA #ANR-19-P3IA-0003).

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

# A MixDVAE algorithm calculation details

## A.1 E-S Step

Here we detail the calculation of the posterior distribution $q_{\phi_{\mathbf{s}}}(\mathbf{s}|\mathbf{o})$. Using (11), the first expectation term in (23) can be developed as:

$$
\mathbb{E}_{q_{\phi_{\mathbf{w}}}(\mathbf{w}|\mathbf{o})}\big[\log p_{\theta_{\mathbf{o}}}(\mathbf{o}|\mathbf{w},\mathbf{s})\big] = \mathbb{E}_{q_{\phi_{\mathbf{w}}}(\mathbf{w}|\mathbf{o})}\left[\sum_{t=1}^{T}\sum_{k=1}^{K_t}\log p_{\theta_{\mathbf{o}}}(\mathbf{o}_{tk}|w_{tk},\mathbf{s}_{t,1:N})\right]
$$

$$
= \sum_{t=1}^{T}\sum_{k=1}^{K_t}\mathbb{E}_{q_{\phi_{\mathbf{w}}}(w_{tk}|\mathbf{o}_{tk})}\big[\log p_{\theta_{\mathbf{o}}}(\mathbf{o}_{tk}|w_{tk},\mathbf{s}_{t,1:N})\big]. \tag{39}
$$

Since for any pair $(t,k)$, the assignment variable $w_{tk}$ follows a discrete posterior distribution, we can denote its values by

$$
q_{\phi_{\mathbf{w}}}(w_{tk}=n|\mathbf{o}_{tk}) = \eta_{tkn},
$$

which will be calculated later in the E-W Step. With this notation, we have:

$$
\mathbb{E}_{q_{\phi_{\mathbf{w}}}(\mathbf{w}|\mathbf{o})}\big[\log p_{\theta_{\mathbf{o}}}(\mathbf{o}|\mathbf{w},\mathbf{s})\big] = \sum_{t=1}^{T}\sum_{k=1}^{K_t}\sum_{n=1}^{N}\eta_{tkn}\log p_{\theta_{\mathbf{o}}}(\mathbf{o}_{tk}|w_{tk}=n,\mathbf{s}_{tn}). \tag{40}
$$

The second expectation in (23) cannot be computed analytically as a distribution on $\mathbf{s}$ because of the non-linearity in the decoder and in the encoder. In order to avoid a tedious sampling procedure and obtain a computationally efficient solution, we further approximate this term by assuming $q_{\phi_{\mathbf{z}}}(\mathbf{z}|\mathbf{s}) \approx q_{\phi_{\mathbf{z}}}(\mathbf{z}|\mathbf{s}=\mathbf{m}^{(i-1)})$, where $\mathbf{m}^{(i-1)}$ is the mean value of the posterior distribution of $\mathbf{s}$ estimated at the previous iteration. By using this approximation, the term $\mathbb{E}_{q_{\phi_{\mathbf{z}}}(\mathbf{z}|\mathbf{s})}\big[\log q_{\phi_{\mathbf{z}}}(\mathbf{z}|\mathbf{s})\big]$ is now considered as a constant.

In addition, we observe that the second term of (23) can be rewritten as:

$$
\mathbb{E}_{q_{\phi_{\mathbf{z}}}(\mathbf{z}|\mathbf{s})}\big[\log p_{\theta_{\mathbf{sz}}}(\mathbf{s},\mathbf{z})\big] = \sum_{n=1}^{N}\mathbb{E}_{q_{\phi_{\mathbf{z}}}(\mathbf{z}_{:,n}|\mathbf{m}_{:,n}^{(i-1)})}\big[\log p_{\theta_{\mathbf{sz}}}(\mathbf{s}_{:,n},\mathbf{z}_{:,n})\big], \tag{41}
$$

since both the DVAE joint distribution and posterior distribution factorise over the sources, as formalized in (15) and (17). As a consequence, the posterior distribution of $\mathbf{s}$ factorises over the sources too:

$$
q_{\phi_{\mathbf{s}}}(\mathbf{s}|\mathbf{o}) = \prod_{n=1}^{N}q_{\phi_{\mathbf{s}}}(\mathbf{s}_{:,n}|\mathbf{o}), \tag{42}
$$

and therefore:

$$
q_{\phi_{\mathbf{s}}}(\mathbf{s}_{:,n}|\mathbf{o}) \propto \exp\left(\mathbb{E}_{q_{\phi_{\mathbf{z}}}(\mathbf{z}_{:,n}|\mathbf{m}_{:,n}^{(i-1)})}\big[\log p_{\theta_{\mathbf{sz}}}(\mathbf{s}_{:,n},\mathbf{z}_{:,n})\big]\right)\prod_{t=1}^{T}\prod_{k=1}^{K_t}\exp\big(\eta_{tkn}\log p_{\theta_{\mathbf{o}}}(\mathbf{o}_{tk}|w_{tk}=n,\mathbf{s}_{tn})\big).
$$

In the above equation, the expectation term cannot be calculated in closed form. As usually done in the DVAE methodology, it is thus replaced by a Monte Carlo estimate using sampled sequences drawn from the DVAE inference model. Let us denote by $\mathbf{z}_{:,n}^{(i)} \sim q_{\phi_{\mathbf{z}}}(\mathbf{z}_{:,n}|\mathbf{m}_{:,n}^{(i-1)})$ such a sampled sequence. In the present work, we use single point estimate, thus obtaining:

$$
q_{\phi_{\mathbf{s}}}(\mathbf{s}_{:,n}|\mathbf{o}) \propto p_{\theta_{\mathbf{sz}}}(\mathbf{s}_{:,n},\mathbf{z}_{:,n}^{(i)})\prod_{t=1}^{T}\prod_{k=1}^{K_t}\exp\big(\eta_{tkn}\log p_{\theta_{\mathbf{o}}}(\mathbf{o}_{tk}|w_{tk}=n,\mathbf{s}_{tn})\big)
$$

$$
\propto \prod_{t=1}^{T}\left(p_{\theta_{\mathbf{s}}}(\mathbf{s}_{tn}|\mathbf{s}_{1:t-1,n},\mathbf{z}_{1:t,n}^{(i)})p_{\theta_{\mathbf{z}}}(\mathbf{z}_{tn}^{(i)}|\mathbf{s}_{1:t-1,n},\mathbf{z}_{1:t-1,n}^{(i)})\prod_{k=1}^{K_t}\exp\big(\eta_{tkn}\log p_{\theta_{\mathbf{o}}}(\mathbf{o}_{tk}|w_{tk}=n,\mathbf{s}_{tn})\big)\right). \tag{43}
$$

We observe that the $t$-th element of the previous factorisation is a distribution over $\mathbf{s}_{tn}$ conditioned by $\mathbf{s}_{1:t-1,n}$. As for $q_{\phi_{\mathbf{z}}}(\mathbf{z}_{:,n}|\mathbf{s}_{:,n})$, the dependency with $\mathbf{s}_{1:t-1,n}$ is non-linear and therefore would impede to obtain a computationally efficient closed-form solution. In the same attempt of avoiding costly sampling strategies, we approximate the previous expression replacing $\mathbf{s}_{1:t-1,n}$ with $\mathbf{s}_{1:t-1,n}^{(i)}$, obtaining:

$$q_{\phi_{\mathbf{s}}}(\mathbf{s}_{:,n}|\mathbf{o}) \approx \prod_{t=1}^{T} q_{\phi_{\mathbf{s}}}(\mathbf{s}_{tn}|\mathbf{s}_{1:t-1,n}^{(i)}, \mathbf{o}), \tag{44}$$

with

$$q_{\phi_{\mathbf{s}}}(\mathbf{s}_{tn}|\mathbf{s}_{1:t-1,n}^{(i)}, \mathbf{o}) \propto p_{\theta_{\mathbf{s}}}(\mathbf{s}_{tn}|\mathbf{s}_{1:t-1,n}^{(i)}, \mathbf{z}_{1:t,n}^{(i)}) \prod_{k=1}^{K_t} \exp\left(\eta_{tkn} \log p_{\theta_{\mathbf{o}}}(\mathbf{o}_{tk}|w_{tk} = n, \mathbf{s}_{tn})\right), \tag{45}$$

since the term $p_{\theta_{\mathbf{z}}}(\mathbf{z}_{tn}^{(i)}|\mathbf{s}_{1:t-1,n}^{(i)}, \mathbf{z}_{1:t-1,n}^{(i)})$ becomes a constant.

Another interesting consequence of sampling $\mathbf{s}_{1:t-1,n}$ is that the dependency with the future observations of $q_{\phi_{\mathbf{s}}}(\mathbf{s}_{tn}|\mathbf{s}_{1:t-1,n}^{(i)}, \mathbf{o})$ disappears. Indeed, since we are sampling *at every time step*, the future posterior distributions $q_{\phi_{\mathbf{s}}}(\mathbf{s}_{t+k,n}|\mathbf{s}_{1:t+k-1,n}^{(i)}, \mathbf{o})$ do not depend on $\mathbf{s}_{tn}$, and therefore the posterior distribution of $\mathbf{s}_{tn}$ will not depend on the future observations.

The two distributions in the above equation are Gaussian distributions defined in (5), and (12). Therefore, it can be shown that the variational posterior distribution of $\mathbf{s}_{tn}$ is a Gaussian distribution: $q_{\phi_{\mathbf{s}}}(\mathbf{s}_{tn}|\mathbf{s}_{1:t-1,n}^{(i)}, \mathbf{o}) = \mathcal{N}(\mathbf{s}_{tn}; \mathbf{m}_{tn}, \mathbf{V}_{tn})$ with covariance matrix and mean vector provided in (27) and (28) respectively, and recalled here for completeness:

$$\mathbf{V}_{tn} = \left(\sum_{k=1}^{K_t} \eta_{tkn} \boldsymbol{\Phi}_{tk}^{-1} + \mathrm{diag}(\boldsymbol{v}_{\theta_{\mathbf{s}}, tn}^{(i)})^{-1}\right)^{-1}, \tag{46}$$

$$\mathbf{m}_{tn} = \mathbf{V}_{tn}\left(\sum_{k=1}^{K_t} \eta_{tkn} \boldsymbol{\Phi}_{tk}^{-1} \mathbf{o}_{tk} + \mathrm{diag}(\boldsymbol{v}_{\theta_{\mathbf{s}}, tn}^{(i)})^{-1} \boldsymbol{\mu}_{\theta_{\mathbf{s}}, tn}^{(i)}\right), \tag{47}$$

where $\boldsymbol{v}_{\theta_{\mathbf{s}}, tn}^{(i)}$ and $\boldsymbol{\mu}_{\theta_{\mathbf{s}}, tn}^{(i)}$ are simplified notations for $\boldsymbol{v}_{\theta_{\mathbf{s}}}(\mathbf{s}_{1:t-1,n}^{(i)}, \mathbf{z}_{1:t,n}^{(i)})$ and $\boldsymbol{\mu}_{\theta_{\mathbf{s}}}(\mathbf{s}_{1:t-1,n}^{(i)}, \mathbf{z}_{1:t,n}^{(i)})$ respectively, denoting the variance and mean vector estimated by the DVAE for source $n$ at time frame $t$.

## A.2  E-Z Step

Here we detail the calculation of the ELBO term (29).

$$\begin{aligned}
\mathcal{L}(\theta_{\mathbf{s}}, \theta_{\mathbf{z}}, \phi_{\mathbf{z}}; \mathbf{o}) &= \mathbb{E}_{q_{\phi_{\mathbf{s}}}(\mathbf{s}|\mathbf{o})}\left[\mathbb{E}_{q_{\phi_{\mathbf{z}}}(\mathbf{z}|\mathbf{s})}\left[\log p_{\theta_{\mathbf{sz}}}(\mathbf{s}, \mathbf{z}) - \log q_{\phi_{\mathbf{z}}}(\mathbf{z}|\mathbf{s})\right]\right] \\
&= \mathbb{E}_{\prod_{n=1}^{N} q_{\phi_{\mathbf{s}}}(\mathbf{s}_{:,n}|\mathbf{o})}\left[\mathbb{E}_{\prod_{n=1}^{N} q_{\phi_{\mathbf{z}}}(\mathbf{z}_{:,n}|\mathbf{s}_{:,n})}\left[\sum_{n=1}^{N} \log p_{\theta_{\mathbf{sz}}}(\mathbf{s}_{:,n}, \mathbf{z}_{:,n})\right] - \mathbb{E}_{\prod_{n=1}^{N} q_{\phi_{\mathbf{z}}}(\mathbf{z}_{:,n}|\mathbf{s}_{:,n})}\left[\sum_{n=1}^{N} \log q_{\phi_{\mathbf{z}}}(\mathbf{z}_{:,n}|\mathbf{s}_{:,n})\right]\right] \\
&= \sum_{n=1}^{N} \mathbb{E}_{q_{\phi_{\mathbf{s}}}(\mathbf{s}_{:,n}|\mathbf{o})}\left[\mathbb{E}_{q_{\phi_{\mathbf{z}}}(\mathbf{z}_{:,n}|\mathbf{s}_{:,n})}\left[\log p_{\theta_{\mathbf{sz}}}(\mathbf{s}_{:,n}, \mathbf{z}_{:,n})\right] - \mathbb{E}_{q_{\phi_{\mathbf{z}}}(\mathbf{z}_{:,n}|\mathbf{s}_{:,n})}\left[\log q_{\phi_{\mathbf{z}}}(\mathbf{z}_{:,n}|\mathbf{s}_{:,n})\right]\right] \\
&= \sum_{n=1}^{N} \mathcal{L}_n(\theta_{\mathbf{s}}, \theta_{\mathbf{z}}, \phi_{\mathbf{z}}; \mathbf{o}), \tag{48}
\end{aligned}$$

with

$$\mathcal{L}_n(\theta_{\mathbf{s}}, \theta_{\mathbf{z}}, \phi_{\mathbf{z}}; \mathbf{o}) = \mathbb{E}_{q_{\phi_{\mathbf{s}}}(\mathbf{s}_{:,n}|\mathbf{o})}\left[\mathbb{E}_{q_{\phi_{\mathbf{z}}}(\mathbf{z}_{:,n}|\mathbf{s}_{:,n})}\left[\log p_{\theta_{\mathbf{sz}}}(\mathbf{s}_{:,n}, \mathbf{z}_{:,n})\right] - \mathbb{E}_{q_{\phi_{\mathbf{z}}}(\mathbf{z}_{:,n}|\mathbf{s}_{:,n})}\left[\log q_{\phi_{\mathbf{z}}}(\mathbf{z}_{:,n}|\mathbf{s}_{:,n})\right]\right]. \tag{49}$$

### A.3 E-W Step

Here we detail the calculation of the posterior distribution $q_{\phi_{\mathbf{w}}}(\mathbf{w}|\mathbf{o})$. Applying the optimal update equation (2) to $\mathbf{w}$, we have:

$$q_{\phi_{\mathbf{w}}}(\mathbf{w}|\mathbf{o}) \propto \exp\left(\mathbb{E}_{q_{\phi_{\mathbf{s}}}(\mathbf{s}|\mathbf{o})q_{\phi_{\mathbf{z}}}(\mathbf{z}|\mathbf{s})}\left[\log p_\theta(\mathbf{o}, \mathbf{w}, \mathbf{s}, \mathbf{z})\right]\right). \tag{50}$$

Using (10), we derive:

$$q_{\phi_{\mathbf{w}}}(\mathbf{w}|\mathbf{o}) \propto p_{\theta_{\mathbf{w}}}(\mathbf{w})\exp\left(\mathbb{E}_{q_{\phi_{\mathbf{s}}}(\mathbf{s}|\mathbf{o})}\left[\log p_{\theta_{\mathbf{o}}}(\mathbf{o}|\mathbf{w}, \mathbf{s})\right]\right). \tag{51}$$

Using (11), the expectation term can be developed as:[11]

$$\mathbb{E}_{q_{\phi_{\mathbf{s}}}(\mathbf{s}|\mathbf{o})}\left[\log p_{\theta_{\mathbf{o}}}(\mathbf{o}|\mathbf{w}, \mathbf{s})\right] = \mathbb{E}_{q_{\phi_{\mathbf{s}}}(\mathbf{s}|\mathbf{o})}\left[\sum_{t=1}^{T}\sum_{k=1}^{K_t}\log p_{\theta_{\mathbf{o}}}(\mathbf{o}_{tk}|w_{tk}, \mathbf{s}_{t,:})\right]$$

$$= \sum_{t=1}^{T}\sum_{k=1}^{K_t}\mathbb{E}_{q_{\phi_{\mathbf{s}}}(\mathbf{s}_{t,:}|\mathbf{o})}\left[\log p_{\theta_{\mathbf{o}}}(\mathbf{o}_{tk}|w_{tk}, \mathbf{s}_{t,:})\right]. \tag{52}$$

Combining (13) and the previous result, we have:

$$q_{\phi_{\mathbf{w}}}(\mathbf{w}|\mathbf{o}) \propto \prod_{t=1}^{T}\prod_{k=1}^{K_t} p_{\theta_{\mathbf{w}}}(w_{tk})\exp\left(\mathbb{E}_{q_{\phi_{\mathbf{s}}}(\mathbf{s}_{t,:}|\mathbf{o})}\left[\log p_{\theta_{\mathbf{o}}}(\mathbf{o}_{tk}|w_{tk}, \mathbf{s}_{t,:})\right]\right), \tag{53}$$

which we can rewrite

$$q_{\phi_{\mathbf{w}}}(\mathbf{w}|\mathbf{o}) \propto \prod_{t=1}^{T}\prod_{k=1}^{K_t} q_{\phi_{\mathbf{w}}}(w_{tk}|\mathbf{o}), \tag{54}$$

with

$$q_{\phi_{\mathbf{w}}}(w_{tk}|\mathbf{o}) = p_{\theta_{\mathbf{w}}}(w_{tk})\exp\left(\mathbb{E}_{q_{\phi_{\mathbf{s}}}(\mathbf{s}_{t,:}|\mathbf{o})}\left[\log p_{\theta_{\mathbf{o}}}(\mathbf{o}_{tk}|w_{tk}, \mathbf{s}_{t,:})\right]\right). \tag{55}$$

The assignment variable $w_{tk}$ follows a discrete distribution and we denote:

$$\eta_{tkn} = q_{\phi_{\mathbf{w}}}(w_{tk} = n|\mathbf{o}) \propto p_{\phi_{\mathbf{w}}}(w_{tk} = n)\exp\left(\mathbb{E}_{q_{\phi_{\mathbf{s}}}(\mathbf{s}_{tn}|\mathbf{o})}\left[\log p_{\theta_{\mathbf{o}}}(\mathbf{o}_{tk}|w_{tk} = n, \mathbf{s}_{tn})\right]\right). \tag{56}$$

Using the fact that both $p_{\theta_{\mathbf{o}}}(\mathbf{o}_{tk}|w_{tk} = n, \mathbf{s}_{tn})$ and $q_{\phi_{\mathbf{s}}}(\mathbf{s}_{tn}|\mathbf{o})$ are multivariate Gaussian distributions (defined in (12) and (26)–(28), respectively), the previous expectation can be calculated in closed form:

$$\mathbb{E}_{q_{\phi_{\mathbf{s}}}(\mathbf{s}_{tn}|\mathbf{o})}\left[\log p_{\theta_{\mathbf{o}}}(\mathbf{o}_{tk}|w_{tk} = n, \mathbf{s}_{tn})\right] = \int_{\mathbf{s}_{tn}} \mathcal{N}(\mathbf{s}_{tn}; \mathbf{m}_{tn}, \mathbf{V}_{tn})\log\mathcal{N}(\mathbf{o}_{tk}; \mathbf{s}_{tn}, \boldsymbol{\Phi}_{tk})d\mathbf{s}_{tn}$$

$$= -\frac{1}{2}\left[\log|\boldsymbol{\Phi}_{tk}| + (\mathbf{o}_{tk} - \mathbf{m}_{tn})^T\boldsymbol{\Phi}_{tk}^{-1}(\mathbf{o}_{tk} - \mathbf{m}_{tn}) + \mathrm{Tr}\left(\boldsymbol{\Phi}_{tk}^{-1}\mathbf{V}_{tn}\right)\right]. \tag{57}$$

By using (14), the previous result, and normalizing to 1, we finally get:

$$\eta_{tkn} = \frac{\beta_{tkn}}{\sum_{i=1}^{N}\beta_{tki}}, \tag{58}$$

where

$$\beta_{tkn} = \mathcal{N}(\mathbf{o}_{tk}; \mathbf{m}_{tn}, \boldsymbol{\Phi}_{tk})\exp\left(-\frac{1}{2}\mathrm{Tr}\left(\boldsymbol{\Phi}_{tk}^{-1}\mathbf{V}_{tn}\right)\right). \tag{59}$$

---

[11]In fact, the posterior distribution $q_{\phi_{\mathbf{s}}}(\mathbf{s}_{t,:}|\mathbf{o})$ is also conditioned on $\mathbf{s}_{1:t-1,:}$ and $\mathbf{z}_{1:t,:}$. We use this abuse of notation for concision.

### A.4 M Step

Here we detail the calculation of $\boldsymbol{\Phi}_{tk}$. In the ELBO expression (19), only the first term depends on $\theta_{\mathbf{o}}$:

$$\mathcal{L}(\theta_{\mathbf{o}}; \mathbf{o}) = \mathbb{E}_{q_{\phi_{\mathbf{w}}}(\mathbf{w}|\mathbf{o})q_{\phi_{\mathbf{s}}}(\mathbf{s}|\mathbf{o})}\big[\log p_{\theta_{\mathbf{o}}}(\mathbf{o}|\mathbf{w}, \mathbf{s})\big]$$

$$= \sum_{n=1}^{N}\sum_{t=1}^{T}\sum_{k=1}^{K_t} \eta_{tkn} \int_{\mathbf{s}_{tn}} \mathcal{N}(\mathbf{s}_{tn}; \mathbf{m}_{tn}, \mathbf{V}_{tn}) \log \mathcal{N}(\mathbf{o}_{tk}; \mathbf{s}_{tn}, \boldsymbol{\Phi}_{tk})d\mathbf{s}_{tn}$$

$$= -\frac{1}{2}\sum_{n=1}^{N}\sum_{t=1}^{T}\sum_{k=1}^{K_t} \eta_{tkn}\Big[\log|\boldsymbol{\Phi}_{tk}| + (\mathbf{o}_{tk} - \mathbf{m}_{tn})^T\boldsymbol{\Phi}_{tk}^{-1}(\mathbf{o}_{tk} - \mathbf{m}_{tn}) + \mathrm{Tr}(\boldsymbol{\Phi}_{tk}^{-1}\mathbf{V}_{tn})\Big]. \quad (60)$$

By computing the derivative of $\mathcal{L}(\theta_{\mathbf{o}}; \mathbf{o})$ with respect to $\boldsymbol{\Phi}_{tk}$ and setting it to 0, we find the optimal value of $\boldsymbol{\Phi}_{tk}$ that maximizes the ELBO:

$$\boldsymbol{\Phi}_{tk} = \sum_{n=1}^{N} \eta_{tkn}\Big((\mathbf{o}_{tk} - \mathbf{m}_{tn})(\mathbf{o}_{tk} - \mathbf{m}_{tn})^T + \mathbf{V}_{tn}\Big). \quad (61)$$

## B  Formulas for SC-ASS

With the adaptations in the model mentioned at the beginning of Section 6 for the SC-ASS task, the solution formulas are as following. In the E-S Step, (27) and (28) become:

$$\mathbf{V}_{tn} = \Big(\sum_{k=1}^{K_t} \eta_{tkn}\mathbf{P}_k^T\boldsymbol{\Phi}_{tk}^{-1} + \mathrm{diag}(\boldsymbol{v}_{\theta_{\mathbf{s}},tn}^{(i)})^{-1}\Big)^{-1}, \quad (62)$$

$$\mathbf{m}_{tn} = \mathbf{V}_{tn}\Big(\sum_{k=1}^{K_t} \eta_{tkn}\mathbf{P}_k^T\boldsymbol{\Phi}_{tk}^{-1}\mathbf{o}_{tk}\Big). \quad (63)$$

The E-Z Step is not changed. In the E-W Step, (33) become:

$$\beta_{tkn} = \mathcal{N}_c(\mathbf{o}_{tk}; \mathbf{P}_k\mathbf{m}_{tn}, \boldsymbol{\Phi}_{tk})\exp\Big(-\frac{1}{2}\mathrm{Tr}\big(\mathbf{P}_k^T\boldsymbol{\Phi}_{tk}^{-1}\mathbf{P}_k\mathbf{V}_{tn}\big)\Big). \quad (64)$$

Finally, in the M Step, (34) become:

$$\boldsymbol{\Phi}_{tk} = \sum_{n=1}^{N} \eta_{tkn}\Big((\mathbf{o}_{tk} - \mathbf{P}_k\mathbf{m}_{tn})(\mathbf{o}_{tk} - \mathbf{P}_k\mathbf{m}_{tn})^T + \mathbf{P}_k\mathbf{V}_{tn}\mathbf{P}_k^T\Big). \quad (65)$$

## C  Cascade initialization in MOT

For the initialization of the source (position) vector, we first split the long sequence indexed by $t \in \{1, 2, ..., T\}$ into $J$ smaller sub-sequences indexed by $\{\{1, ..., t_1\}, \{t_1 + 1, ..., t_2\}, ..., \{t_{J-1} + 1, ..., T\}\}$. For the first subsequence, the mean vector sequence $\mathbf{m}_{1:t_1,n}$ is initialized as the detected vector at the first frame $\mathbf{o}_{1k}$ repeated for $t_1$ times with a arbitrary order of assignment. Thus, there are as many tracked sources as initial detections, i.e., this implicitly sets $N = K_1$. The subsequence of source position vectors $\mathbf{s}_{1:t_1,n}$ is initialized with the same values as for the mean vector. Then, we run the MixDVAE algorithm on the first subsequence for $I_0$ iterations. Next, we initialize the mean vector sequence $\mathbf{m}_{t_1+1:t_2,n}$ of the second subsequence with $\mathbf{m}_{t_1n}$ repeated for $t_2 - t_1$ times (and the same for $\mathbf{s}_{t_1+1:t_2,n}$). And so on for the following subsequences. Finally, the initialized subsequences are concatenated together to form the initialized whole sequence. The pseudo-code of the cascade initialization can be found in Algorithm 2.

---

**Algorithm 2** Cascade initialization of the position vector sequence

---

**Input:**

    Detected bounding boxes at the first frame $\mathbf{o}_{1,1:K_1}$;

    Pre-trained DVAE parameters $\{\theta_{\mathbf{s}}, \theta_{\mathbf{z}}, \phi_{\mathbf{z}}\}$;

    Initialized observation model covariance matrices $\{\boldsymbol{\Phi}_{tk}^{(0)}\}_{t,k=1}^{T,K_t}$;

    Initialized covariance matrices $\{\mathbf{V}_{tn}^{(0)}\}_{t,n=1}^{T,N}$;

**Output:**

    Initialized mean position vector sequence $\{\mathbf{m}_{tn}^{(0)}\}_{t,n=1}^{T,N}$;

    Initialized sampled position vector sequence $\{\mathbf{s}_{tn}^{(0)}\}_{t,n=1}^{T,N}$;

1: Split the whole observation sequence $\mathbf{o}$ into $J$ sub-sequences indexed by $\{t_0 = 1, ..., t_1\}$, $\{t_1 + 1, ..., t_2\}$, ..., $\{t_{J-1} + 1, ..., t_J = T\}$;

2: **for** $j == 1$ **do**

3:     **for** $k \leftarrow 1$ to $K_1$ **do**

4:         $n \leftarrow k$;

5:         **for** $t \leftarrow 1$ to $t_1$ **do**

6:             $\mathbf{m}_{tn}^{(0)}, \mathbf{s}_{tn}^{(0)} = \mathbf{o}_{1k}$;

7:         **end for**

8:     **end for**

9: **end for**

10: **for** $j \leftarrow 2$ to $J$ **do**

11:     **for** $n \leftarrow 1$ to $N$ **do**

12:         **for** $t \leftarrow t_{j-1} + 1$ to $t_j$ **do**

13:             $\mathbf{m}_{tn}^{(0)}, \mathbf{s}_{tn}^{(0)} = \mathbf{m}_{t_{j-1}n}^{(I_0)}$;

14:         **end for**

15:         $\{\mathbf{m}_{tn}^{(I_0)}\}_{t=t_{j-1}+1}^{t_j} = \mathbf{MixDVAE}(I_0, \{\{\theta_{\mathbf{s}}, \theta_{\mathbf{z}}, \phi_{\mathbf{z}}\},$

16:         $\{\boldsymbol{\Phi}_{tk}^{(0)}, \mathbf{m}_{tn}^{(0)}, \mathbf{V}_{tn}^{(0)}, \mathbf{s}_{tn}^{(0)}\}_{t=t_{j-1}+1,k=1,n=1}^{t_j,K_t,N}\})$;

17:     **end for**

18: **end for**

19: $\mathbf{m}_{1:T,1:N}^{(0)} = \left[\mathbf{m}_{1:t_1,1:N}^{(0)}, ..., \mathbf{m}_{t_{J-1}+1:T,1:N}^{(0)}\right]$;

20: $\mathbf{s}_{1:T,1:N}^{(0)} = \left[\mathbf{s}_{1:t_1,1:N}^{(0)}, ..., \mathbf{s}_{t_{J-1}+1:T,1:N}^{(0)}\right]$;

---

## D SRNN implementation details

The SRNN generative model is implemented with a forward LSTM network, which embeds all the past information of the sequence $\mathbf{s}$. Then, a dense layer with the tanh activation function plus a linear layer provide the parameters $\boldsymbol{\mu}_{\theta_{\mathbf{s}}}, \boldsymbol{v}_{\theta_{\mathbf{s}}}$. Similarly, the parameters $\boldsymbol{\mu}_{\theta_{\mathbf{z}}}, \boldsymbol{v}_{\theta_{\mathbf{z}}}$ are computed with two dense layers with tanh activation function plus a linear layer appended to the LSTM as well. The inference model shares the hidden variables of the forward LSTM network of the generative model and uses two dense layers with the tanh activation function plus a linear layer to compute the parameters $\boldsymbol{\mu}_{\phi_{\mathbf{z}}}, \boldsymbol{v}_{\phi_{\mathbf{z}}}$.

In the MOT set-up, both $\mathbf{s}_t$ and $\mathbf{z}_t$ are of dimension 4. While in the SC-ASS set-up, $\mathbf{s}_t$ is of dimension 513 and $\mathbf{z}_t$ is of dimension 16. The SRNN generative distributions in the right-hand side of (36) are implemented as:

$$\mathbf{h}_t = d_h(\mathbf{s}_{-1}, \mathbf{h}_{-1}), \tag{66}$$

$$\left[\boldsymbol{\mu}_{\theta_{\mathbf{z}}}, \boldsymbol{v}_{\theta_{\mathbf{z}}}\right] = d_z(\mathbf{h}_t, \mathbf{z}_{t-1}), \tag{67}$$

$$p_{\theta_{\mathbf{z}}}(\mathbf{z}_t | \mathbf{s}_{1:t-1}, \mathbf{z}_{t-1}) = \mathcal{N}\left(\mathbf{z}_t; \boldsymbol{\mu}_{\theta_{\mathbf{z}}}, \mathrm{diag}(\boldsymbol{v}_{\theta_{\mathbf{z}}})\right), \tag{68}$$

$$\left[\boldsymbol{\mu}_{\theta_{\mathbf{s}}}, \boldsymbol{v}_{\theta_{\mathbf{s}}}\right] = d_s(\mathbf{h}_t, \mathbf{z}_t), \tag{69}$$

$$p_{\theta_{\mathbf{s}}}(\mathbf{s}_t | \mathbf{s}_{1:t-1}, \mathbf{z}_t) = \mathcal{N}\left(\mathbf{s}_t; \boldsymbol{\mu}_{\theta_{\mathbf{s}}}, \mathrm{diag}(\boldsymbol{v}_{\theta_{\mathbf{s}}})\right), \tag{70}$$

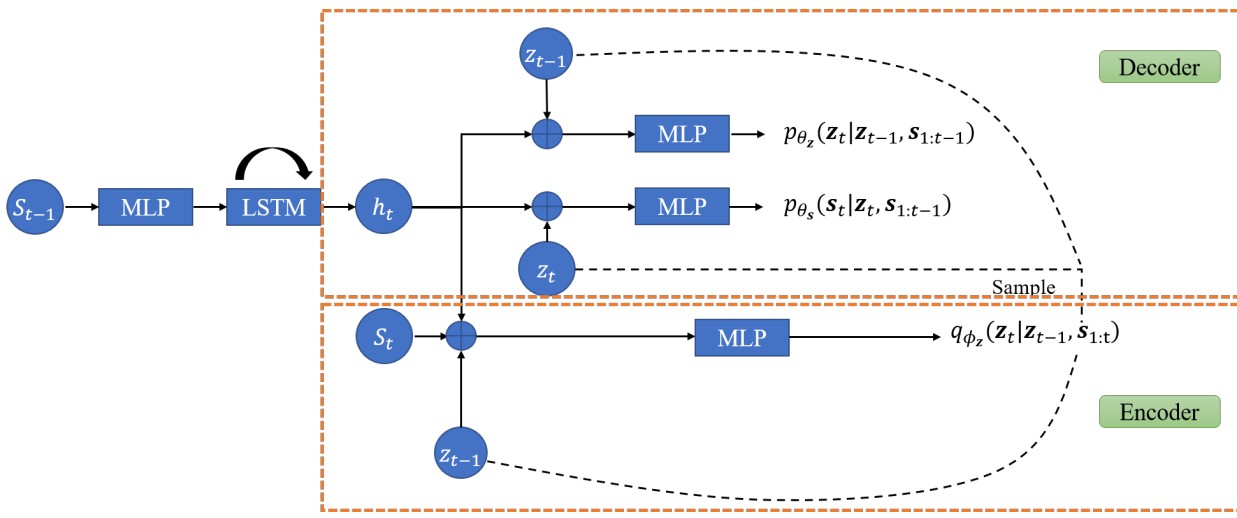

Figure 5: Schema of the SRNN model architecture. The "plus" symbol represents the concatenation of the input vectors.

where the function $d_h$ in (66) is implemented by a forward RNN and $\mathbf{h}_t$ denotes the RNN hidden state vector, the dimension of which is set to 8 for MOT and 128 for SC-ASS. In practice, LSTM networks are used. The function $d_s$ in (69) is implemented by a dense layer of dimension 16 for MOT and of dimension 256 for SC-ASS, with the tanh activation function, followed by a linear layer, which outputs are the parameters $\boldsymbol{\mu}_{\theta_\mathbf{s}}, \boldsymbol{v}_{\theta_\mathbf{s}}$. The function $d_z$ in (67) is implemented by two dense layers of dimension 8, 8 respectively for MOT and of dimension 64, 32 respectively for SC-ASS, with the tanh activation function, followed by a linear layer, which outputs are the parameters $\boldsymbol{\mu}_{\theta_\mathbf{z}}, \boldsymbol{v}_{\theta_\mathbf{z}}$.

The SRNN inference model in the right-hand side of (37) is implemented as:

$$\left[\boldsymbol{\mu}_{\phi_\mathbf{z}}, \boldsymbol{v}_{\phi_\mathbf{z}}\right] = e_\mathbf{z}(\mathbf{h}_t, \mathbf{s}_t, \mathbf{z}_{t-1}), \tag{71}$$

$$q_{\phi_\mathbf{z}}(\mathbf{z}_t|\mathbf{z}_{t-1}, \mathbf{s}_{1:t}) = \mathcal{N}\left(\mathbf{z}_t; \boldsymbol{\mu}_{\phi_\mathbf{z}}, \mathrm{diag}(\boldsymbol{v}_{\phi_\mathbf{z}})\right), \tag{72}$$

where the function $e_\mathbf{z}$ in (71) is implemented by two dense layers of dimension 16 and 8 respectively for MOT and of dimension 64 and 32 respectively for SC-ASS, with the tanh activation function, followed by a linear layer, which outputs are the parameters $\boldsymbol{\mu}_{\phi_\mathbf{z}}, \boldsymbol{v}_{\phi_\mathbf{z}}$.

The SRNN architecture is schematized in Figure 5. It can be noted that the RNN internal state $\mathbf{h}_t$ cumulating the information on $\mathbf{s}_{1:t-1}$ is shared by the encoder and the decoder, see (Girin et al., 2021, Chapter 4) for a discussion on this issue.

# E MOT dataset processing

## E.1 Synthetic trajectory dataset generation

To generate bounding boxes with reasonable size, we generate the coordinates of the top-left point (noted as $x_t^{\mathrm{L}}$ and $x_t^{\mathrm{T}}$) plus the height (noted as $a_t$) and width (noted as $b_t$) of the bounding boxes and deduce the coordinates of the bottom-right point. The width-height ratio is sampled randomly, and kept constant during the trajectory. While the trajectory of one coordinate is generated using piece-wise combinations of elementary functions, which are: static $a(t) = a_0$, constant velocity $a(t) = a_1 t + a_0$, constant acceleration $a(t) = a_2 t^2 + a_1 t + a_0$, and sinusoidal (allowing for circular trajectories) $a(t) = a \sin(\omega t + \phi_0)$. That is to say, we split the whole sequence into several segments, and each segment is dominated by a certain elementary

---

**Algorithm 3** Synthetic trajectories generation

---

**Input:**

    Total sequence length $T$;

    Maximum sub-sequence number $s_{max}$;

    Distribution parameters $\mu_{b_0}$, $\sigma_{b_0}$, $\mu_r$, $\sigma_r$, $\mu_{a_1}$, $\sigma_{a_1}$, $\mu_{a_2}$, $\sigma_{a_2}$, $\mu_\omega$, $\sigma_\omega$, $\mu_{\phi_0}$, $\sigma_{\phi_0}$;

    Discrete probability distribution of different elementary trajectory function types $p = [p_1, p_2, p_3, p_4]$;

**Output:**

    Synthetic bounding box position sequence $gen\_seq = \{(x_t^{\mathrm{L}}, x_t^{\mathrm{T}}, x_t^{\mathrm{R}}, x_t^{\mathrm{B}})\}_{t=1}^T$;

1: **function** GENSEQ($x_0$, $s$, $t_{split}$, $params\_prob$, $p$)
2:     $start = x_0$;
3:     **for** $i \leftarrow 0$ to $s$ **do**
4:         Sample $function\_type$ using $p$;
5:         Sample trajectory function parameters $params\_list$ using $params\_prob$;
6:         $t_i = t_{split}[i]$;
7:         $x\_sub_i = \textbf{GenTraj}(start, func\_type,$
8:         $params\_list)$;
9:         $start = x\_sub_i[t_i]$;
10:     **end for**
11:     $x = [x\_sub_0, ..., x\_sub_{s-1}]$;
12:     **return** x;
13: **end function**
14: Sample $x_0$, $y_0$ from $\mathcal{U}(0, 1)$;
15: Sample $b_0$ from $\log\mathcal{N}(\mu_{b_0}, \sigma_{b_0})$;
16: Sample $r_{ab}$ from $\mathcal{N}(\mu_r, \sigma_r)$;
17: Randomly sample $s$ in $\{0, ..., s_{max}\}$;
18: Randomly sample $t_{split} = \{t_0, ..., t_{s-1}\}$ in $\{1, ..., T\}$;
19: $x = \textbf{GenSeq}(x_0, s, t_{split}, params\_prob, p)$;
20: $y = \textbf{GenSeq}(y_0, s, t_{split}, params\_prob, p)$;
21: $b = \textbf{GenSeq}(w_0, s, t_{split}, params\_prob, p)$;
22: $a = b * r_{ab}$;
23: $gen\_seq = [x, y, x + b, y - a]$;

---

function. An example of a 3-segment combination could be:

$$a(t) = \begin{cases} a_0 & 1 \leq t < t_1, \\ a_2 t^2 + a_1 t + a_0' & t_1 \leq t < t_2, \\ a_3 \sin(\omega t + \phi_0) & t_1 \leq t \leq T, \end{cases} \tag{73}$$

where the segments length is sampled from some pre-defined distributions to generate reasonable and continuous trajectories. The number of segments $s$ is first uniformly sampled in the set $\{1, \ldots, s_{\max}\}$. We then sample $s$ segment lengths that sum up to $T$. This defines the segment boundaries $t_1, \ldots, t_{s-1}$. For each segment, one of the four elementary functions is randomly selected. The function parameters are sampled as follow: $a_1 \sim \mathcal{N}(\mu_{a_1}, \sigma_{a_1}^2)$, $a_2 \sim \mathcal{N}(\mu_{a_2}, \sigma_{a_2}^2)$, $\omega \sim \mathcal{N}(\mu_\omega, \sigma_\omega^2)$ and $\phi_0 \sim \mathcal{N}(\mu_{\phi_0}, \sigma_{\phi_0}^2)$. The two remaining parameters, $a_0$ and $a$, are set to the values needed to ensure continuous trajectories, thus initialising the trajectories at every segment, except for the first one. The very initial trajectory point is sampled randomly from $\mathcal{U}(0, 1)$. And the initial width is sampled from a log-normal distribution $b_0 \sim \log\mathcal{N}(\mu_{b_0}, \sigma_{b_0}^2)$. Finally, the ratio between the height and width is supposed to be constant with respect to time. It is sampled from a log-normal distribution $r_{ab} = \frac{a}{b} \sim \log\mathcal{N}(\mu_r, \sigma_r^2)$ and the height is obtained by multiplying the width and the ratio. More implementation details can be found in Algorithm 3.

In our experiments, the total sequence length of the generated trajectories for DVAE pre-training equals to $T = 60$ frames. And the maximum number of segments is set to $s_{max} = 3$. The parameters of the $a_1$, $a_2$, $\omega$, $\phi_0$, $w_0$, and $r_{hw}$ distributions are determined by estimating the statistical characteristics of publicly published detections of the MOT17 training dataset. More precisely, we estimated the empirical mean and

standard deviation of the speed and acceleration for all matched detection sequences (i.e., the first and second order differentiation of the position sequences).

### E.2 MOT17-3T dataset construction

To construct the MOT17-3T dataset, first, we matched the detected bounding boxes to the ground-truth bounding boxes using the Hungarian algorithm (Kuhn, 1955) and retained only the matched detected bounding boxes (i.e., the detected bounding boxes that were not matched to any ground-truth bounding boxes were discarded). The cost matrix were computed according to the the Intersection-over-Union (IoU) distance between bounding boxes. We split each complete video sequence into subsequences of length $T$ (three different values of $T$ are tested in our experiments, as detailed below) and only kept the tracks with a length no shorter than $T$. For each subsequence, we randomly chose three tracks that appeared in this subsequence from the beginning to the end. The detected bounding boxes of these three tracks form one test data sample. We have tested three values for the sequence length $T$ to evaluate its influence on the tracking performance of our algorithm: 60, 120, and 300 frames (respectively corresponding to 2, 4, and 10 seconds at 30 fps). Among the three public detection results provided with the MOT17 dataset, SDP has the best detection performance. So, we used the detection results of SDP to create our dataset.

## F  MOT baselines implementation details

ArTIST Saleh et al. (2021) is a probabilistic auto-regressive model which consists of two main blocks: MA-Net and the ArTIST model. MA-Net is a recurrent autoencoder that is trained to learn a representation of the dynamical interaction between all agents in the scene. ArTIST is an RNN that takes as input a 4D velocity vector of the current frame for one object as well as the corresponding 256-dimensional interaction representation learned by MA-Net, and outputs a probability distribution for each dimension of the motion velocity for the next frame. As indicated in Saleh et al. (2021), the models are trained on the MOT17 training set and the PathTrack Manen et al. (2017) dataset. We have reused the trained models as well as the tracklet scoring and inpainting code provided by the authors and reimplemented the object tracking part according to the paper, as this part was not provided. The tracklets are initialized with the bounding boxes detected in the first frame. For any time frame $t$, the score of assigning a detection $\mathbf{o}_{tk}$ to a tracklet $n$ is obtained by evaluating the likelihood of this detection under the distribution estimated by the ArTIST model. The final assignment is computed using the Hungarian algorithm. For any tentatively alive tracklet whose last assignment is prior to $t-1$ with a non-zero gap (implying that there exists a detection absence), the algorithm first performs tracklet inpainting to fill the gap up to $t-1$, then computes the assignment score with the inpainted tracklet. As described in Saleh et al. (2021), the inpainting is done with multinominal sampling, and a tracklet rejection scheme (TRS) is applied to select the best inpainted trajectory. In order to eliminate possible inpainting ambiguities, the Hungarian algorithm is run twice, once only for the full sequences without gaps and the second time for the inpainted sequences. The number of candidates for multinominal sampling is set to 50. For the TRS, the IoU threshold used in Saleh et al. (2021) is 0.5. In our test scenario, there are less tracklets and the risk of false negative is much greater than that of false positive. So, we decreased the threshold to 0.1, which provided better results than the original value.

## G  More MOT tracking examples

The first example plotted in Fig. 6 illustrates the case where two persons cross each other. This is one of the most complicated situations that may cause an identity switch and even lead to tracking loss. Considering the limited space for the figure, we display the bounding boxes every ten frames to view the whole process of crossing. For $t = 60$, when the ground-truth bounding boxes of Sources 2 and 3 ($s_2$ and $s_3$ in the figure) strongly overlap, Detection $o_2$ disappears. Again, ArTIST exhibits frequent identity switches. Besides, at $t = 20$, the estimated bounding box $m_1$ is totally overlapped with that of $m_3$. And at $t = 90$, the estimated bounding boxes for all of the three sources are getting very close to each other. This indicates that the identity switches can cause unreasonable trajectories estimation. For VKF, the observations for both Sources 2 and 3 are assigned to the same target $s_3$ all along the sequence, due to $s_2$ and $s_3$ being close

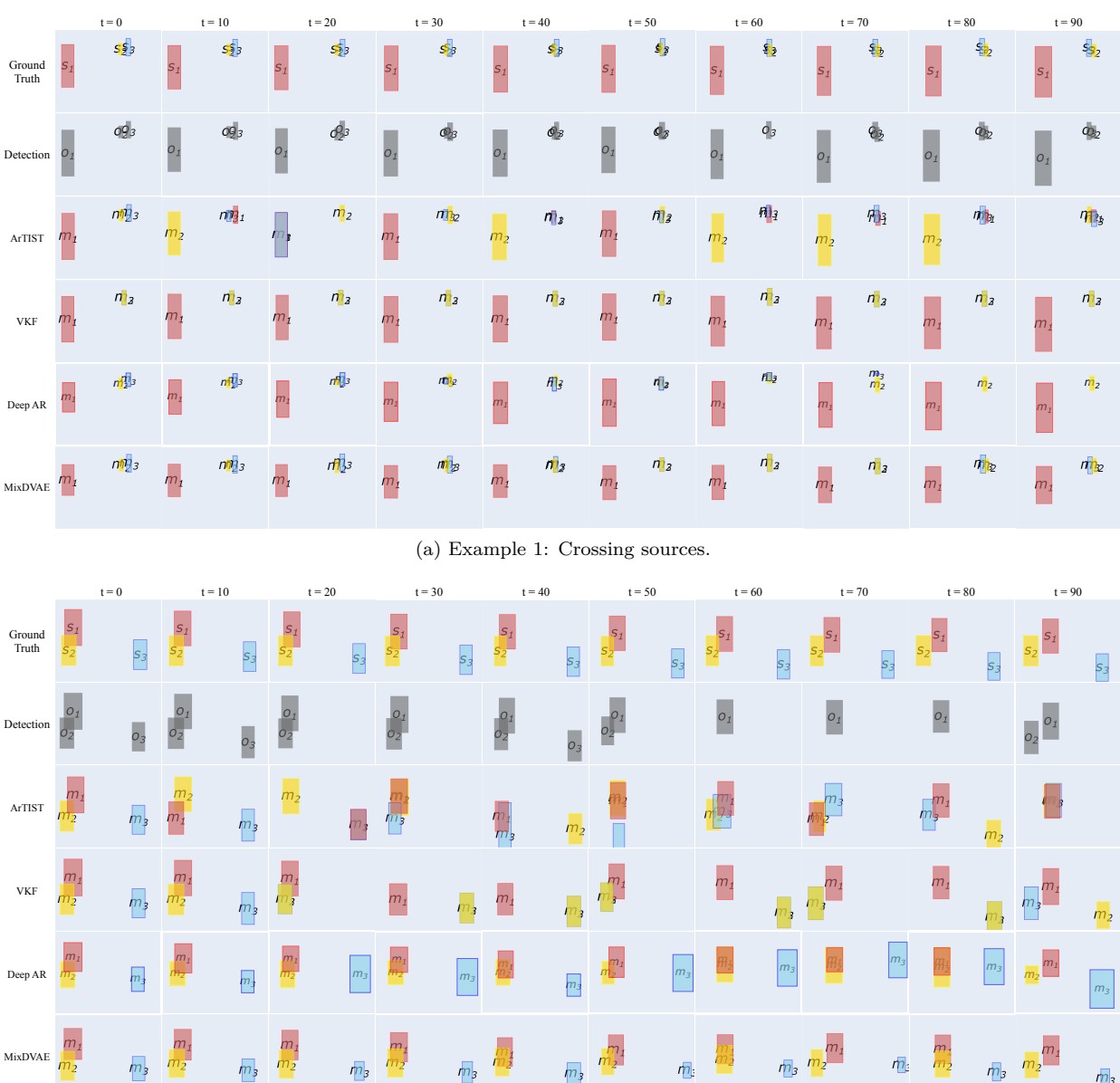

(a) Example 1: Crossing sources.

(b) Example 2: Crossing sources with frequent detection absence.

Figure 6: Examples of tracking result obtained with the proposed MixDVAE algorithm and the two baselines. For clarity of presentation, the simplified notations $s_1$, $o_1$, and $m_1$ denote the ground-truth source position, the observation, and the estimated position, respectively (for Source 1, and the same for the two other sources). Best seen in color.

to each other, so that the estimated bounding boxes $m_2$ and $m_3$ overlap completely. For the Deep AR, the estimation of $m_3$ becomes inaccurate from $t = 70$ and it disappears at $t = 80$ and $t = 90$ (the estimation is out of the frame). In contrast, MixDVAE displays a consistent tracking of the three sources. For $t = 50, 60$, and 70, the estimated bounding boxes $m_2$ and $m_3$ overlap due to the ground-truth bounding boxes $s_2$ and $s_3$ strongly overlap each other. However, the tracking is correctly resumed at $t = 80$, with no identity switch (i.e., the crossing of Sources 2 and 3 is correctly captured by the model).

The second example displayed in Fig. 6 is another more complicated situation with two sources very close to each other and frequent detection absence. At $t = 20$ when observation $o_3$ disappears, both ArTIST and

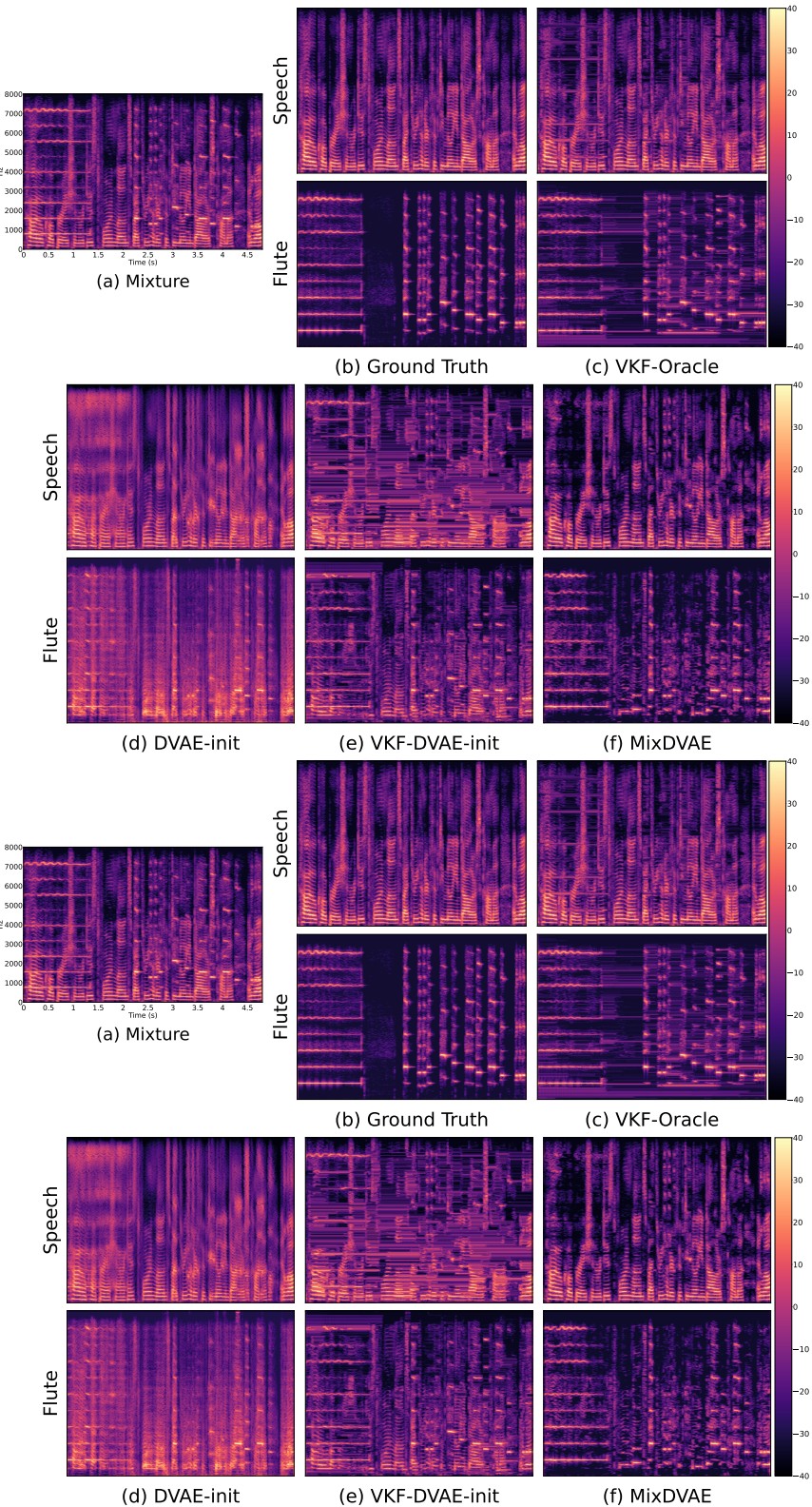

Figure 7: Examples of audio source separation results obtained with the proposed MixDVAE algorithm and the baselines. Best seen in color.

Table 4: Results obtained by MixDVAE on MOT17-3T (short sequences subset) for different values of $r_\Phi$. The values on the left (resp. right) side of the slashes are obtained without (resp. with) the fine-tuning of SRNN in the E-Z Step.

| $r_\Phi$ | MOTA↑ | MOTP↑ | IDF1↑ | #IDs↓ | %IDs↓ | MT↑ | ML↓ | #FP↓ | %FP↓ | #FN↓ | %FN↓ |
|---|---|---|---|---|---|---|---|---|---|---|---|
| 0.01 | 35.9/32.8 | **84.5**/**84.8** | 66.6/65.5 | **4914**/3216 | **1.6**/1.0 | 2946/2714 | 916/913 | 96438/102062 | 31.3/33.1 | 96438/102062 | 31.3/33.1 |
| 0.02 | 65.5/61.8 | 84.2/84.7 | 81.3/79.8 | 5319/3073 | 1.7/1.0 | 3932/3652 | 407/379 | 50596/57291 | 16.4/18.6 | 50596/57291 | 16.4/18.6 |
| 0.03 | 74.9/70.0 | 83.1/84.3 | 86.1/84.4 | 5088/**2853** | 1.7/**0.9** | 4232/3931 | 158/160 | 36165/43777 | 11.7/14.2 | 36165/43777 | 11.7/14.2 |
| 0.04 | **79.1**/75.1 | 81.3/83.5 | **88.4**/86.7 | 4966/2862 | 1.6/0.9 | **4370**/4067 | 50/64 | **29808**/36990 | **9.7**/11.9 | **29808**/36990 | **9.7**/11.9 |
| 0.05 | 76.4/**75.6** | 79.2/82.6 | 87.1/**87.1** | 4982/2919 | 1.6/0.9 | 4268/**4066** | **42**/53 | 33924/**36088** | 11.0/**11.7** | 33924/**36088** | 11.0/**11.7** |
| 0.06 | 69.2/70.1 | 76.9/82.0 | 83.5/84.4 | 5297/3005 | 1.7/1.0 | 3978/3845 | 73/137 | 44793/44598 | 14.5/14.5 | 44793/44598 | 14.5/14.5 |
| 0.07 | 59.8/66.8 | 74.8/80.3 | 78.9/82.9 | 5146/3000 | 1.7/1.0 | 3688/3775 | 188/285 | 59348/49646 | 19.2/16.1 | 59348/49646 | 19.2/16.1 |
| 0.08 | 48.5/60.6 | 73.1/79.4 | 73.3/79.9 | 5097/3119 | 1.7/1.0 | 3303/3637 | 337/432 | 76865/59220 | 24.9/19.2 | 76865/59220 | 24.9/19.2 |

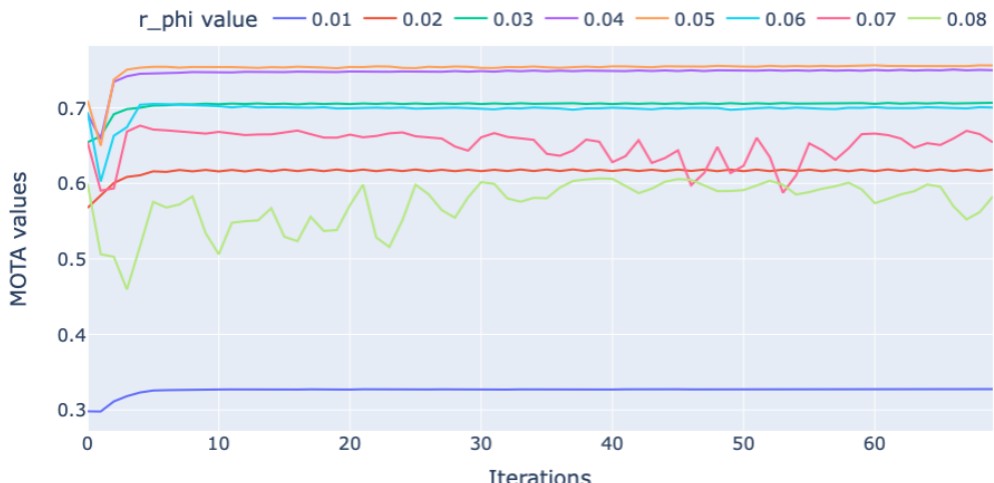

Figure 8: MOTA score obtained by MixDVAE as a function of the number of VEM iterations, for different values of $r_\Phi$.

VKF lose one of the tracks, whereas MixDVAE keeps a reasonable tracking of the three tracks. From $t = 60$ to 80, both $o_2$ and $o_3$ are absent. The tracks inpainted by ArTIST are not consistent anymore and VKF still misses one track. The estimations of Deep AR are inaccurate when the detections are absent. However, even in this difficult scenario, MixDVAE keeps on providing three reasonable trajectories.

## H  More SC-ASS examples

In Figure 7 we plot two other SC-ASS examples.

## I  Ablation study

In order to better understand the MixDVAE model, we conducted ablation studies on the influence of the pre-trained DVAE model quality, the influence of fine-tuning the DVAE, and the influence of the observation variation matrix ratio $r_\Phi$.

### I.1  Influence of the pre-trained DVAE model quality

We have conducted an ablation study on the influence of the pre-trained DVAE model quality on the whole MixDVAE algorithm, for both the MOT task and the SC-ASS task. Specifically, we have pre-trained the DVAE model at different data scales and tested the performance of MixDVAE using these different pre-trained models.

Table 5: Capacity of the SRNN model pre-trained at three data scales of the synthetic trajectories dataset. SRNN-full, SRNN-half, and SRNN-quarter stand for SRNN pre-trained on the totality, half of and quarter of our original training set, respectively.

| Model name | Training loss | Validation loss |
|---|---|---|
| SRNN-full | -40.77 | -40.15 |
| SRNN-half | -39.36 | -38.86 |
| SRNN-quarter | -36.55 | -35.27 |

Table 6: MOT results obtained by MixDVAE with SRNN pre-trained at the three data scales. The results are reported for the short sequence test subset ($T = 60$ frames).

| Model name | MOTA↑ | MOTP↑ | IDF1↑ | #IDs↓ | %IDs↓ | MT↑ | ML↓ | #FP↓ | %FP↓ | #FN↓ | %FN↓ |
|---|---|---|---|---|---|---|---|---|---|---|---|
| SRNN-full | **79.1** | 81.3 | **88.4** | **4966** | **1.6** | **4370** | 50 | **29808** | 9.7 | **29808** | 9.7 |
| SRNN-half | 74.7 | **84.4** | 86.6 | 5624 | 1.8 | 4039 | 94 | 38153 | 12.4 | 38153 | 12.4 |
| SRNN-quarter | 75.2 | **84.4** | 86.9 | 5598 | 1.8 | 4040 | 91 | 37443 | 12.2 | 37443 | 12.2 |

Table 7: Capacity of the SRNN model pre-trained at three data scales of the WSJ0 and the CBF datasets. SRNN-full, SRNN-half, and SRNN-quarter stand for SRNN pre-trained on the totality, half of and quarter of our original training set, respectively.

| Model name | WSJ0 | | CBF | |
|---|---|---|---|---|
| | Training loss | Validation loss | Training loss | Validation loss |
| SRNN-full | 353.89 | 373.61 | 521.76 | 779.69 |
| SRNN-half | 358.13 | 389.58 | 489.53 | 949.11 |
| SRNN-quarter | 361.58 | 383.64 | 646.55 | 1106.27 |

**MOT task.** We conducted pre-training of the SRNN model on three separate datasets with different scales: the full synthetic trajectories training set used in Section 5, consisting of 12,105 trajectories, a dataset containing half of these synthetic trajectories, randomly selected (6,052 trajectories), and another dataset with a quarter of these synthetic trajectories, randomly selected (3,026 trajectories). We use the ELBO loss to represent the quality of the resulting pre-trained SRNN models. The ELBO loss values are reported in Table 5. As expected, we observe that by decreasing the training data size, the performance of the SRNN model drops (with higher training and validation loss).

We run the MixDVAE inference algorithm with the three pre-trained SRNN models on the short sequence test subset ($T = 60$ frames), and the obtained results are reported in Table 6. We can see that the overall performance of the MixDVAE algorithm with SRNN-half and SRNN-quarter drops compared to that with SRNN-full, but this drop is relatively limited, at least for some of the metrics, including the key MOTA metric. Moreover, the difference between the performance of MixDVAE with SRNN-half and with SRNN-quarter is quite small. Therefore, even if it is hard to draw a general conclusion from a single experiment with three dataset sizes, this seems to indicate some robustness of MixDVAE w.r.t. the DVAE training dataset size, and confirm its interest as a data-frugal weakly supervised method (here for the MOT application).

**SC-ASS task**. Similar to the MOT task, we also generated two additional subsets of the training data for both the WSJ0 and CBF datasets, comprising half and a quarter of our original training dataset (used in Section 6), randomly selected. The two new subsets of WSJ0 contains 12.45 and 6.29 hours of speech recordings respectively. And the two new subsets of CBF contains 1.07 and 0.55 hours of CBF recordings respectively. The performance of the SRNN model pre-trained on these different datasets is reported in Table 7. For the WSJ0 dataset, we observe that the training and validation losses are relatively close to each other, and both increase when decreasing the training data size, but the increase is moderate. Therefore,

Table 8: SC-ASS results obtained by MixDVAE with SRNN pre-trained at the three data scales. The results are reported for the short sequence test subset ($T = 50$).

| Model name | Speech | | | Chinese bamboo flute | | |
|---|---|---|---|---|---|---|
| | RMSE ↓ | SI-SDR ↑ | PESQ ↑ | RMSE ↓ | SI-SDR ↑ | PESQ ↑ |
| SRNN-full | **0.006** | 9.23 | 1.73 | **0.007** | **13.50** | **2.30** |
| SRNN-half | **0.006** | **9.66** | **1.82** | 0.009 | 12.29 | 2.28 |
| SRNN-quarter | 0.007 | 8.83 | 1.79 | 0.011 | 10.29 | 2.13 |

Table 9: MOT results obtained by MixDVAE with and without the fine-tuning of SRNN. The results are reported for the short, medium and long sequence test subsets ($T = 60$, 120, and 300 frames, respectively).

| Dataset | Fine-tuning | MOTA↑ | MOTP↑ | IDF1↑ | #IDs↓ | %IDs↓ | MT↑ | ML↓ | #FP↓ | %FP↓ | #FN↓ | %FN↓ |
|---|---|---|---|---|---|---|---|---|---|---|---|---|
| Short | Yes | 75.1 | **83.5** | 86.7 | **2862** | **0.9** | 4067 | 64 | 36990 | 11.9 | 36990 | 11.9 |
| | No | **79.1** | 81.3 | **88.4** | 4966 | 1.6 | **4370** | **50** | **29808** | **9.7** | **29808** | **9.7** |
| Medium | Yes | 73.1 | **84.0** | 85.9 | **3044** | **0.7** | 2705 | 136 | 54604 | 13.1 | 54604 | 13.1 |
| | No | **78.6** | 82.2 | **88.0** | 6107 | 1.5 | **2907** | **120** | **41747** | **9.9** | **41747** | **9.9** |
| Long | Yes | 65.6 | **84.9** | 81.6 | **8670** | **0.8** | 2286 | 67 | 171515 | 13.8 | 171515 | 13.8 |
| | No | **83.2** | 82.4 | **90.0** | 23081 | 2.3 | **2890** | **12** | **74550** | **7.3** | **74550** | **7.3** |

the capacity of SRNN drops, but quite slightly. However, for the CBF dataset, the gap between the training and validation losses is higher, and the training loss of SRNN-half decreases compared to SRNN-full while the validation loss increases significantly, increasing the gap. Both the training loss and the validation loss of SRNN-quarter are higher than that of SRNN-full and SRNN-half, and the gap between training and validation is also relatively large. This shows that the size of the (full) CBF dataset may be a bit too limited, and reducing this dataset may harm the generalization capacity of SRNN.

The SC-ASS results obtained by MixDVAE with SRNN pre-trained at the three different data scales are reported in Table 8. The experiments are conducted on the short sequence subset ($T = 50$). We find that, surprisingly, the separation performance of MixDVAE with SRNN-half on the speech signals has been slightly improved over SRNN-full, whereas (much less surprisingly) the performance on the CBF signals has decreased. This may be caused by the lower generalization ability of SRNN-half on the CBF dataset. For SRNN-quarter, the performance of MixDVAE on both the speech and the CBF decrease, but the decrease for the speech is quite moderate (0.4 dB SI-SDR w.r.t. SRNN-full; the PESQ value is even slightly better), whereas the CBF is loosing about 3.2 dB SI-SDR. Again, even if it is difficult to draw a general conclusion from this single experiment, those results seem to indicate a relative robustness of MixDVAE to the limitation of the DVAE training dataset size, provided that the DVAE keeps a sufficient generalization capability.

### I.2 Influence of the DVAE fine-tuning

As mentioned in Section 4.2, the DVAE model can either be fine-tuned or not in the MixDVAE algorithm. We have studied the effect of fine-tuning SRNN on both MOT and SC-ASS tasks.

**MOT task.** Table 9 shows the MOT scores obtained by MixDVAE on the three test subsets with and without the fine-tuning of SRNN in the E-Z step. We observe that for all three datasets, not fine-tuning the DVAE model leads to the best overall performance (as measured by MOTA in particular). Though fine-tuning the DVAE model can indeed increase the MOTP score and decrease the number of identity switches, it does not improve the overall tracking performance. Indeed, fine-tuning increases the FP and FN numbers/proportions, and thus decreases the MOTA scores. Especially on the long sequence dataset, the MOTA score drops from 83.2 to 65.6.

**SC-ASS task.** Table 10 shows the performance of MixDVAE on the three test subsets with and without fine-tuning SRNN in the E-Z step. Similar to the MOT task, for all three datasets, not fine-tuning SRNN leads to the best overall source separation performance (on all of the evaluation metrics).

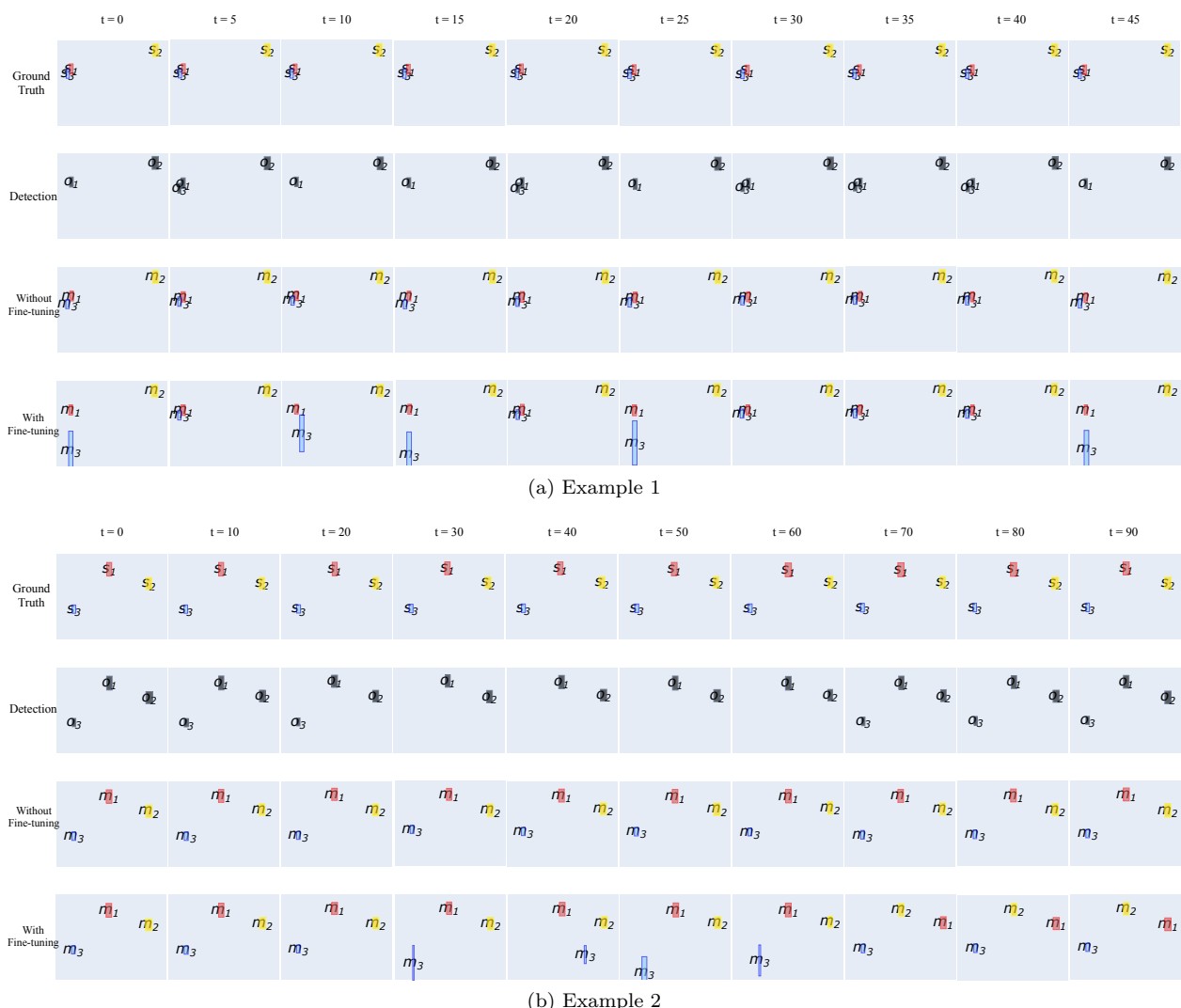

(a) Example 1

(b) Example 2

Figure 9: Two examples of tracking result obtained with the proposed MixDVAE algorithm, with and without fine-tuning during the E-Z step. For clarity of presentation, the simplified notations $s_1$, $o_1$, and $m_1$ denote the ground-truth source position, the observation, and the estimated position, respectively (for Source 1, and the same for the two other sources). Best seen in color.

Table 10: SC-ASS results obtained by MixDVAE with and without the fine-tuning of SRNN. The results are reported for the short ($T = 50$), medium ($T = 100$) and long ($T = 300$) test sequence subsets.

| Dataset | Finetuning | Speech | | | Chinese bamboo flute | | |
|---|---|---|---|---|---|---|---|
| | | RMSE ↓ | SI-SDR ↑ | PESQ ↑ | RMSE ↓ | SI-SDR ↑ | PESQ ↑ |
| Short | Yes | 0.007 | 8.00 | 1.63 | 0.007 | 12.73 | 2.15 |
| | No | **0.006** | **9.23** | **1.73** | **0.007** | **13.50** | **2.30** |
| Medium | Yes | 0.008 | 8.00 | 1.55 | 0.008 | 12.23 | 2.02 |
| | No | **0.007** | **9.32** | **1.65** | **0.007** | **13.05** | **2.16** |
| Long | Yes | 0.008 | 7.02 | 1.49 | 0.008 | 11.40 | 1.88 |
| | No | **0.007** | **9.06** | **1.64** | **0.007** | **12.92** | **2.06** |

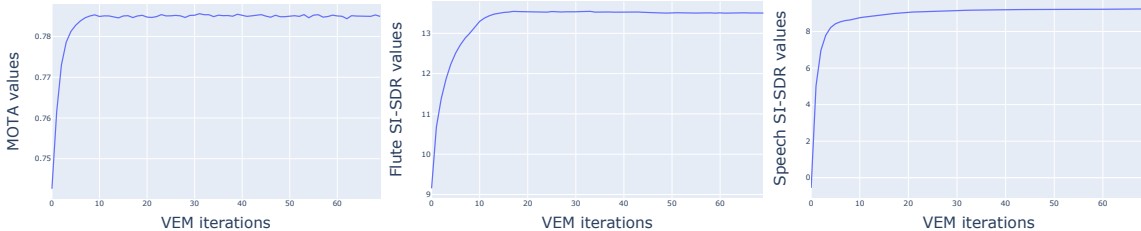

Figure 10: Evolution of the performance of MixDVAE as a function of the number of VEM iterations (MOTA score for the MOT task and SI-SDR scores for the SC-ASS task).

We therefore observe that for both tasks, fine-tuning the DVAE model results in performance degradation. The possible reason is that fine-tuning could make the model more sensible to observation noise, and lead to a generative model with worse performance. To verify this conjecture and to better understand the effect of fine-tuning, we have plotted in Figure 9 two examples for the MOT task, extracted from the long sequence test subset ($T = 300$ frames). To make possible the display of a long sequence in a limited space, the first example is plotted every 5 frames, whereas the second example is plotted every 10 frames. In Example 1, we observe that the detection for source $\mathbf{s}_3$ is missed for $t = 0$, $t = 10$, $t = 15$, $t = 25$ and $t = 45$. At these frames, MixDVAE without SRNN fine-tuning can still make a good estimation of $\mathbf{s}_3$'s position, whereas MixDVAE with SRNN fine-tuning can not make an accurate prediction. In the latter case, this caused a large error between the estimated source position and the ground truth. We can see a similar phenomena in Example 2. At frame $t = 30$, $t = 40$, $t = 50$ and $t = 60$, when the detection bounding box for source $\mathbf{s}_3$ is absent, the estimation obtained by MixDVAE with SRNN fine-tuning is bad (it is particularly bad for $t = 40$). This phenomenon confirms our conjecture that the observation noise, particularly the lack of observations, can introduce unforeseen effects during fine-tuning, resulting in a model with degraded performance.

### I.3  Influence of the observation variance ratio

Table 4 reports the MOT scores obtained with MixDVAE as a function of $r_{\mathbf{\Phi}}$. These experiments are conducted on the subset of short sequences. We report the results for both with and without fine-tuning SRNN in the E-Z step. Apart from the value of $r_{\mathbf{\Phi}}$ and the fine-tuning option, all other conditions are exactly the same across experiments. Table 4 shows that, whether fine-tuning SRNN in the E-Z step or not, the MOT scores first globally increase with $r_{\mathbf{\Phi}}$,[12] reach their optimal values for $r_{\mathbf{\Phi}} = 0.04$ or $0.05$ (for most metrics), and then decrease for greater $r_{\mathbf{\Phi}}$ values. For confirmation, we have also computed the averaged empirical ratio $\hat{r}_{\mathbf{\Phi}}$ of the detected bounding boxes (with the SDP detector), which is calculated as $\frac{1}{4T} \sum_{t=1}^{T} \frac{1}{K_t} \sum_{k=1}^{K_t} \left( \frac{|s_{tk}^{\mathrm{L}} - o_{tk}^{\mathrm{L}}|}{o_{tk}^{\mathrm{R}} - o_{tk}^{\mathrm{L}}} + \frac{|s_{tk}^{\mathrm{T}} - o_{tk}^{\mathrm{T}}|}{o_{tk}^{\mathrm{T}} - o_{tk}^{\mathrm{B}}} + \frac{|s_{tk}^{\mathrm{R}} - o_{tk}^{\mathrm{R}}|}{o_{tk}^{\mathrm{R}} - o_{tk}^{\mathrm{L}}} + \frac{|s_{tk}^{\mathrm{B}} - o_{tk}^{\mathrm{B}}|}{o_{tk}^{\mathrm{T}} - o_{tk}^{\mathrm{B}}} \right)$.[13] This value equals to $0.053$, $0.053$ and $0.047$ respectively for the short, medium and long sequence dataset. These values, which are close to each other because we used the same detector, correspond well to the $r_{\mathbf{\Phi}}$ value for the best performing model in Table 4. We can conclude that the model has better performance if the value of $r_{\mathbf{\Phi}}$ corresponds (empirically) to the detector performance. Besides, we have also observed that the value of $r_{\mathbf{\Phi}}$ has an impact on the convergence of the MixDVAE algorithm. Fig. 8 displays the MOTA score as a function of the number of MixDVAE iterations (here with the fine-tuning of the DVAE model). It appears clearly that for too high values of $r_{\mathbf{\Phi}}$, the model exhibits a lower and more hectic performance than for the optimal value.

---

[12]Except for the MOTP score, which continually decreases with the increase of $r_{\mathbf{\Phi}}$. This can be explained as follows. MOTP measures the precision of the position estimation for the matched bounding boxes. The estimated position $\mathbf{m}_{tn}$ in (28) is a weighted combination of the observation and the DVAE prediction. When $\mathbf{\Phi}_{tk}$ increases, the contribution of the observation decreases and $\mathbf{m}_{tn}$ is closer to the DVAE prediction. Since the error of the DVAE prediction may accumulate over time, this finally decreases the position estimation accuracy.

[13]Note that here $s_{tk}$ denotes the position of the target matched with the observation $o_{tk}$ at time frame $t$. We omit the target positions that are not matched with any observation.

Table 11: Averaged processing time per sequence for the MOT task.

| Sequence length (frames) | 60 | | 120 | | 300 | |
|---|---|---|---|---|---|---|
| # sources | 3 | 6 | 3 | 6 | 3 | 6 |
| Computation time per sequence (s) | 23.01 | 57.29 | 45.05 | 110.41 | 112.93 | 272.94 |

Table 12: Pre-training computational cost on different datasets at different scales.

| Task | Data set | Data scale | One epoch training time (s) |
|---|---|---|---|
| MOT | Synthetic trajectories | Full | 15 |
| | | Half | 7.8 |
| | | Quarter | 4.8 |
| SC-ASS | WSJ0 | Fall | 190.8 |
| | | Half | 121.2 |
| | | Quarter | 63 |

## J   Discussion on the computational complexity

The proposed method is based on two parts: (i) the pre-training of a DVAE model on a single-source dataset, and (ii) the MixDVAE variational EM algorithm for source tracking. The computational cost for the pre-training stage mainly depends on the data type and data size of the single-trajectory dataset. To give a general idea, we measured the average training time required for a single epoch (iteration over the whole training set) on both the synthetic trajectories dataset for MOT and the WSJ0 dataset for SC-ASS. The measurement is conducted on an NVIDIA Quadro RTX 8000, in a machine with an Intel(R) Xeon(R) Gold 5218R CPU @ 2.10GHz. The obtained results for different data scales as mentioned in Section I.1 is reported in Table 12. We have observed that doubling the size of the training data results in almost a doubling of the training time. On the other hand, the computation complexity of the MixDVAE algorithm mainly depends on three factors: the number of VEM iterations, the number of sources to track and separate, and the sequence length. Typically, the performance of MixDVAE exhibits an initial rapid increase over the VEM iterations, followed by stabilization towards a plateau. In Figure 10, we plot the evolution of the averaged performance of MixDVAE over the medium sequence test dataset as a function of the number of VEM iteration (the performance is represented by the MOTA score for the MOT task and by the SI-SDR score for the SC-ASS task). We observe that for the MOT task, the performance of MixDVAE has been stabilized from around 10 iterations, whereas for the SC-ASS task, the performance has been stabilized from around 20 iterations. In practice, we run the algorithm for more iterations to guarantee the convergence. Taking computational time optimization into account, it is possible to identify an optimal number of iterations by applying a grid search, for a specific task and dataset. To quantify the computational time of the MixDVAE algorithm, we compute the averaged processing time for one sequence on the MOT task, for the three considered values of the sequence length, and for the case of 3 and 6 sources. This average processing time is measured on an NVIDIA Quadro RTX 4000 GPU, in a machine with an Intel(R) Xeon(R) W-2145 CPU@3.70GHz, and it is averaged on 10 test sequences. The results are reported in Table 11. We observe a linear increase of the computation time as a function of the sequence length. As the number of sources to track doubles, the computation time exhibits more than a twofold increase. The computation complexity can be a bottleneck for the MixDVAE especially for long sequences with a large number of sources. However, further algorithm and code optimization might be possible, since we did not focus on this aspect of the problem so far.

