# OpenReview forum: "Mixture of Dynamical Variational Autoencoders for Multi-Source Trajectory Modeling and Separation"
_TMLR — Accepted by TMLR_

### Review · Reviewer_Bi6u · 2023-07-23

**Summary Of Contributions:**

In this paper, the authors propose a multiple source variational encoder model. In this proposed model, each source is assigned in it's own VAE, and there is a pretraining step for each source separately, which if I understand correctly restricts the model to the specific type of source it was trained on.

The authors propose an EM type algorithm for learning the model parameters. The evaluation is done on multi-source tracking and audio/speech source separation.



**Audience:**

Yes

**Broader Impact Concerns:**

Nothing very obvious stands out to me to point out as a concern.

**Claims And Evidence:**

Yes

**Requested Changes:**

- It would be good to compare with state-of-the-art speech/audio separation models. This can include supervised source separation models such as Sepformer, or unsupervised models such as Mixit. At its current state, I don't understand the benefit of using the proposed method in source separation. For generative source separation, you can also compare with vanilla NMF, or it's time dependent versions.
- Overall, the contributions are not very clear to me. It would be good to include an itemized contributions list at the end of section 1.2.
- If possible you can shorten the paper. It would be better to highlight the advantages of the approach rather than focusing on the technical details.

**Strengths And Weaknesses:**

Strengths:
- The paper describes the learning and the inference algorithm in great detail.
- Experimental results indicate slight performance improvement over the other methods that the authors compare against. Though I am not sure if this comparison is exhaustive / extensive.

Weaknesses
- I am not an expert on multi-target object tracking. But on audio/speech source separation the proposed algorithm seems rather restrictive. The proposed method requires pretraining on specific source types (similar to generative source separation with NMF), which limits the general applicability of the method. These days we are able to train source separation methods in a source-indepent way using techniques such permutation invariant loss, or embedding methods such as Deep=clustering based source separation. And, even if the models are trained with source independent methods on speech separation it is possible to obtain SI-SNR values above 20dB. Given this state-of-the-art I am not sure what is the benefit of the proposed approach on speech separation.
- In absolute sense the source separation results do not seem very good. (Below 2.00 PESQ, 10dB for speech)
- The paper seems very long and verbose. I might be possible to compress it and make it more to-the-point.
- I am not very sure about how novel the proposed model is.

---

> ### Author Response · Authors · 2023-09-06
> **Response to Reviewer Bi6u (Part 1)**
>
> We thank the reviewer for their feedback on our paper. In a general manner, we are glad that the reviewer acknowledged our comprehensive description of the method and the demonstrated performance improvement over the baselines. However, we were a bit disappointed to see that the the reviewer mainly focused their review on the audio source separation application (and in particular on the relatively modest performance of our model in this application) and, in a way, poorly considered the core of the paper, which is a 'generic' methodological contribution that can be applied to a vast number of applications. Below, we address the specific concerns raised by the reviewer.
>
> **"In this paper, the authors propose a multiple source variational encoder model. In this proposed model, each source is assigned in its own VAE, and there is a pretraining step for each source separately, which if I understand correctly restricts the model to the specific type of source it was trained on." and later in the review: ``The proposed method requires pretraining on specific source types (similar to generative source separation with NMF), which limits the general applicability of the method."**
>
> First, the reviewer's formulation "each source is assigned in its own VAE, and there is a pretraining step for each source separately" is a bit misled. In fact, we propose a latent-variable generative approach to model the dynamics of a system composed of multiple sources. The dynamics of a given type of source are modeled via a dynamical variational auto-encoder (DVAE), and, as we explain in Section 1.2, there is one single DVAE model for each type of source, with different instances of this DVAE model for the different sources of this type. The assignment is not from source to its own DVAE, but from observations to sources. This assignment is not intrinsic or straightforward; rather, it is achieved by introducing a discrete assignment random variable, whose posterior distribution is estimated using the proposed variational expectation-maximization algorithm. Also, there is not "a pretraining step for each source" but a pretraining step for each source type. Nevertheless, the reviewer is right when he says that the model restricts to the type of source it was trained on, as opposed to source-independent methods. We agree that for the audio source separation task, pre-training a DVAE for each source type may limit the applicability of the proposed method to certain scenarios. However, of course, this does not prevent mixing different types of sources or applying the same pre-trained DVAE to all sources as in the multi-object tracking application.

---

> ### Author Response · Authors · 2023-09-06
> **Response to Reviewer Bi6u (Part 2)**
>
> **"These days we are able to train source separation methods in a source-independent way using techniques such permutation invariant loss, or embedding methods such as Deep-clustering based source separation. And, even if the models are trained with source independent methods on speech separation it is possible to obtain SI-SNR values above 20dB. Given this state-of-the-art I am not sure what is the benefit of the proposed approach on speech separation.
> In absolute sense the source separation results do not seem very good. (Below 2.00 PESQ, 10dB for speech)."**
>
> One major benefit of the proposed method is that, as explained in Section 1.2, it can be considered as data-frugal and weakly supervised since it only uses one or several single-source(-type) dataset(s) of moderate size for training, in contrast to supervised state-of-the-art models, which use massive labeled datasets of mixtures of simultaneous sources (in the case of audio source separation, a massive dataset of parallel/aligned mixture and single-source samples). Labeled multi-source datasets used by supervised (source-independent) methods must be much larger than single-source datasets, since the mixture process intrinsically multiplies the content diversity, and can be very costly and difficult to obtain. We have slightly modified/enriched the third paragraph of Section 1.2 to make all this even more clear. Because they do not use the same kind and amount of data, we believe it is unfair, and thus irrelevant, to directly compare the performance of the proposed method with that of the state-of-the-art supervised speech/audio separation methods. We mention this point in Section 6.2/Baselines. Please, note that we acknowledge in Section 1.4 that the performance of SotA supervised audio source separation models are impressive. We would like to make clear that the goal of applying MixDVAE to audio source separation is not to compete with \textcolor{red}{supervised} SotA methods on this task, but it is rather to illustrate its versatility, i.e.~deliver a proof-of-concept for its applicability to tasks that are as diverse as MOT in computer vision and audio source separation (honestly, we cannot find much models in the literature that can do that).
> Finally, still regarding the relatively modest performance of MixDVAE in the presented audio source separation experiments, we would like to report a sentence from the 'Acceptance criteria' section of the TMLR website: "it should not be used as a reason to reject work that isn't considered “significant” or “impactful” because it isn't achieving a new state-of-the-art on some benchmark."
>
> **"The paper seems very long and verbose. I might be possible to compress it and make it more to-the-point."**
>
> We agree that the paper is long and we are sorry that the reviewer finds it verbose. We have tried to correctly position the paper w.r.t. the current literature and provide the necessary technical details to understand in depth the proposed MixDVAE model and algorithm. We would like to point out that, in essence, a VEM algorithm is `long' to describe, since there is an E-step for each latent variable of the model. In Section 4, we have tried to report only the essential steps and results for the different E-steps (and also for the M-step), and give the details of the derivations in Appendix A. We have also placed many other technical details in the other appendixes. Therefore, we believe we have already made a substantial effort to limit the size of the paper and make the technical content relatively 'to-the-point'.
>
> **"I am not very sure about how novel the proposed model is."**
>
> We address the difficult problem of multi-source trajectory modeling and separation. Our contribution, described in Section 1.2, is essentially a combination of a deep latent-variable sequential model with a discrete latent-variable  assignment model in a variational inference framework. To the best of our knowledge, this is the first time such a combination is proposed, together with corresponding efficient learning and inference algorithm. However, if the reviewer has particular research papers in mind that appear to be close to the proposed MixDVAE model, we will be happy to include them in the discussion and evaluate the novelty of MixDVAE w.r.t. these articles.

---

> ### Author Response · Authors · 2023-09-06
> **Response to Reviewer Bi6u (Part 3)**
>
> **"It would be good to compare with state-of-the-art speech/audio separation models. This can include supervised source separation models such as Sepformer, or unsupervised models such as Mixit. At its current state, I don't understand the benefit of using the proposed method in source separation. For generative source separation, you can also compare with vanilla NMF, or it's time dependent versions."**
>
> As stated above, we find it unfair to compare the proposed weakly supervised method with the state-of-the-art supervised speech/audio separation models, which are trained on massive datasets of parallel mixtures and isolated sources. For the benefit of using the proposed method, please see our answer to the second remark above. That being said, we agree that it is interesting to compare our method with unsupervised methods and with other weakly-supervised methods using single-source datasets only. Therefore, following the suggestion of the reviewer, in the revised version of the paper, we have updated our experimental study by comparing MixDVAE with the unsupervised method Mixit and with two weakly-supervised methods based on vanilla NMF and NMF with temporal extensions. Please, see Sections 6.2/Baselines and 6.3.
>
> **"Overall, the contributions are not very clear to me. It would be good to include an itemized contributions list at the end of section 1.2."**
>
> Thank you for this suggestion. Following the suggestion of the reviewer, we have included a short summary of the contributions in the form of a list at the end of Section 1.2.
>
> **"If possible you can shorten the paper. It would be better to highlight the advantages of the approach rather than focusing on the technical details."**
>
> Please see our answer to the similar remark above. Again, we would like to mention that, in our opinion, the main manuscript contains only the most important technical information about the methodological background, the proposed MixDVAE model, and the  corresponding VEM algorithm. Many details of the algorithm derivation are already given in the Appendix.

---

### Review · Reviewer_PS4D · 2023-07-27

**Summary Of Contributions:**

The authors propose a latent-variable generative approach for modeling the multi-source trajectory. The proposed approach is called MixDVAE, which extends the existing sequential variational models (DVAEs as mentioned by the authors) with the ability to model multiple sources. The optimization is done first by pre-training a DVAE system for modeling single source data, and then by using the variational expectation-maximization algorithm to estimate the assignment variable and the representation of each source.

The authors demonstrate the effectiveness of their methods in two tasks — multi-object tracking and single-channel audio source separation.

**Audience:**

Yes

**Broader Impact Concerns:**

I did not find anything that could have potential negative impact to the society.

**Claims And Evidence:**

Yes

**Requested Changes:**

Overall, the paper is easy to follow, and the overall idea sounds reasonable.

I would recommend the authors to address the bullet points regarding the potential weakness I listed in the ‘strength/weakness’ section.

Finally, as a generic method described in a long paper, the authors could also consider applying their task to one more domain (e.g., processing text), and see how things are going on.

**Strengths And Weaknesses:**

Strength:

— The method is a natural extension of the sequential VAE, and being able to model the complicated multi-source mixture of trajectories.

— The proposed variational expectation maximization algorithm looks sound.

— The proposed methods show overall better or comparable performance compared to the baseline systems that were compared with.


Weakness:



— The impact of the DVAE is not carefully studied. How critical is the pre-trained DVAE? If pre-trained DVAE is of relatively low quality, will it heavily affect the whole system? I would suggest the authors study DAVE pre-training in different data scales, and show their effect on the whole system. Also, I would recommend the authors compare the whole system with/without fine-tuning the pre-trained DVAE.



— The complexity of the algorithm should be discussed. Overall, the algorithm’s computation positively relates to the number of iterations of the EM, number of sources, number of assignments, and also length of the sequence. How’s the performance evolve as the number of EM iterations goes on? What’s the computation given a different number of sources and sequence length?

The authors would need some more ablation study to demonstrate the proposed method can scale up to large datasets, or to state computation cost is one bottleneck. The pre-training of DVAE should also be considered as part of the computation cost.



— It’s unclear to me if the presented approach is a ‘plug-in’ system, or needs some adaptation for using a different sequential VAE system.



— The generalization to unknown sources is unknown. Basically, the whole system assumes the number of sources are already known. What if there is a new type of source that is unknown during inference, or the number of sources is beyond N? What if there are some unseen types of data that during inference but not in training?

---

> ### Author Response · Authors · 2023-09-06
> **Response to Reviewer PS4D (Part 1)**
>
> We sincerely appreciate the valuable comments and feedback provided by the reviewer. We are glad to see that the reviewer found the proposed algorithm sound, the paper easy to follow, the overall idea reasonable, and noted that the performance of the proposed method is better or comparable to the baselines. We address the concerns raised by the reviewer as below.
>
> **"The impact of the DVAE is not carefully studied. How critical is the pre-trained DVAE? If pre-trained DVAE is of relatively low quality, will it heavily affect the whole system? I would suggest the authors study DVAE pre-training in different data scales, and show their effect on the whole system. Also, I would recommend the authors compare the whole system with/without fine-tuning the pre-trained DVAE."**
>
> Following the remark of the reviewer, we have conducted a new ablation study on the influence of the pre-trained DVAE quality by pre-training the DVAE model at different data scales, for both the MOT and the SC-ASS tasks. The results are presented and discussed in Appendix I of the revised version (Section I.1). As for the influence of with/without fine-tuning the DVAE model during the E-Z step on the whole system, results were already presented in Appendix I of the initial version, but they were only for the MOT task. We have completed with new results on the SC-ASS task, see Section I.2.
>
> **"The complexity of the algorithm should be discussed. Overall, the algorithm’s computation positively relates to the number of iterations of the EM, number of sources, number of assignments, and also length of the sequence. How’s the performance evolve as the number of EM iterations goes on? What’s the computation given a different number of sources and sequence length? The authors would need some more ablation study to demonstrate the proposed method can scale up to large datasets, or to state computation cost is one bottleneck. The pre-training of DVAE should also be considered as part of the computation cost."**
>
> Following the suggestion of the reviewer, a discussion on the complexity of the algorithm has been added in Appendix J of the revised version. In particular, we report that the performance of MixDVAE first increases and then stabilizes with the number of EM iterations. For a given number of EM iterations, we have measured the average processing time for one sequence with different number of sources and sequence lengths. We acknowledge that the computation cost of the VEM algorithm may be a limitation of the proposed method. We have added a sentence in this line in the third paragraph of Section 1.2. As for the computational cost of the DVAE pre-training, we agree with the reviewer that it should also be considered as part of the computation cost. However, because the DVAE is pre-trained on a single-source dataset of moderate size, we can consider that the DVAE pre-training cost is much lower than the cost of training a supervised model on a massive multi-source dataset (please, see our answer to Remark 2 of Reviewer Bi6u; and see the clarifications made in the third paragraph of Section 1.2). This point is also now discussed in Appendix J.
>
> **"It’s unclear to me if the presented approach is a ‘plug-in’ system, or needs some adaptation for using a different sequential VAE system."**
>
> The proposed MixDVAE model is versatile in many respects, and one of them is that it can be applied to various types of DVAE models. One can flexibly choose a DVAE model depending on the specific application scenario and `plug' it into the MixDVAE model. In our paper, for the purpose of generality, the MixDVAE model is presented in its most general form, i.e.~using the most general equations characterizing the  DVAE class of models (Eqs (4) to (8)). However, when using a specific DVAE model (e.g., SRNN, as we did in our experiments), the probabilistic dependencies between the source vectors $\mathbf{s}_{1:T}$ and the latent vectors $\mathbf{z}_{1:T}$ might change (be simplified). Therefore, slight adaptations are required in the MixDVAE algorithm compared to the general equations in Section 4. Obviously, we cannot present the adaptation of these equations to every DVAE model, and we naturally stick to the most general derivations in the paper.

---

> ### Author Response · Authors · 2023-09-06
> **Response to Reviewer PS4D (Part 2)**
>
> **"The generalization to unknown sources is unknown. Basically, the whole system assumes the number of sources are already known. What if there is a new type of source that is unknown during inference, or the number of sources is beyond N? What if there are some unseen types of data that during inference but not in training?"**
>
> The reviewer raises two different (but non exclusive) important issues: a varying number of sources and an unknown type of source. In this work, we assume a known and fixed number of sources during inference. For example, we do not account for the 'birth' and 'death' of source trajectories, as is usually done in complete MOT systems in computer vision. Many methods in the MOT literature use ad-hoc rules to manage the birth and death of tracks. These rules are most often independent of the tracking method itself and are validated mostly experimentally. This is discussed in Section 1.3. On the audio source separation side, the vast majority of the existing methods assume a known and fixed number of sources, and a small number of them have an external method estimating the number of active sources (in other words, a source counting method/module). The goal of our paper is not to present a complete 'functional' MOT or audio source separation system including all these complementary modules, but it is rather to show that a DVAE can be combined with an observation-to-source  assignment model for multi-source trajectory estimation in a weakly supervised framework. Therefore, we develop this methodology assuming a fixed and known number of sources. As for the generalization to an unknown type of sources, we acknowledge that this can be an important issue in some applications, but this is a fundamental problem in machine learning in general (the generalization to data distributions not considered during training). So far we have not consider this problem in the proposed model. We acknowledge this limitation and leave if for future work.
>
> **"Finally, as a generic method described in a long paper, the authors could also consider applying their task to one more domain (e.g., processing text), and see how things are going on."**
>
> We kindly ask the reviewer if this suggestion conditions their acceptance of the paper or not. The reason for asking this is that we would prefer not to add new experiments in the paper, for several reasons.
> First, conducting and reporting new experiments on data that we are not used to manipulate (we mostly work with video and audio signals) would be quite long and the delay for revision is quite short in comparison. Therefore, we could not fit this request in the rebuttal timeline (especially given that, following the suggestions of the two other reviewers, we have conducted and reported experiments with additional baseline methods), and we would need more time to do things properly. Second, even if this may sound irrelevant to the reviewer, Reviewer Bi6fu already finds the paper too long and suggested to shorten it (we may not implement this suggestion, but at the very least, we would prefer not making it longer). Finally, and importantly, we would like to point the attention of the reviewer to the fact that a model that is able to address the multi-source trajectory modeling and separation problem in domains such diverse as MOT in computer vision and audio source separation is not common in the literature. To our knowledge, we are not aware of any other paper presenting a generic method validated in these two distinct tasks, which are generally addressed in two different research communities. Even our own baselines needed to be adapted. In our opinion, this is already a very good illustration of the versatility and flexibility of the proposed model and algorithm.

---

### Review · Reviewer_UTAB · 2023-08-23

**Summary Of Contributions:**

Finding scalable methods to track individual temporal sources in a mixed signal is still an unsolved problem. The authors exemplify this through two applications: audio source separation and multi-object tracking. There exists a trove of other applications that would be interesting to analyze further with this method. The introduced method adds a DVAE, in this study an SRNN, into a larger framework optimized through EM. The true power of the method is that it allows for only training on the distinct sources, hence, there is no need for a dataset that holds both the separate sources in combination with the mixed sources.

**Audience:**

Yes

**Broader Impact Concerns:**

No concerns.

**Claims And Evidence:**

Yes

**Requested Changes:**

1. Ablation study of the model and the training method (where possible)
2. An explanation of the training complexity
3. An analysis of the problem with fine-tuning the SRNN during optimization

**Strengths And Weaknesses:**

The method is scalable (weakly supervised) as one can train on independent source data and combine the instances of models. The results are very convincing since the methodology compares (on certain metrics) and proves superior (on other metrics) to supervised alternatives.

It is slightly difficult for the reader to discern whether a simpler framework could have done equally well. The study could become much stronger if the authors included an ablation study on the latent variables of the model.

Since it is a rather complex methodology proposed, it would be beneficial to the reader to understand training complexity. It would also be good to understand the practical way of choosing the variation matrix ratio.

When fine-tuning the SRNN it seems like performance has a tendency to decrease. You write:
"The possible reason is that fine-tuning could make the model more sensible to detection noise, and lead to a generative model with worse performance."
It would be nice to understand this better, and it would be interesting to see it visualized through experiments.

---

> ### Author Response · Authors · 2023-09-06
> **Response to Reviewer UTAB**
>
> We sincerely appreciate the constructive comments and valuable feedback provided by the reviewer. We are glad to see that the reviewer found our proposed method interesting to be applied to a trove of other applications and the obtained results convincing. We address the concerns raised by the reviewer as below.
>
> **"It is slightly difficult for the reader to discern whether a simpler framework could have done equally well. The study could become much stronger if the authors included an ablation study on the latent variables of the model.''**
>
> We are not sure to completely understand what the reviewer means by "an ablation study on the latent variables of the model". We suppose that this discussion is about the dynamical model used for single-trajectory modeling, i.e.~the DVAE model, and the reviewer is interested in seeing how a simpler deep dynamical model without latent variables would compete with the DVAE. This is a very interesting point.
> We have thus conducted new experiments with a deep auto-regressive (AR) dynamical model without latent variables, which is implemented in practice by a simple LSTM layer. This deep AR / LSTM baseline is now introduced in Sections 5.2/Baselines and 6.2/Baselines of the revised version, and the results are reported and discussed in Sections 5.3 and 6.3. We found out that MixDVAE with SRNN performs better than this new baseline deep AR model, with a similar size of the LSTM layers in both models.
> Considering 'a simpler framework', as suggested by the reviewer, please note that we already had included in the paper the comparison with a  linear dynamical model, which is the Variational Kalman Filter (VKF). And we have demonstrated experimentally that the proposed MixDVAE model shows better performance on both the MOT and SC-ASS tasks. In summary, thanks to the reviewer's suggestion, we now have one latent-variable non-deep (linear) baseline (VKF) and one deep non-latent-variable baseline (deep AR).
>
> **"Since it is a rather complex methodology proposed, it would be beneficial to the reader to understand training complexity.''**
>
> The proposed method is based on (i) the pre-training of a DVAE model on a single-source dataset, and (ii) the MixDVAE variational EM algorithm for source tracking. The computational complexity issue mostly concerns (ii), therefore we assume the reviewer is interested in the overall complexity of the proposed method, and not only in the `training complexity'.  We have added a discussion on the computation complexity in the revised version in Appendix J. Specifically, the computation time of the proposed MixDVAE algorithm mainly depends on three factors, the number of EM iterations, the number of sources to track and the sequence length. We first studied the evolution of the performance of MixDVAE with the number of EM iterations. We find that the performance of MixDVAE first increases and then stabilizes over the EM iterations. The optimal number of EM iterations can be obtained by a grid search over these parameters. For a given number of EM iterations, we find that the computation time increases almost linearly with the number of sources and the sequence length. We acknowledge that the computation complexity can be a limitation for MixDVAE, especially for long sequences with a large number of sources. Beyond the discussion of Appendix J, we have added a sentence in this line in the third paragraph
> of Section 1.2 to announce that clearly in the general presentation of our model.
>
> **"It would also be good to understand the practical way of choosing the variation matrix ratio."**
>
> The value of the variance matrix ratio is related to the measurement errors and depends on the specific task and dataset. For example, in the MOT task, this ratio is related to the errors between the detection bounding box and the true source bounding box. While in the source separation task, this ratio is related to the errors between the mixture signal and the single source signal at a given time-frequency bin. In practice, this value is selected with an empirical grid search.
>
> **"When fine-tuning the SRNN it seems like performance has a tendency to decrease. You write: "The possible reason is that fine-tuning could make the model more sensible to detection noise, and lead to a generative model with worse performance." It would be nice to understand this better, and it would be interesting to see it visualized through experiments."**
>
> In order to gain a deeper comprehension of the impact of fine-tuning, we have included an analysis of two tracking examples for the MOT task with and without fine-tuning the SRNN, in Appendix I.2 of the revised version. We hope this addresses the concern raised by the reviewer.

---

### Decision · Action_Editor_dA8b · 2023-11-27

**Recommendation:** Accept as is

**Comment:**

The authors engaged with the reviewers in the discussion period providing a very thorough rebuttal, addressing most of the raised concerns. Specifically, apart from adding several clarification notes, the author incorporated baselines (e.g. deep AR, LSTM, Mixit and NMF with variants), included ablation studies showing how the performance of the algorithm changes with the quality of the pre-trained DVAEs and detained how fine-tuning the DVAE at inference time affects performance. The updated version of the paper is clearer describing the strengths and limitations of the approach. The quality of the manuscript has been significantly improved.

Both reviewers PS4D and UTAB lean to accept the paper and consider that all their concerns were addressed by the authors. Reviewer Bi6u leans to rejecting it, their main concern being the relevance of the results in source separation.

Specifically, the reviewer pointed out that the performance of the proposed method is very low compared to the state of the art, even considering source-independent methods. One could add that the problem tackled here (separating speech from a musical instrument) is not a particularly challenging task, as the spectral characteristics and dynamics of the two sources are quite different. The authors acknowledged this point in their response, and stated that the purpose of this example is to show the versatility of the approach, and to show that the method works beyond MOT. While the AE agrees with the points raised by the reviewer, it also finds the explanation provided by the authors satisfactory.

It is certainly positive that the model requires a small amount of data. However, it is hard to make this case when using speech signals for which huge amounts of (cheap) data is available. Also, many methods work by synthetically generating the mixture signals requiring the same type of data than models training source-specific models (but more computational cost). This method however could be used in applications for which training data is not that easily available. The paper would make a stronger case in one of these scenarios. Having said that, the baselines included by the authors operate in the low data regime and are the strongest methods in it.Thus the results are not insignificant and certainly serve the purpose of showing that the MixDVAE is able to model the dynamics of complex signals and perform the separation (beyond MOT).

Below some other aspects discussed in the reviewing process.

One limitation pointed out by Reviewer PS4D is that the model cannot handle unknown types of signals, meaning a DVAE needs to be pre-trained using available data similar to the one that will be encountered at inference time. This is an natural assumption for both applications presented in this work, and seems reasonable to the AE.

Reviewers PS4D wanted an experimental study showing how the performance of the model degrades with the quality of the pre-trained DVAE(s). The authors included a detailed ablation study addressing this point by pre-training the DVAE model at different data scales, for both examples presented in this work. Results indicate that the degradation is very smooth, showing robustness with respect to the source models.

Another concern raised by reviewers PS4D and UTAB is the effect of fine-tuning the DVAE model during the E-Z step of the algorithm. The original manuscript already included some initial results on this showing a reduction in overall performance when including the fine-tuning step. The authors further added experiments to study the root causes of this effect. Given that this step is certainly costly and makes the algorithm less practical (e.g. harder to deploy), the authors should make clear that this is the recommended way of using it. Particularly in Section 4.

Reviewer PS4D noted that when using a specific DVAE model, small adaptations are required in the MixDVAE algorithm. However, they don’t find this a reason to reject the work. The AE agrees with this view.

The computational complexity is one of the limitations of the proposed method. This is clearly acknowledged in the paper. In response to inquiries by Reviewer UTAB the authors revised the manuscript to include a study that shows how the performance of the proposed MixDVAE changes with the computational complexity (mainly the number of EM iterations, the number of sources to track and the sequence length at inference time). The AE finds the analysis insightful and Reviewer UTAB found it satisfactory.

Given the above, the AE recommends accepting the work and considers the paper a good contribution that will be of interest to many of the readers of TMLR.

**Audience:**

The paper proposed a model that extends to dynamic VAEs to the multi source setting, it also provides a competitive model for the multi-object tracking problem, which is very relevant in computer vision.

**Claims And Evidence:**

The paper presents a new model MixDVAE to model multiple sources under the assumption that each source behaves
independently. The model consists of a dynamic VAE (DVAE) is used to model the dynamics of each source in isolation and a discrete latent variable is used to assign the observations to the corresponding sources. The DVAE(s) (could be a single model instantiated for each source) are pre-trained using single-source data. The learning process of the joint model is solved via a VEM algorithm combining structured mean-field approximation and amortized inference principles. Experimental validation of the proposed method is presented in two domains: MOT and single channel source separation.

The paper is well written and the proposed method is novel. All reviewers agree that the algorithm is well presented and sound. Experimental results show that the algorithm is versatile and produces sound results in two domains: MOT and single channel source separation.The proposed approach is able to perform source separation in two difficult settings. The results on MOT are particularly good.